# Precise Learning Curves and Higher-Order Scaling Limits for Dot Product Kernel Regression

**Lechao Xiao***
Google Research, Brain Team
xlc@google.com

**Hong Hu***
University of Pennsylvania
huhong@wharton.upenn.edu

**Theodor Misiakiewicz***
Stanford University
misiakie@stanford.edu

**Yue M. Lu**
Harvard University
yuelu@seas.harvard.edu

**Jeffrey Pennington**
Google Research, Brain Team
jpennin@google.com

## Abstract

As modern machine learning models continue to advance the computational frontier, it has become increasingly important to develop precise estimates for expected performance improvements under different model and data scaling regimes. Currently, theoretical understanding of the learning curves that characterize how the prediction error depends on the number of samples is restricted to either large-sample asymptotics ($m \to \infty$) or, for certain simple data distributions, to the high-dimensional asymptotics in which the number of samples scales linearly with the dimension ($m \propto d$). There is a wide gulf between these two regimes, including all higher-order scaling relations $m \propto d^r$, which are the subject of the present paper. We focus on the problem of kernel ridge regression for dot-product kernels and present precise formulas for the mean of the test error, bias, and variance, for data drawn uniformly from the sphere with isotropic random labels in the $r$th-order asymptotic scaling regime $m \to \infty$ with $m/d^r$ held constant. We observe a peak in the learning curve whenever $m \approx d^r/r!$ for any integer $r$, leading to multiple sample-wise descent and nontrivial behavior at multiple scales. We include a colab[2] notebook that reproduces the essential results of the paper.

## 1 Introduction

Modern machine learning has entered an era in which scaling is arguably the most critical ingredient to improve performance. Recent breakthroughs such as GPT-3 [24] and PaLM [11] have demonstrated that performance of various learning algorithms improves in a *predictable* manner as the amount of data and computational resources used in training increases. The functional relationships between performance and resources are loosely referred to as learning curves. While extrapolation of empirical learning curves is widely used to make predictions about how a model might perform when extra resources become available, a rigorous theoretical understanding is lacking. A fundamental obstacle in developing a detailed theoretical model of such learning curves is that they depend on many moving parts, e.g. the data distribution, the network architecture, the training algorithm, among others. In addition, even in the simplest possible settings, the learning curves can exhibit non-trivial structure that naive scaling laws fail to model, e.g. the well-known double-descent phenomenon [7, 3].

In the past couple years, a large amount of effort from the community has improved our theoretical understanding of such phenomena and in some cases precise characterizations of learning curves

---

*LX, HH and TM contributed equally. HH's work was done while at Harvard University.

[2]Available at: https://tinyurl.com/2nzym7ym

36th Conference on Neural Information Processing Systems (NeurIPS 2022).

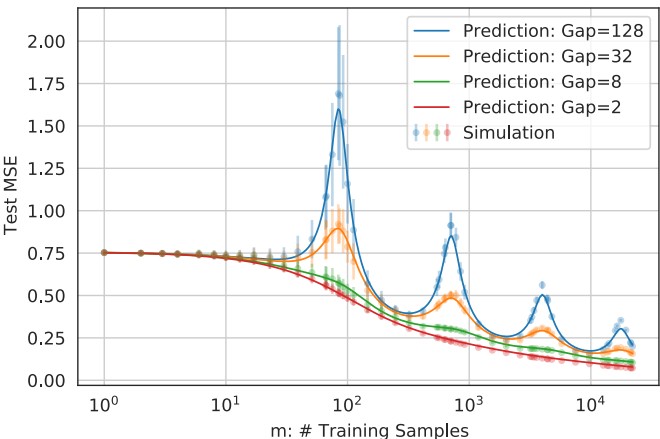

Figure 1: **Precise Sample-wise Learning Curves for One-hidden Layer CNN kernels.** The theoretical predictions (Eq. (18), solid lines) agree with finite-size simulations (markers) across several orders of magnitude and captures cases in which the curves are relatively simple (**monotonically decreasing**, small spectral gap) and complex (**multiple-descent**, large spectral gap). Simulations are obtained from kernel regression with one-layer CNN kernels averaged over 50 runs. The input is of shape $d = d_0 \times p$ with size $d_0 = 14$ and number of patches $p = 6$. We vary the kernels by varying the ratio (aka spectral gap) between consecutive eigenspaces, where the ratio $\mathrm{Gap} \in [\mathbf{2}, \mathbf{8}, \mathbf{32}, \mathbf{128}]$.

have been obtained (see e.g., [21, 1, 31, 34, 28]). These results have helped clarify several puzzling empirical observations, such as the origin of the double-descent peak [2, 27, 13] and linear trends between in- and out-of-distribution generalization performance [38, 39, 30], among many others. However, the precise predictions from many of these analyses have been possible only in the linear high-dimensional scaling regime in which the number of training samples $m$ scales linearly with the dimension $d$, i.e. $m \propto d$. In these asymptotics, the model's effective capacity is limited to linear functions of the features. In contrast, many state-of-the-art models operate in a regime where the amount of data is much larger than the data dimensionality; for example, large text corpora can contain trillions of tokens, whereas the effective input dimensionality of language models is at most millions . Therefore, going beyond the linear scaling regime ($m \propto d$) to higher-order scaling regimes ($m \propto d^r$) is essential in improving our understanding of modern machine learning systems, and is the focus of the current paper.

Several works have investigated the behavior of the learning curves for nonlinear scalings in the dot-product kernel or random features setting, but they have done so only in the noncritical regime where $m \not\propto d^r$ [17, 32, 26]. [8, 10] also derive the closed-form predictions of the learning curves for both the critical and the noncritical scalings, but they have done so via nonrigorous statistical physics methods and a "Gaussian equivalence conjecture" [12, 22, 16, 18, 19, 20]. Rigorously extending these results to include the critical regime $m \propto d^r$ is nontrivial, both from the technical perspective, namely, proving a "Gaussian equivalence conjecture", and also from the phenomenological perspective, as we shall see the critical behavior induces nonmonotonicity and multiple sample-wise descents.

In this work, we obtain precise formulas for the sample-wise learning curves in the kernel ridge regression setting for a family of dot-product kernels for spherical input data in the polynomial scaling regimes $m \propto d^r$ for all $r \in \mathbb{N}^*$. This family of kernels includes the neural network Gaussian Process (NNGP) kernels and Neural Tangent Kernels (NTK) associated with multi-layer fully-connected networks or convolutional networks. Both kernels serve as important starting points towards a deeper understanding of neural networks as they often capture the first order learning dynamics of neural networks in certain scaling limits [23, 25, 4].

## 1.1 Contributions

Our primary contributions are to establish the following, for data drawn uniformly from the sphere:

1. The empirical spectral density of the Gram matrix induced by degree-$r$ spherical harmonics converges to a Marchenko-Pastur distribution when $(d^r/r!)/m$ converges to a positive constant as $d \to \infty$ (Theorem 1);

2. A precise closed-form formula for the sample-wise learning curves for dot-product kernel regression when $m \propto d^r$ for all $r \in \mathbb{N}^*$ as $d \to \infty$ (Theorem 2);

3. Empirically, the theoretical predictions agree with finite-size simulations surprisingly well even in the strong finite-size correction regime (Fig. 1);

4. An extension of the above results to convolutional kernels (Section 5).

Finally, we note that our results also assume the high-degree coefficients of the label function to be random and isotropic; see Eq. (11). It remains an open question to prove similar results[3] when the label function is deterministic.

## 2 Notation and Setup

Let $\mathcal{X} = \mathbb{S}_{d-1}$ denote the input space, where $\mathbb{S}_{d-1}$ is the unit sphere in $\mathbb{R}^d$ and $\mathcal{X}$ is equipped with the normalized uniform measure $\sigma$. We use $\Delta_d$ to represent any quantity (a scalar, vector or a matrix) with $|||\Delta_d||| \to 0$ as $d \to \infty$ (in probability if $\Delta_d$ is stochastic), where $||| \cdot |||$ can be the absolute value of a scalar, the norm of a vector or the operator norm of a matrix.

Let $\boldsymbol{X} \in \mathbb{R}^{m \times d}$ be the training inputs where the $i$-th row of $\boldsymbol{X}$ is $\boldsymbol{x}_i^\top$. We assume $\{\boldsymbol{x}_i\}_{i \in [m]}$ is sampled uniformly, iid from $\mathcal{X}$. The label function $f : \mathbb{S}_{d-1} \to \mathbb{R}$ will be defined in Section 4. Let $K = K^{(d)}$ be a dot-product kernel defined on $\mathbb{S}_{d-1} \times \mathbb{S}_{d-1}$, i.e., $K(\boldsymbol{x}, \boldsymbol{x}') = h(\boldsymbol{x}^\top \boldsymbol{x}')$ for some function $h \in [-1, 1] \to \mathbb{R}$. We assume $h$ has the following decomposition

$$h(t) = \sum_{k=1}^{\infty} \hat{h}_k^2 P_k(t), \quad \text{with} \quad \sum_{k=1}^{\infty} \hat{h}_k^2 < \infty, \tag{1}$$

where $P_k$ is the $k$-th order Legendre polynomials in $d$ dimensions. For simplicity, we assume $\hat{\boldsymbol{h}} = (\hat{h}_k)_{k \geq 1}$ is a sequence that is independent of $d$ and $\hat{h}_k \neq 0$ for all $k \leq k_0$ where $k_0$ is sufficiently large. As such, we can decompose the kernel function using sperical harmonics,

$$K(\boldsymbol{x}, \boldsymbol{x}') = \sum_{k=1}^{\infty} \sigma_k^2 \sum_{l \in [N(d,k)]} Y_{k,l}(\boldsymbol{x}) Y_{k,l}(\boldsymbol{x}') = \sum_{k=1}^{\infty} \sigma_k^2 Y_k(\boldsymbol{x})^\top Y_k(\boldsymbol{x}'), \tag{2}$$

where $Y_{k,l}$ is the $l$-th spherical harmonic of degree $k$, $N(d,k) = d^k/k! + O(d^{k-1})$ is the total number of degree $k$ spherical harmonics in $d$ dimensions, $\sigma_k^2 = \hat{h}_k^2/N(d,k)$ is the eigenvalue of $Y_{kl}$, and $Y_k(\boldsymbol{x})$ is the column vector $[Y_{k,l}(\boldsymbol{x})]_{l \in [N(d,k)]}^\top$. We also denote by $Y_k(\boldsymbol{X})$ the $m \times N(d,k)$ matrix whose $i$-th row is $Y_k(\boldsymbol{x}_i)^\top$.

## 3 Structure of the Empirical Kernel and Marchenko-Pastur Distribution

The structure of the empirical kernel matrix $K(\boldsymbol{X}, \boldsymbol{X})$ plays a critical role in characterizing the sample-wise test error for the kernel ridge regressor associated to $K$. We assume the training set size scales polynomially, i.e. $m \sim d^r$ for some positive integer $r \in \mathbb{N}^*$. Decompose this kernel into low-, critical- and high-frequency modes as follows,

$$K(\boldsymbol{X}, \boldsymbol{X}) = \sum_{k<r} \sigma_k^2 Y_k(\boldsymbol{X}) Y_k(\boldsymbol{X})^\top + \sigma_r^2 Y_r(\boldsymbol{X}) Y_r(\boldsymbol{X})^\top + \sum_{k>r} \sigma_k^2 Y_k(\boldsymbol{X}) Y_k(\boldsymbol{X})^\top. \tag{3}$$

The low- and high-frequency parts have simple structures since $N(d,k)/m$ either diverges to infinity or converges to zero with rate as least $d^{\pm 1}$, yielding concentration that results in significant simplification. To be precise, for high-frequency modes $k > r$, $Y_k(\boldsymbol{X})$ is a "fat" matrix and

---

[3]See Sec. 6 for empirical evidences in favor of these results.

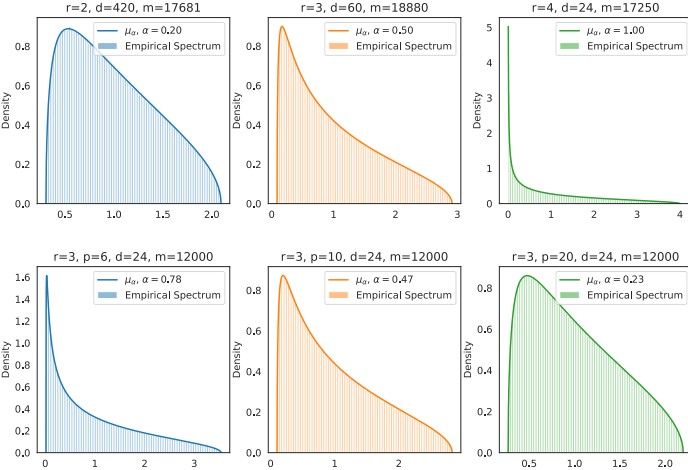

Figure 2: **Marchenko-Pastur Distribution of Spherical Harmonics.** Top: the empirical distribution of product kernels $Y_r(\boldsymbol{X})Y_r(\boldsymbol{X})^\top/N(d,r)$ vs theory prediction from $\mu_\alpha$ for various degrees $r$, input dimensions $d$ and number of samples $m$ as indicated in the titles. Bottom: the empirical distribution of the CNN kernel $Y_r(\boldsymbol{X})Y_r(\boldsymbol{X})^\top/pN(d,r)$ vs theoretical prediction. We fix $r, d$ and $m$ but varying the number of patches $p \in \{6, 10, 20\}$.

$Y_k(\boldsymbol{X})Y_k(\boldsymbol{X})^\top/N(d,k) = \boldsymbol{I}_m + \boldsymbol{\Delta}_d$ where $\boldsymbol{\Delta}_d$ vanishes as $d \to \infty$ [32]. Thus, the high-frequency parts behave like a regularizer in the following sense,

$$\sum_{k>r} \sigma_k^2 Y_k(\boldsymbol{X})Y_k(\boldsymbol{X})^\top = \sum_{k>r} \sigma_k^2 N(d,k)\boldsymbol{I}_m + \boldsymbol{\Delta}_d = \left(\sum_{k>r} \hat{h}_k^2\right)\boldsymbol{I}_m + \boldsymbol{\Delta}_d. \quad (4)$$

On the other hand, when $k < r$, $Y_k(\boldsymbol{X})$ is a $m \times N(d,k)$ "tall" matrix with $N(d,k)/m = O(d^k/m) = O(d^{-(r-k)}) \to 0$. Similarly, Mei et al. [32] show that $Y_k(\boldsymbol{X})^\top Y_k(\boldsymbol{X})/m = \boldsymbol{I}_{N(d,k)} + \boldsymbol{\Delta}_d$, implying that when restricted to the subspace spanned by low-frequency functions $\{Y_{kl}\}_{k<r}$, the regressor associated to the empirical kernel $K(\boldsymbol{X}, \boldsymbol{X})$ acts like a pure multiplicative scaling.

It remains to understand the critical-frequency mode $Y_r(\boldsymbol{X})^\top Y_r(\boldsymbol{X})$. It turns out that if $N(d,k)/m \to \alpha \in (0,\infty)$, then the empirical spectral measure of the random matrix $Y_r(\boldsymbol{X})^\top Y_r(\boldsymbol{X})/m$ converges to the Marchenko-Pastur distribution $\mu_\alpha$, whose density is given by

$$\mu_\alpha(t) = \left(1 - \frac{1}{\alpha}\right)^+ \delta_0(t) + \frac{\sqrt{(\alpha_+ - t)(t - \alpha_-)}}{2\pi\alpha t}\mathbf{1}_{[\alpha_-,\alpha_+]}(t), \text{ where } \alpha_\pm = (1 \pm \sqrt{\alpha})^2. \quad (5)$$

where $\delta_0(t) = 0$ if $t \neq 0$ else 1. See Fig. 2 for visualizations of $\mu_\alpha$. The $r = 1$ case is obvious as $Y_1(\boldsymbol{X}) = c_d\boldsymbol{X}$ for some normalizing constant $c_d$ and it is clear $\frac{1}{m}Y_1(\boldsymbol{X})^\top Y_1(\boldsymbol{X}) = \frac{c_d^2}{m}\boldsymbol{X}^\top\boldsymbol{X}$ converges to the Marchenko-Pastur distribution $\mu_\alpha$ if $d/m \to \alpha \in (0,\infty)$ as $d \to \infty$ [37]. Our first result show that this result continues to hold for all degrees.

**Theorem 1.** *For fixed $r \in \mathbb{N}$ and $\alpha \in (0,\infty)$, if $N(d,r)/m \to \alpha \in (0,\infty)$ as $d \to \infty$, then the empirical spectral distribution of $\frac{1}{m}Y_r(\boldsymbol{X})^\top Y_r(\boldsymbol{X})$ converges in distribution to the Marchenko-Pastur distribution $\mu_{\mathrm{MP}(\alpha)}$.*

In the top panel of Fig. 2, we generate the empirical spectra[4] of $\frac{1}{m}Y_r(\boldsymbol{X})^\top Y_r(\boldsymbol{X})$ for various values of $r, d$, and $\alpha$. The Marchenko-Pastur distribution $\mu_\alpha$ perfectly captures the empirical measures of the random matrices $\frac{1}{m}Y_r(\boldsymbol{X})^\top Y_r(\boldsymbol{X})$ for all $r$ considered. We sketch the main steps of the proof of the theorem below; see Appendix B for the whole proof.

---

[4]In the plot, we generate the spectra of the kernel matrix $Y_r(\boldsymbol{X})Y_r(\boldsymbol{X})^\top$ instead of the covariate matrix $Y_r(\boldsymbol{X})^\top Y_r(\boldsymbol{X})$. Although both of them have the same set of non-zero eigenvalues, the former can be easily implemented via Legendre polynomials $P_r(\boldsymbol{x}^\top\boldsymbol{x}')$.

*Sketch of Proof.* From Bai and Zhou [5, Theorem 1.1], it suffices to prove concentration of the following quadratic forms: for every sequence of $N(d,k) \times N(d,k)$ matrices $\{A_d\}$ with operator norm $\|A_d\|_{\mathrm{op}} \leq 1$, the variance

$$N(d,r)^{-2}\mathbb{V}(Y_r(\boldsymbol{x})^\top A_d Y_r(\boldsymbol{x}) - \mathrm{Tr}(A_d)) \to 0 \quad \text{as} \quad d \to \infty. \tag{6}$$

For the purpose of illustration, we assume $A \equiv A_d$ is a diagonal matrix. Then we only need to show

$$N(d,k)^{-2} \sum_{l,l' \in [N(d,k)]} A_{ll} A_{l'l'} (\mathbb{E}_{\boldsymbol{x}} Y_{k,l}^2(\boldsymbol{x}) Y_{k,l'}^2(\boldsymbol{x}) - 1) \to 0. \tag{7}$$

By hypercontractivity of spherical harmonics [6],

$$\mathbb{E}_{\boldsymbol{x}} Y_{k,l}^2(\boldsymbol{x}) Y_{k,l'}^2(\boldsymbol{x}) \leq \left( \mathbb{E}_{\boldsymbol{x}} Y_{k,l}(\boldsymbol{x})^4 \mathbb{E}_{\boldsymbol{x}} Y_{k,l'}^4(\boldsymbol{x}) \right)^{1/2} \leq C_k \mathbb{E}_{\boldsymbol{x}} Y_{k,l}(\boldsymbol{x})^2 \mathbb{E}_{\boldsymbol{x}} Y_{k,l'}(\boldsymbol{x})^2 = C_k, \tag{8}$$

where $C_k$ is some absolute constant. Since $|A_{ll}| \leq \|A\|_{\mathrm{op}} \leq 1$, we can drop any $o(N(d,l)^2)$ pairs of $(l, l')$ in Eq. (7). We show that for the remaining pairs $(l, l')$, the eigenfunctions are asymptotically uncorrelated in the sense

$$\mathbb{E}_{\boldsymbol{x}} Y_{k,l}^2(\boldsymbol{x}) Y_{k,l'}^2(\boldsymbol{x}) = \mathbb{E}_{\boldsymbol{x}} Y_{k,l}^2(\boldsymbol{x}) \mathbb{E}_{\boldsymbol{x}} Y_{k,l'}^2(\boldsymbol{x}) + O(d^{-1}) = 1 + O(d^{-1}) \tag{9}$$

which implies Eq. (7). $\qquad\square$

## 4   Generalization Error of Dot-Product Kernel Regression

In this section, we establish the *average* generalization error for the kernel regression in the asymptotic regime $N(d,r)/m \to \alpha$, for some $\alpha \in (0, \infty)$ and $r \geq 1$ fixed. We assume the label function $f \in L^2(\mathbb{S}_{d-1})$ is given by

$$f(\boldsymbol{x}) = \sum_{k \geq 1} \sum_{l \in [N(d,k)]} \hat{f}_{kl} Y_{kl}(\boldsymbol{x}) = \sum_{k \geq 1} \hat{\boldsymbol{f}}_k^\top Y_k(\boldsymbol{x}), \tag{10}$$

where $\hat{f}_{kl}$ are the "Fourier" coefficients and $\hat{\boldsymbol{f}}_k = [f_{kl}]_{l \in N(d,k)}^\top$. We need to make a technical assumption that for $k', k \geq r$

$$\mathbb{E}\hat{\boldsymbol{f}}_k = \mathbf{0}, \qquad \mathbb{E}\hat{\boldsymbol{f}}_k \hat{\boldsymbol{f}}_k^\top = \frac{\hat{F}_k^2}{N(d,k)} \boldsymbol{I}_{N(d,k)} \quad \text{and} \quad \mathbb{E}\hat{\boldsymbol{f}}_k \hat{\boldsymbol{f}}_{k'}^\top = \mathbf{0}_{N(d,k) \times N(d,k')} \tag{11}$$

i.e. $\hat{\boldsymbol{f}}_k$ is centered with isotropic covariance and $\{\hat{\boldsymbol{f}}_k\}_{k \geq r}$ are mutually uncorrelated. Note that we allow $\hat{f}_{kl}$ to be deterministic for $k < r$. We let $\boldsymbol{F} = (\hat{F}_k)_{\geq 1}$ be a fixed sequence with $\sum_{k \geq 1} \hat{F}_k^2 < \infty$, where $\hat{F}_k^2 = \sum_{l \in [N(d,k)]} \hat{f}_{kl}^2$ for $k < r$. For convenience, set $\hat{F}_{>j}^2 = \sum_{k > j} \hat{F}_k^2$ (similarly for $\hat{F}_{\geq j}^2$, $\hat{F}_{\leq j}^2$, etc.) and use $\boldsymbol{f}$ to denote the random vector $\{\hat{f}_{kl}\}_{kl}$. Given training inputs $\boldsymbol{X}$ and observed labels $\boldsymbol{Y} = f(\boldsymbol{X}) + \boldsymbol{\epsilon}$, where $\boldsymbol{\epsilon} \sim \mathcal{N}(\mathbf{0}, \sigma_\epsilon^2 \boldsymbol{I}_m)$ is the noise, the prediction using kernel function $K$ is given by

$$y(\boldsymbol{x}) = K(\boldsymbol{x}, \boldsymbol{X})(K(\boldsymbol{X}, \boldsymbol{X}) + \lambda \boldsymbol{I}_m)^{-1}(f(\boldsymbol{X}) + \boldsymbol{\epsilon}). \tag{12}$$

Here $\lambda \geq 0$ is the regularization. As such, the *mean* test error over the random labels is given by

$$\mathrm{Err}(\boldsymbol{X}; \lambda, \boldsymbol{F}, \hat{\boldsymbol{h}}) = \mathbb{E}_{\boldsymbol{f}} \mathrm{Err}(\boldsymbol{X}; \lambda, \boldsymbol{f}, \hat{\boldsymbol{h}}) \quad \text{where} \quad \mathrm{Err}(\boldsymbol{X}; \lambda, \boldsymbol{f}, \hat{\boldsymbol{h}}) = \mathbb{E}_{\boldsymbol{x}, \epsilon} |y(\boldsymbol{x}) - f(\boldsymbol{x})|^2. \tag{13}$$

To state our results, we need to introduce two functions $\chi_B$ and $\chi_V$ which are related to the bias and variance in the generalization error,

$$\chi_B(\alpha, \xi) = \int (1 + \xi t)^{-2} \mu_\alpha(t) dt \quad \text{and} \quad \chi_V(\alpha, \xi) = \alpha \xi^2 \int t(1 + \xi t)^{-2} \mu_\alpha(t) dt. \tag{14}$$

Both $\chi_B$ and $\chi_V$ have closed-form representations; see Appendix C.5. Define the effective regularization associated to the $r$-th order scaling to be

$$\xi_r(\hat{\boldsymbol{h}}, \lambda, \alpha) = \frac{\hat{h}_r^2}{\alpha(\lambda + \hat{h}_{>r}^2)} \tag{15}$$

Finally, we define the bias and variance associated to the $r$-th order scaling to be

$$\mathrm{B}_r(\alpha) = \mathrm{B}_r(\alpha; \lambda, \boldsymbol{F}, \hat{\boldsymbol{h}}) = \chi_B(\alpha, \xi_r(\hat{\boldsymbol{h}}, \lambda, \alpha)) \hat{F}_r^2 + \hat{F}_{>r}^2 \tag{16}$$

$$\mathrm{V}_r(\alpha) = \mathrm{V}_r(\alpha; \lambda, \boldsymbol{F}, \hat{\boldsymbol{h}}) = \chi_V \left( \alpha, \xi_r(\hat{\boldsymbol{h}}, \lambda, \alpha) \right) \left( \hat{F}_{>r}^2 + \sigma_\epsilon^2 \right) \tag{17}$$

The following is our main result, which characterizes the test error in the asymptotic regime $m \propto d^r$.

**Theorem 2.** *Let $\alpha \in (0, \infty)$ and $r \geq 1$ be fixed. Assume $N(d, r)/m \to \alpha$ as $d \to \infty$. Then the average test error is given by*

$$\mathrm{Err}(\boldsymbol{X}; \lambda, \boldsymbol{F}, \hat{\boldsymbol{h}}) = \mathrm{B}_r(\alpha; \lambda, \boldsymbol{F}, \hat{\boldsymbol{h}}) + \mathrm{V}_r(\alpha; \lambda, \boldsymbol{F}, \hat{\boldsymbol{h}}) + \Delta_d, \tag{18}$$

*where $\Delta_d \to 0$ in probability.*

### 4.1 Interpretations

We provide some high-level interpretations of the bias term $\mathrm{B_r}$ and variance term $\mathrm{V_r}$.

**The Bias.** From Eq. (16), the regressor learns all low-frequency modes ($k < r$) but none of the high-frequency modes ($k > r$) as the bias $\mathrm{B}_r$ contains no low-frequency modes (i.e. $k < r$) but all high-frequency modes $\hat{F}^2_{>r}$. Importantly, the regressor is progressively learning the critical-frequency mode $Y_r$ as the training size $m = \frac{1}{\alpha} N(d, r)$ increases, i.e. from $\alpha = \infty$ to $\alpha = 0^+$ since $\chi_B(\alpha, \xi_r(\hat{\boldsymbol{h}}, \lambda, \alpha)) \to 1$ if $\alpha \to \infty$ and $\chi_B(\alpha, \xi_r(\hat{\boldsymbol{h}}, \lambda, \alpha)) \to 0$ if $\alpha \to 0^+$. See Fig.3 for the illustration.

**The Variance.** From Eq. (17), the variance term $\chi_V$ treats all high-frequency modes $\hat{F}^2_{>r}$ the same as the noise term $\boldsymbol{\epsilon}$. Moreover, $\chi_V \to 0$ as $\alpha \to 0$ or $\infty$ and is peaked at $\alpha = 1$. The height of the peak depends on the effective regularization $\xi_r$ and it diverges to infinity with rate $\xi_r^{\frac{1}{2}}$ as $\xi_r \to \infty$. Indeed, when $\alpha = 1$ and $\xi^{-1}/2 \leq t \leq \xi^{-1}$, we have $t(1+\xi t)^{-2}\mu_\alpha(t) \propto \xi^{-1/2}$ which implies $\chi_V(1, \xi) \geq \xi^2 \int t(1+\xi t)^{-2}\mu_\alpha(t) \mathbf{1}_{\xi^{-1}/2 \leq t \leq \xi^{-1}} dt \propto \xi^{1/2}$.

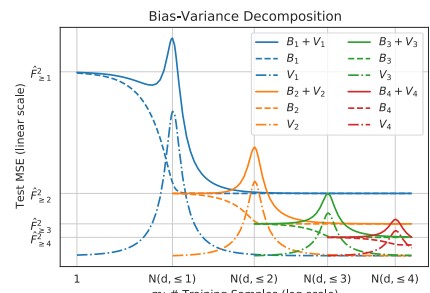

Bias-Variance Decomposition

Finally, Eq. (18) not only gives precise generalization formula (up to a vanishing term $\Delta_d$) when $m \approx N(d, r) \sim d^r$ but also when $d^{r-1+\delta} \lesssim m \lesssim d^{r-\delta}$ (i.e. when "$\alpha = \infty$") and when $d^{r+\delta} \lesssim m \lesssim m^{r+1-\delta}$ (i.e. when "$\alpha = 0^+$") for any $\delta \in (0, 1/2)$. Indeed, in the non-critical scaling regime $d^{r-1+\delta} \lesssim m \lesssim d^{r-\delta}$, $\alpha = N(d, k)/m \to \infty$ as $d \to \infty$ and

**Figure 3: Multi-scale Bias-Variance Decomposition.** Theoretical predictions of the bias and variance from Eq.16 and Eq.17. For each $r$, the variance is non-monotonic and has a peak at $N(d, \leq r) = \sum_{k \leq r} N(d, k)$.

$$\mathrm{B_r}(\alpha = \infty) + \mathrm{V_r}(\alpha = \infty) = \hat{F}^2_{\geq r} + 0 = \hat{F}^2_{\geq r}. \tag{19}$$

As such, the regressor learns all low-frequency modes but none of the critical- and high-frequency modes ($k \geq r$), which is consistent with the result in [17, 32]. A similar argument also shows $\mathrm{B_r}(\alpha = 0^+) + \mathrm{V_r}(\alpha = 0^+) = \hat{F}^2_{>r}$, namely, the regressor also learns the $r$-frequency mode. This observation implies that we can glue together Eq. (18) for $r \geq 1$ and remove all duplicate terms to generate a sample-wise learning curve (LC):

$$\mathrm{LC}(m; \lambda, \boldsymbol{f}, \hat{\boldsymbol{h}}) = \sum_{r \geq 1} \left( \mathrm{B}_r\left(\frac{N(d, r)}{m}; \lambda, \boldsymbol{f}, \hat{\boldsymbol{h}}\right) - (r-1)\hat{F}^2_r \right) + \mathrm{V}_r\left(\frac{N(d, \leq r)}{m}; \lambda, \boldsymbol{f}, \hat{\boldsymbol{h}}\right) \tag{20}$$

where $N(d, \leq r) = \sum_{k=1}^{r} N(d, k)$. The "$-(r-1)\hat{F}^2_r$" term in the above equation is due to the fact that $\hat{F}^2_r$ is over-counted $(r-1)$ many times (one in each $\mathrm{B}_k$ for $k = 1, ..., (r-1)$.) It is worth mentioning that using $\alpha = N(d, \leq r)/m$ rather than $\alpha = N(d, r)/m$ in the variance $\mathrm{V}_r$ captures the finite-size correction more accurately. See Eq. (206) in Appendix.

**Corollary 1.** *If, for $1 \leq r \in \mathbb{N}$, (1) $N(d, r)/m \to \alpha$ for some $\alpha \in (0, \infty)$, or (2) $d^{r-1+\delta} \lesssim m \lesssim d^{r-\delta}$ for some $\delta \in (0, 1/2)$, then*

$$\mathrm{Err}(\boldsymbol{X}; \boldsymbol{f}, \hat{\boldsymbol{h}}, \lambda) = \mathrm{LC}(m; \lambda, \boldsymbol{f}, \hat{\boldsymbol{h}}) + \Delta_d \tag{21}$$

*where $\Delta_d \to 0$ in probability as $d \to \infty$.*

Recall that for each $r$, the variance term $\mathrm{V}_r$ could diverge to infinity as $\xi_r \to \infty$ at $\alpha = 1$. Thus we might expect a peak in the learning curve for each $r$, yielding the multiple-descent phenomenon, as shown in Fig.1. However, such phenomena can disappear by making the heights of the peaks small via choosing $\xi_r$ small. We will discuss this point in the experimental section.

## 4.2 Proof Sketch

The proof of this theorem is quite involved; see Appendix C. For simplicity, we assume the observed labels are noiseless, i.e. $\sigma_\epsilon^2 = 0$. An ingredient is to understand the structure of the operator

$$T_K f(\boldsymbol{x}) = K(\boldsymbol{x}, \boldsymbol{X})(K(\boldsymbol{X}, \boldsymbol{X}) + \lambda \boldsymbol{I}_m)^{-1} f(\boldsymbol{X}). \tag{22}$$

The high-level strategy is as follows. We decompose the function into low-, critical- and high-frequency parts $f = f_{<r} + f_r + f_{>r}$. As such, the test error is roughly

$$\mathrm{Err}(\boldsymbol{X}) \approx \mathrm{Err}_{<r}(\boldsymbol{X}) + \mathrm{Err}_r(\boldsymbol{X}) + \mathrm{Err}_{>r}(\boldsymbol{X}) \text{ where } \mathrm{Err}_r(\boldsymbol{X}) = \mathbb{E}_{\boldsymbol{f}} \mathbb{E}_{\boldsymbol{x}} |T_K f_r(\boldsymbol{x}) - f_r(\boldsymbol{x})|^2,$$

and similarly for $\mathrm{Err}_{<r}(\boldsymbol{X})$ and $\mathrm{Err}_{>r}(\boldsymbol{X})$. The next step is to estimate each part separately.

**Low-frequencies.** Using the fact that the low-frequency parts of the kernel function $K$ is almost an isometric operator on the column space of $Y_{<r}(\boldsymbol{X})$, one can show that $\mathbb{E}_{\boldsymbol{x}} |T_K(f_{<r})(\boldsymbol{x}) - f_{<r}(\boldsymbol{x})|^2 = \Delta_d \to 0$ in probability, *pointwisely*.

**Critical-frequency.** Up to a vanishing term, one can remove all non-critical frequencies in the kernel function $K$ in $T_K$ in the sense of making the following substitutions

$$K(\boldsymbol{x}, \boldsymbol{X}) \to \sigma_r^2 Y_r(\boldsymbol{x})^\top Y_r(\boldsymbol{X})^\top \quad \text{and} \quad K_r(\boldsymbol{X}, \boldsymbol{X}') \to \sigma_r^2 Y_r(\boldsymbol{X}) Y_r(\boldsymbol{X})^\top + \hat{h}_{>r}^2 \boldsymbol{I}_m. \tag{23}$$

Thus, with $\gamma = (\lambda + \hat{h}_{>r}^2)$, $T_K f_r(\boldsymbol{x}) - f_r(\boldsymbol{x}) = Y_r(\boldsymbol{x})^\top \boldsymbol{M}_r(\boldsymbol{X}) \boldsymbol{f}_r + \Delta_d$, where

$$\boldsymbol{M}_r(\boldsymbol{X}) = \left( \boldsymbol{I}_{N(d,r)} - \sigma_r^2 Y_r(\boldsymbol{X})^\top (\sigma_r^2 Y_r(\boldsymbol{X}) Y_r(\boldsymbol{X})^\top + \gamma \boldsymbol{I}_m)^{-1} Y_r(\boldsymbol{X}) \right). \tag{24}$$

Taking expectation with respect to $\boldsymbol{x}$ (using orthogonality of $Y_r(\boldsymbol{x})$) and then with respect to $\boldsymbol{f}_r$,

$$\mathbb{E}_{\boldsymbol{f}_r} \mathbb{E}_{\boldsymbol{x}} |Y_r(\boldsymbol{x})^\top \boldsymbol{M}_r(\boldsymbol{X}) \boldsymbol{f}_r|^2 = \mathbb{E}_{\boldsymbol{f}_r} |\boldsymbol{M}_r(\boldsymbol{X}) \boldsymbol{f}_r|^2 = \hat{F}_r^2 \mathrm{Tr}(\boldsymbol{M}_r^2)/N(d, r). \tag{25}$$

Applying the Sherman–Morrison–Woodbury formula and then Theorem 1,

$$\hat{F}_r^2 \mathrm{Tr} \left( Y_r(\boldsymbol{X})^\top Y_r(\boldsymbol{X})/m + m\sigma_r^2/\gamma \boldsymbol{I}_{N(d,r)} \right)^{-2} /N(d, r) \to \hat{F}_r^2 \int \frac{\mu_\alpha(t)}{(t + \xi_r)^2} dt.$$

**High-frequencies.** The cross term $\mathbb{E}_{\boldsymbol{f}} \mathbb{E}_{\boldsymbol{x}} T_K f_{>r}(\boldsymbol{x}) f_{>r}(\boldsymbol{x}) = \Delta_d$ and thus

$$\mathbb{E}_{\boldsymbol{f}, \boldsymbol{x}} |T_K f_{>r}(\boldsymbol{x}) - f_{>r}(\boldsymbol{x})|^2 = \mathbb{E}_{\boldsymbol{f}, \boldsymbol{x}} |T_K f_{>r}(\boldsymbol{x})|^2 + \mathbb{E}_{\boldsymbol{f}, \boldsymbol{x}} |f_{>r}(\boldsymbol{x})|^2 + \Delta_d. \tag{26}$$

The second term is equal to $\hat{F}_>^2$. The calculation of the first term is similar to that of the critical frequency above (namely, we remove all high-/low-frequency components in $K$.)

## 5 Convolutional Kernels

### 5.1 One hidden layer

Our analysis can be extended to analyzing NNGP kernel and NT kernel for one-layer convolution [35, 36, 43]. In this case, we assume the input space is $\mathcal{X} = \mathbb{S}_{d_0-1}^p$, where $d_0$ is the dimension of a patch, $p$ is the number of patches, and $d = pd_0$ is the total dimensions of the inputs. The measure associated to $\mathcal{X}$ is the product of the uniform measure on $\mathbb{S}_{d_0-1}$. We assume that both the filter size and stride of the convolution are equal to $d_0$. As such, after the first convolutional layer, the input is reduced to a vector of dimension $p$. We then apply a non-linearity and a dense layer to map this $p$-dimensional vector to a scalar. The NNGP and NT kernel have the following general form. Let $\boldsymbol{x} = (\boldsymbol{x}_i)_{i \in [p]} \in \mathcal{X}$, where $\boldsymbol{x}_i \in \mathbb{S}_{d_0-1}$ is the $i$-th patch

$$K(\boldsymbol{x}, \boldsymbol{x}') = \frac{1}{p} \sum_{i \in [p]} h(\boldsymbol{x}_i^\top \boldsymbol{x}_i') = \frac{1}{p} \sum_{i \in [p]} \sum_{k \geq 1} \hat{h}_k^2 P_k(\boldsymbol{x}_i^\top \boldsymbol{x}_i') = \sum_{k \geq 1} \frac{\sigma_k^2}{p} \sum_{i \in [p]} \sum_{l \in [N(d_0, k)]} Y_{kl}(\boldsymbol{x}_i) Y_{kl}(\boldsymbol{x}_i').$$

Denote $Y_{kl}^{(i)}(\boldsymbol{x}) = Y_{kl}(\boldsymbol{x}_i)$ and $Y_k(\boldsymbol{x}) = [Y_{kl}^{(i)}(\boldsymbol{x})^\top]_{l \in [N(d_0,k)], i \in [p]}^\top$. Then $Y_k(\boldsymbol{x})$ is the degree $k$ spherical harmonics associated to this kernel, which span a space of dimension $pN(d_0, k)$.

**Theorem 3.** *Let $r \in \mathbb{N}^*$ and $\alpha \in (0, \infty)$ be fixed. If $pN(d_0, r)/m \to \alpha \in (0, \infty)$ as $d_0 \to \infty$ and the rows of $\boldsymbol{X}$ are sampled uniformly, iid from $\mathbb{S}^p_{d_0-1}$, then the empirical spectral distribution of $\frac{1}{m} Y_r(\boldsymbol{X})^\top Y_r(\boldsymbol{X})$ tends to the Marchenko-Pastur distribution $\mu_\alpha$ as $d_0 \to \infty$.*

The assumptions on the label function are similar to that of dot-product kernel, e.g.[5]

$$f(\boldsymbol{x}) = \sum_{k \geq 1} \boldsymbol{f}_k^\top Y_k(\boldsymbol{x}), \quad \text{with} \quad \boldsymbol{f}_k \sim \mathcal{N}\left(0, \frac{\hat{F}_k^2}{pN(d_0, k)} \boldsymbol{I}_{pN(d_0, k)}\right) \text{ if } k \geq r, \qquad (27)$$

otherwise $\boldsymbol{f}_k$ is deterministic with $\|\boldsymbol{f}_k\|_2^2 = \hat{F}_k^2$.

**Theorem 4.** *Let $\alpha \in (0, \infty)$ and $r \geq 1$ be fixed. Assume $pN(d_0, r)/m \to \alpha$ as $d_0 \to \infty$. Then the average test error is given by*

$$\mathrm{Err}(\boldsymbol{X}; \lambda, \boldsymbol{F}, \hat{\boldsymbol{h}}) = \mathrm{B}_r(\alpha; \lambda, \boldsymbol{f}, \hat{\boldsymbol{h}}) + \mathrm{V}_r(\alpha; \lambda, \boldsymbol{f}, \hat{\boldsymbol{h}}) + \Delta_{d_0}, \qquad (28)$$

*where $\Delta_{d_0} \to 0$ in probability as $d_0 \to \infty$.*

**Corollary 2.** *If, for $1 \leq r \in \mathbb{N}$, (1) $pN(d_0, r)/m \to \alpha$ for some $\alpha \in (0, \infty)$, or (2) $pd_0^{r-1+\delta} \lesssim m \lesssim pd_0^{r-\delta}$ for some $\delta \in (0, 1/2)$, then*

$$\mathrm{Err}(\boldsymbol{X}; \boldsymbol{f}, \hat{\boldsymbol{h}}, \lambda) = \mathrm{LC}(m; \lambda, \boldsymbol{f}, \hat{\boldsymbol{h}}) + \Delta_{d_0} \qquad (29)$$

*where $\Delta_{d_0} \to 0$ in probability as $d_0 \to \infty$.*

## 5.2 Deep Convolutional Kernels

The eigenstructure of general CNN kernels are much more complicated as they depend on both the frequencies (i.e. the order of the polynomials) and the topologies of the networks [42]. To rigorously describe the eigenstructure, a heavy dose of notation must be introduced, which is beyond the scope of the paper. Nevertheless, the approach developed here is readily extended to cover general CNN kernels. We briefly describe the main ideas.

Following [42], we assume the input space is still $\mathcal{X} = \mathbb{S}^p_{d_0-1}$, where $p$ is the number of patches. For simplicity, we assume the network has $L$ convolutional layers and in each layer, the filter size and the stride are all equal to $d_0$. Thus the spatial dimension of the input is reduced to 1 after $L$ convolutional layers. We then add a non-linearality and a dense layer to generate the logits. The kernel has the following form

$$K(\boldsymbol{x}, \boldsymbol{x}') = \sum_{\boldsymbol{k} \in \mathbb{N}^p} \sum_{\boldsymbol{l} \in \prod_{i \in [p]} [N(d_0, k_i)]} \sigma_{\boldsymbol{k}, \boldsymbol{l}}^2 Y_{\boldsymbol{k}, \boldsymbol{l}}(\boldsymbol{x}) Y_{\boldsymbol{k}, \boldsymbol{l}}(\boldsymbol{x}'), \quad \text{where} \quad Y_{\boldsymbol{k}, \boldsymbol{l}}(\boldsymbol{x}) = \prod_{i \in [p]} Y_{k_i, l_i}(\boldsymbol{x}_i). \quad (30)$$

Unlike dot-product kernels in which $\boldsymbol{k}$ is a scalar and the eigenvalues depend only on $|\boldsymbol{k}|$ (i.e. the frequencies), $\sigma_{\boldsymbol{k}, \boldsymbol{l}}^2$ depends on both $|\boldsymbol{k}|$ and the spatial structure of the vector $\boldsymbol{k}$ in a rather complicated manner. Nevertheless, as $d_0 \to \infty$, $\sigma_{\boldsymbol{k}, \boldsymbol{l}}^2 \sim d_0^{-j_{\boldsymbol{k}}} = d^{-j_{\boldsymbol{k}}/L}$ for some $L \leq j_{\boldsymbol{k}} \in \mathbb{N}$. We can then categorize the eigenvectors according to the decay order of $\sigma_{\boldsymbol{k}, \boldsymbol{l}}^2$. Unlike the case of dot-product kernels or the one-hidden layer CNN kernels, in which eigenvectors with same-order eigenvalues are in the same eigenspace (i.e. the eigenvalues are the same), multiple-layer CNN kernels can have *multiple* eigenspaces with the same-order eigenvalues. Although this results in extra challenges (see below), our overall approach carries over. Consider the critical scaling regime $m \sim d^r$, for $r = j/L$ for some $L \leq j \in \mathbb{N}$. Likewise, we can decompose the kernel into low-, critical- and high-frequency parts according to $j_{\boldsymbol{k}} < r$, $j_{\boldsymbol{k}} = r$ and $j_{\boldsymbol{k}} > r$, resp. Following similar assumptions on the labels and eigenvalues, the bias and the variance can be essentially reduced to computing

$$\chi_B = \frac{1}{N_r} \mathrm{Tr}\, \boldsymbol{R}_r^2 (\boldsymbol{R}_r + Y_r(\boldsymbol{X})^\top Y_r(\boldsymbol{X})/m)^{-2} \qquad (31)$$

$$\chi_V = \frac{N_{\leq r}}{m} \frac{1}{N_{\leq r}} \mathrm{Tr}(\boldsymbol{R}_{\leq r} + Y_{\leq r}(\boldsymbol{X})^\top Y_{\leq r}(\boldsymbol{X})/m)^{-2} Y_{\leq r}(\boldsymbol{X})^\top Y_{\leq r}(\boldsymbol{X})/m \qquad (32)$$

---

[5]The Gaussian assumption is unessential. We use it here for convenience.

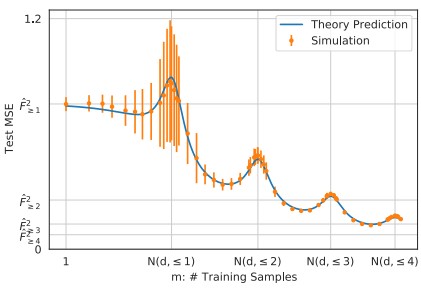
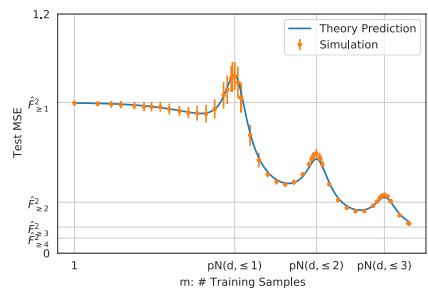

| (a) Dot Product Kernel | (b) One-layer CNN Kernel |

Figure 4: **Simulation vs Prediction.** We generate the learning curves obtain from **kernel regression** by densely varying $m$ from 1 to $24000$. For each $m$, we average the MSE over 20 runs. The **closed-form prediction** from Eq. (18) captures the simulations surprising well even for small $d$. Left: dot product kernel with $d = 24$. Right: one-hidden layer CNN kernel with $d_0 = 20$ and $p = 6$. The spectral gap is $\mathrm{Gap} = 32$ in both plots.

where $\boldsymbol{R}_r$ and $\boldsymbol{R}_{\leq r}$ are diagonal matrices whose entries are determined by the eigenvalues of the critical-frequency modes. In the dot-product kernels or one-hidden layer CNN kernels setting, $\boldsymbol{R}_r/\boldsymbol{R}_{\leq r}$ is a scaled identity matrix and simple, closed-form expressions for the above traces straightforwardly follow from the Marcenko-Pastur distribution. However, for a general diagonal matrix $\boldsymbol{R}_r$ with bounded limiting spectra, $\boldsymbol{R}_r$ does not commute with $Y_r^\top(\boldsymbol{X})Y_r(\boldsymbol{X})$, and a more detailed random matrix analysis is needed. See the supplementary material for more details.

## 6 Experiments

We provide experiments to show that our learning curves (Eq. (21)) accurately capture empirical sample-wise learning curve even when the ambient dimensions remains small. Even though our theoretical results require averaging the test error over random labels (aka, mean test error), our experimental results suggest this is unnecessary, i.e. the learning curve Eq. (21) can capture the test error accurately for any given draw of label function.

**Experimental setup.** We generate a polynomial kernel function $h(t) = \sum_{k=1}^{7} \hat{h}_k^2 P_k(t)$, where $P_k$ is the degree-$k$ Legendre polynomial in $d$ dimensions. The kernel function can be efficiently computed via $K(\boldsymbol{x}, \boldsymbol{x}') = h(\boldsymbol{x}^\top \boldsymbol{x}')$. We choose the label function to be $f(\boldsymbol{x}) = \sum_{k=1}^{7} \hat{F}_k y_k(\boldsymbol{x})$, where $y_k(\boldsymbol{x}) = \sum_{j \in [d]} w_{k,j} \prod_{i=j}^{j+k-1} x_i$, and the coefficients $w_{k,j}$ are randomly sampled from a Gaussian and then normalized so that $\mathbb{E}_{\boldsymbol{x}} |y_k(\boldsymbol{x})|^2 = 1$ for each $k$. Therefore $\mathbb{E}_{\boldsymbol{x}} |f(\boldsymbol{x})|^2 = \sum_{k=1}^{7} \hat{F}_k^2$. For simplicity, we also set $\sigma_\epsilon^2 = 0$ (i.e. noiseless) and $\lambda = 0$ (i.e. ridgeless). Note that when $m \lesssim d^r$, the regressor still contains "effective noise" $\hat{F}_{>r}^2$ from un-learnable high-frequency modes and "effective regularization" $\hat{h}_{>r}^2$. Finally, in our experiments, we choose $\hat{F}_k^2 = k^{-2}$ and $\hat{h}_k^2 = \mathrm{Gap}^{-(k-1)}$, where we will vary the value of the spectral gap: $\mathrm{Gap} = \hat{h}_k^2 / \hat{h}_{k+1}^2$. Under this setup, the predicted learning curve $\mathrm{LC}(m) = \mathrm{LC}(m; \mathrm{Gap})$ depends only on the spectral gap of the kernel.

To simulate higher-order scaling ($r \geq 3$), the dimension $d$ has to be very small as we need to invert a sequence of matrices of size ranging from $m = 1$ to $m \propto d^r/r!$. Due to the constraints in compute and memory, the largest $m$ we can have is typically $m_{\max} \approx 25,000$ for one single GPU and $d$ in our experiments is typically around $d = 24$. As such we are in a regime with strong finite-size corrections. Finally, all experiments are run in a single A-100 using Google Cloud Colab Notebooks.

**Learning Curves Accurately Capture Simulations.** In Fig. 4, we generate the empirical sample-wise learning curve by applying kernel regression Eq. (13) with training set $\boldsymbol{X}$. We vary the training set size $m$ densely in $[1, m_{\max}]$ and for each $m$ we sample 20 independent $\boldsymbol{X}$ to get the **errorbar plot** for the test error. The closed-form **learning curve** is obtained from Eq. (21) and the calculation is done in Sec.C.5. Even in the low-dimensional regime with $d = 24$ for dot-product kernel ($d_0 = 20$ and $p = 6$ for one-hidden layer CNN kernel), the predicted learning curve captures the empirical

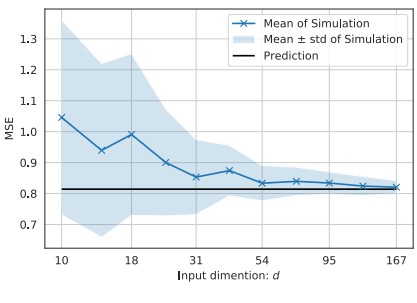
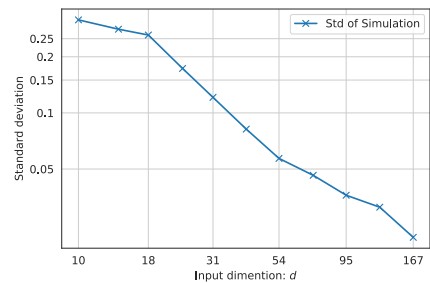

(a) Mean of Simulations.  (b) Standard Deviation of Simulations.

Figure 5: **Simulations approach predictions as** $d \to \infty$. The mean and the standard deviation are computed over 32 runs. The predictions and simulations are obtained via the second peak $r = 2$, namely, $m = N(d, \leq 2)$.

learning curve surprisingly well, which has a highly non-trivial multiple-descent behavior. It is worth mentioning that, from the simulation, the deviation of the test error from its mean is relatively large when $m$ is small but vanishes quickly as $m$ becomes larger. This suggests Theorem 2 and Corollary 1 should hold in a pointwise fashion, i.e. without averaging the test error over random labels.

**Finite-size correction vanishes as** $d \to \infty$. Our theoretical results assume that the input dimension $d$ is sufficiently large and these results are exact when $d = \infty$. To visualize the finite-size correction, we plot the dependence of the correction (between simulations and predictions) on the the input dimension $d$. Fig. 5 (a) shows that the means of simulations are converging to the theoretical prediction. Fig. 5 (b) shows that the standard deviations are converging zero.

**Small Spectral Gap Eliminates Multiple-descent.** In Fig. 1, we plot both the predicted learning curves and simulations when $\mathrm{Gap}$ ranging in $[2, 8, 32, 128]$. For $1 \leq r \leq 6$, we have $\xi_r = \mathrm{Gap}^{-(r-1)} / \sum_{k=r}^{7} \mathrm{Gap}^{-(k-1)}$ and $\xi_r \approx \mathrm{Gap}$ when $\mathrm{Gap}$ is large. Recall that the variance term $\mathrm{V}_r$ peaks at $\alpha = 1$ and the peak scales like $\xi_r^{1/2} \approx \mathrm{Gap}^{1/2}$. When $\mathrm{Gap}$ is large, e.g. $\mathrm{Gap} = 32, 128$, the variance is also large near $\alpha = 1$, the multiple-descent phenomena are more prominent. On the other hand, when $\mathrm{Gap}$ is small, e.g. $\mathrm{Gap} = 8, 2$, such phenomena disappear and learning curves become monotonic.

## 7 Conclusion

In this work, we establish precise asymptotic formulas for the sample-wise learning curves in the kernel ridge regression setting for a family of dot-product kernels in the polynomial scaling regimes $m \propto d^r$ for all $r \in \mathbb{N}^*$. We demonstrate that these formulas can capture empirical learning curves surprisingly well even in the regime where strong finite-size corrections would be expected. We rigorously prove that the learning curves can be non-monotonic near $m \propto d^r / r!$ for each $r \in \mathbb{N}^*$. There are a couple limitations of our approach which could be improved in future work. The first one is the strong assumption on the distribution of the input data, namely, the uniform distribution on the spherical type of data. In addition, the learning curves are obtained only in the kernel regression setting and extending the results to the random feature setting (see, e.g., [29]) and the feature learning setting [44] would be meaningful future directions.

## Acknowledgement

We thank Ben Adlam for providing valuable feedback on a draft. T.M. was supported by NSF through award DMS-2031883 and the Simons Foundation through Award 814639 for the Collaboration on the Theoretical Foundations of Deep Learning. T.M. also acknowledge the NSF grant CCF-2006489 and the ONR grant N00014-18-1-2729. The work of Yue M. Lu is supported by a Harvard FAS Dean's competitive fund award for promising scholarship, and by the US National Science Foundation under grant CCF-1910410.

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
