# A  Appendix Guidelines

The appendix is organized as follows. We prove Theorem 1 and Theorem 3 in Sec. B. In Sec. C, we prove the test errors for dot-product kernels, namely, Theorem 2 and for one-hidden-layer convolutional kernels, namely, Theorem 4. The proof also shows how to reduce the test error of multiple-layer convolutional kernels to evaluating Eq. (31) and Eq. (32). Finally, in Sec.D, we provide additional plots to empirically verify that the finite-size correction becomes smaller as $d$ grows larger.

# B  Proof of Theorem 1

We begin with some notations. For positive numbers $a$ and $b$, we use $a \lesssim b$ to mean there is a constant independent of $d$ such that $a \leq Cb$. In addition, $a \sim b$, if $a \lesssim b$ and $b \lesssim a$.

The proof of Theorem 3 is similar. We only present the proof of Theorem 1. Our proof is based on the following result from [5].

**Lemma 1** ([5]). *Let $\boldsymbol{x}_p \in \mathbb{R}^p$ be random vectors and $\boldsymbol{X} = [\boldsymbol{x}_{p1}, \ldots \boldsymbol{x}_{pn}]$ be a $p \times m$ matrix with iid columns. If for every $\{\boldsymbol{A}_p\}_p$, $p \times p$ matrix with uniform operator norm,*

$$\frac{1}{p^2} \mathbb{E}|\boldsymbol{x}_p^T \boldsymbol{A}_p \boldsymbol{x}_p - \operatorname{Tr}(\boldsymbol{A}_p))|^2 \to 0, \tag{33}$$

*then the empirical spectral distribution of $\frac{1}{m}\boldsymbol{X}\boldsymbol{X}^T$ converges to $\mu_\alpha$ weakly if $p/m \to \alpha \in (0, \infty)$.*

We prove a slightly more general version.

**Theorem 5.** *Let $r \in \mathbb{N}$ and $\alpha \in (0, \infty)$ be fixed. Assume $m = m(d)$ with $N(d, r)/m \to \alpha \in (0, \infty)$ as $d \to \infty$. Let $\boldsymbol{u} = \boldsymbol{u}(d)$ be a sequence of functions defined on $\mathbb{S}_{d-1}$ such that*

*(1.)  the cardinality of $\boldsymbol{u}$ satisfies $|\boldsymbol{u}|/d^r \to 0$ as $d \to \infty$;*

*(2.)  the functions in $\boldsymbol{u}$ and $Y_r$ are mutually orthogonal;*

*(3.)  for any unit vector $\theta$, let $\mathbb{E}_{\boldsymbol{x}}|\theta^T Z_r(\boldsymbol{x})|^4 \lesssim 1$ uniformly of $d$ and $\theta$, where $Z_r(\boldsymbol{x})$ is the concatenation of $\boldsymbol{u}(\boldsymbol{x})$ and $Y_r(\boldsymbol{x})$.*

*Let $Z_r(\boldsymbol{X})$ be the concatenation of $Y_r(\boldsymbol{X})$ and $\boldsymbol{u}(\boldsymbol{X})$. Then the empirical spectral distribution of $\frac{1}{m}Z_r(\boldsymbol{X})^T Z_r(\boldsymbol{X})$ converges in distribution to the Marchenko-Pastur distribution $\mu_\alpha$.*

We mainly use the case when $\boldsymbol{u}$ is the empty set, i.e. $Z_r = Y_r$ and the case when $\boldsymbol{u} = [Y_{kl}]_{k<r}^T$, i.e. $Z_r = Y_{\leq r}$.

*Proof of Theorem 5.*  We apply Lemma 1 to $Z_r^T(\boldsymbol{X})$. We only need to show for matrices $\boldsymbol{A} = \boldsymbol{A}^{(d)}$ with $\|\boldsymbol{A}\|_{op} \leq 1$,

$$(|\boldsymbol{u}| + N(d, r))^{-2} \mathbb{E}_{\boldsymbol{x}}|Z_r(\boldsymbol{x})^T \boldsymbol{A} Z_r(\boldsymbol{x}) - \operatorname{Tr}(\boldsymbol{A})|^2 \to 0 \quad \text{as} \quad d \to \infty, \tag{34}$$

or, equivalently

$$N(d, r)^{-2} \mathbb{E}_{\boldsymbol{x}}|Z_r(\boldsymbol{x})^T \boldsymbol{A} Z_r(\boldsymbol{x}) - \operatorname{Tr}(\boldsymbol{A})|^2 \to 0 \quad \text{as} \quad d \to \infty. \tag{35}$$

since $|\boldsymbol{u}| = o(d^r)$. The assumption $\|\boldsymbol{A}\|_{op} \leq 1$ implies that the absolute values of all entries of $\boldsymbol{A}$ are bounded 1. A key observation in proving the above estimate is that, up to a unitary transformation, almost all functions in $\{Y_{r,l}(\boldsymbol{x})\}_l$ are monomials of the form

$$g_{\boldsymbol{i}}(\boldsymbol{x}) = C_{d,r} \prod_{i \in \boldsymbol{i}} x_i \tag{36}$$

where $\boldsymbol{i} \subseteq [d]$ with $|\boldsymbol{i}| = r$ and $C_{d,r}$ is a normalizing factor such that

$$C_{d,r}^2 \int_{\mathbb{S}_{d-1}} \prod_{i \in \boldsymbol{i}} |x_i|^2 d\boldsymbol{x} = 1. \tag{37}$$

We prove later that for any finite integer $r \geq 1$,

$$\int_{\mathbb{S}_{d-1}} \prod_{i \in \boldsymbol{i}} |x_i|^2 d\boldsymbol{x} = \prod_{i \in \boldsymbol{i}} \int_{\mathbb{S}_{d-1}} |x_i|^2 d\boldsymbol{x} + O(d^{-r-1}) = d^{-r} + O(d^{-r-1}) \tag{38}$$

Now we proceed to prove Eq. (35). First note that $\mathbb{G} = \{g_{\boldsymbol{i}} : \boldsymbol{i} \subseteq [d], |\boldsymbol{i}| = r\}$ is an orthonormal set. This can be proved by noticing that if $\boldsymbol{i} \neq \boldsymbol{j}$, then there is $i \in \boldsymbol{i}$ but $i \notin \boldsymbol{j}$. Clearly, the symmetries of the measure on $\mathbb{S}_{d-1}$ implies

$$\int_{\mathbb{S}_{d-1}} g_{\boldsymbol{i}}(\boldsymbol{x}) g_{\boldsymbol{j}}(\boldsymbol{x}) d\boldsymbol{x} = 0 \,. \tag{39}$$

We choose $\mathbb{B} = \{b_j\}_{j \in [p]}$ so that $\mathbb{G} \sqcup \mathbb{B}$ forms an orthonormal basis of $Z_r(\boldsymbol{x})$. Note that the cardinality of $\mathbb{B}$ is $o(d^r)$. Indeed,

$$p = |\boldsymbol{u}| + N(d,r) - \binom{d}{r} = |\boldsymbol{u}| + \binom{d+r-2}{r} + \binom{d+r-3}{r-1} - \binom{d}{r} = |\boldsymbol{u}| + O(d^{r-1}) = o(d^r) \tag{40}$$

Thus $\frac{p}{N(d,r)} \to 0$ as $d \to \infty$. After a change of basis, we can assume $Z_r(\boldsymbol{x}) = [\boldsymbol{g}(\boldsymbol{x})^T, \boldsymbol{b}(\boldsymbol{x})^T]^T$, where $\boldsymbol{g} = [g_{\boldsymbol{i}}]_{\boldsymbol{i}}^T$ and $\boldsymbol{b} = [b_j]_{j \in [p]}^T$. Here we use the fact that Eq. (35) holds for all $\boldsymbol{A}$ with uniform operator norms is equivalent to that it holds for $\boldsymbol{Q}^T \boldsymbol{A} \boldsymbol{Q}$ for all such $\boldsymbol{A}$ and any unitary matrix $\boldsymbol{Q}$. We write,

$$\boldsymbol{A} = \begin{bmatrix} \boldsymbol{A}_{11} & \boldsymbol{A}_{12} \\ \boldsymbol{A}_{21} & \boldsymbol{A}_{22} \end{bmatrix} \tag{41}$$

where $\boldsymbol{A}_{11}$ is the upper left $\binom{d}{r} \times \binom{d}{r}$ block of $\boldsymbol{A}$, $\boldsymbol{A}_{22}$ is the lower right $p \times p$ block of $\boldsymbol{A}$ and the other two blocks are defined similarly. Note that $\|\boldsymbol{A}_{ij}\|_{op} \leq \|\boldsymbol{A}\|_{op} \leq 1$ for $i, j \in \{1, 2\}$. As such, we have

$$N(d,r)^{-2} \mathbb{E}_{\boldsymbol{x}} |Z_r(\boldsymbol{x})^T \boldsymbol{A} Z_r(\boldsymbol{x}) - \text{Tr}(\boldsymbol{A})|^2 \leq 5(I_1 + I_2 + I_3 + I_4 + I_5) \tag{42}$$

where

$$I_1 = N(d,r)^{-2} \mathbb{E}_{\boldsymbol{x}} |\boldsymbol{g}(\boldsymbol{x})^T \boldsymbol{A}_{11} \boldsymbol{g}(\boldsymbol{x}) - \text{Tr}(\boldsymbol{A}_{11})|^2 \tag{43}$$

$$I_2 = N(d,r)^{-2} \mathbb{E}_{\boldsymbol{x}} |\boldsymbol{b}(\boldsymbol{x})^T \boldsymbol{A}_{22} \boldsymbol{b}(\boldsymbol{x})|^2 \tag{44}$$

$$I_3 = N(d,r)^{-2} \mathbb{E}_{\boldsymbol{x}} |\boldsymbol{g}(\boldsymbol{x})^T \boldsymbol{A}_{12} \boldsymbol{b}(\boldsymbol{x})|^2 \tag{45}$$

$$I_4 = N(d,r)^{-2} \mathbb{E}_{\boldsymbol{x}} |\boldsymbol{b}(\boldsymbol{x})^T \boldsymbol{A}_{21} \boldsymbol{g}(\boldsymbol{x})|^2 \tag{46}$$

$$I_5 = N(d,r)^{-2} |\text{Tr}(\boldsymbol{A}_{22})|^2 \tag{47}$$

We prove $I_i \to 0$ for $1 \leq i \leq 5$. The $i = 1$ case is the most difficult and the others are straightforward since $pN(d,r)^{-1} \to 0$. E.g., when $i = 3$

$$I_3 \leq N(d,r)^{-2} \mathbb{E}_{\boldsymbol{x}} \|\boldsymbol{g}(\boldsymbol{x})\|_{l_2}^2 \|\boldsymbol{A}_{12}\|_{op}^2 \|\boldsymbol{b}(\boldsymbol{x})\|_{l_2}^2 \tag{48}$$

$$\leq N(d,r)^{-2} \left( \mathbb{E}_{\boldsymbol{x}} \|\boldsymbol{g}(\boldsymbol{x})\|_{l_2}^4 \mathbb{E}_{\boldsymbol{x}} \|\boldsymbol{b}(\boldsymbol{x})\|_{l_2}^4 \right)^{1/2} \tag{49}$$

$$\leq N(d,r)^{-2} \max_{\boldsymbol{i},j} (\mathbb{E}_{\boldsymbol{x}} |g_{\boldsymbol{i}}(\boldsymbol{x})|^4 \mathbb{E}_{\boldsymbol{x}} |b_j(\boldsymbol{x})|^4)^{1/2} pN(d,r) \tag{50}$$

$$= CpN(d,r)^{-1} \to 0 \tag{51}$$

where we have set $C = (\mathbb{E}_{\boldsymbol{x}} |g_{\boldsymbol{i}}(\boldsymbol{x})|^4 \mathbb{E}_{\boldsymbol{x}} |b_j(\boldsymbol{x})|^4)^{1/2}$, which is $O(1)$ due to assumption (3.) in Theorem 5. The bounds for $I_2$ and $I_4$ can be obtained similarly. For $I_5$, we simply use $\text{Tr}(\boldsymbol{A}_{22}) \leq p\|\boldsymbol{A}\|_{op} \leq p$.

It remains to control $I_1$. To ease the notation, denote $\boldsymbol{B} = \boldsymbol{A}_{11}$. We split $I_1 \leq I_{11} + I_{12}$, where $I_{11}$ and $I_{12}$ are the diagonal and the off-diagonal parts, resp.,

$$I_{11} = 2N(d,r)^{-2} \mathbb{E}_{\boldsymbol{x}} |\sum_{\boldsymbol{i}} B_{\boldsymbol{i}\boldsymbol{i}} (g_{\boldsymbol{i}}^2(\boldsymbol{x}) - 1)|^2 \tag{52}$$

$$I_{12} = 2N(d,r)^{-2} \mathbb{E}_{\boldsymbol{x}} |\sum_{\boldsymbol{i} \neq \boldsymbol{j}} B_{\boldsymbol{i}\boldsymbol{j}} g_{\boldsymbol{i}} g_{\boldsymbol{j}}(\boldsymbol{x})|^2 \tag{53}$$

**Bounding the diagonal part $I_{11}$.** Using $\mathbb{E}_{\boldsymbol{x}} g_i(\boldsymbol{x})^2 = 1$, we have

$$I_{11} = 2N(d,r)^{-2} \mathbb{E}_{\boldsymbol{x}} \sum_{i,j} B_{ii} B_{jj} (g_i^2(\boldsymbol{x}) g_j^2(\boldsymbol{x}) - 1) \tag{54}$$

We spit the proof into two cases: $\boldsymbol{i} \cap \boldsymbol{j} = \emptyset$ and $\boldsymbol{i} \cap \boldsymbol{j} \neq \emptyset$. The following is the key estimate to handle the first case.

**Lemma 2.** *If $\boldsymbol{i} \cap \boldsymbol{j} = \emptyset$,*

$$\max_{\boldsymbol{i},\boldsymbol{j}} |\mathbb{E}_{\boldsymbol{x}} g_i^2(\boldsymbol{x}) g_j^2(\boldsymbol{x}) - 1| \lesssim d^{-1}. \tag{55}$$

We prove this lemma later. We show how to use this lemma to handle the $\boldsymbol{i} \cap \boldsymbol{j} = \emptyset$ case. Recall that $|B_{\boldsymbol{i},\boldsymbol{j}}| \leq 1$ and the number of tuples $(\boldsymbol{i}, \boldsymbol{j})$ is fewer than $N(d,r)^2$. We have

$$N(d,r)^{-2} |\mathbb{E}_{\boldsymbol{x}} \sum_{\boldsymbol{i},\boldsymbol{j}, \boldsymbol{i} \cap \boldsymbol{j} = \emptyset} B_{ii} B_{jj} (g_i^2(\boldsymbol{x}) g_j^2(\boldsymbol{x}) - 1)| \tag{56}$$

$$\leq N(d,r)^{-2} \sum_{\boldsymbol{i},\boldsymbol{j}, \boldsymbol{i} \cap \boldsymbol{j} = \emptyset} \max_{\boldsymbol{i},\boldsymbol{j}} |\mathbb{E}_{\boldsymbol{x}} g_i^2(\boldsymbol{x}) g_j^2(\boldsymbol{x}) - 1| \tag{57}$$

$$\leq \max_{\boldsymbol{i},\boldsymbol{j}} |\mathbb{E}_{\boldsymbol{x}} g_i^2(\boldsymbol{x}) g_j^2(\boldsymbol{x}) - 1| \lesssim d^{-1}. \tag{58}$$

We turn to $|\boldsymbol{i} \cap \boldsymbol{j}| = t, 1 \leq t \leq r$. For each fixed $\boldsymbol{i}$, the number of choices of $\boldsymbol{j}$ is

$$\sum_{1 \leq t \leq r} \binom{r}{t} \binom{d-r}{r-t} \lesssim \sum_{1 \leq t \leq r} d^{r-t} \sim d^{r-1} \tag{59}$$

As such,

$$2N(d,r)^{-2} \mathbb{E}_{\boldsymbol{x}} \sum_{\boldsymbol{i},\boldsymbol{j}, \boldsymbol{i} \cap \boldsymbol{j} \neq \emptyset} |B_{ii} B_{jj} (g_i^2(\boldsymbol{x}) g_j^2(\boldsymbol{x}) - 1)| \tag{60}$$

$$\lesssim N(d,r)^{-2} N(d,r) d^{r-1} \max_{\boldsymbol{i},\boldsymbol{j}} \mathbb{E}_{\boldsymbol{x}} |g_i^2(\boldsymbol{x}) g_j^2(\boldsymbol{x}) - 1| \tag{61}$$

$$\lesssim d^{-1} \tag{62}$$

**Off-diagonal terms $I_{12}$.** Bounding the off-diagonal terms can be reduced to a combinatorics problem, which is similar to the random tensor model considered in [9]. We need to estimate

$$I_{12} = 2N(d,r)^{-2} \sum_{i \neq j, l \neq k} B_{ij} B_{lk} \mathbb{E}_{\boldsymbol{x}} g_i(\boldsymbol{x}) g_j(\boldsymbol{x}) g_l(\boldsymbol{x}) g_k(\boldsymbol{x}) \tag{63}$$

By symmetries of the uniform measure on the sphere, we can assume the monomial $g_i(\boldsymbol{x}) g_j(\boldsymbol{x}) g_l(\boldsymbol{x}) g_k(\boldsymbol{x})$ has no linear factor, that is the degree of any $x_i$ in this monomial must be at least 2 if not 0. In addition, for such monomials, the Holder inequality and hypercontractivities yield,

$$|\mathbb{E}_{\boldsymbol{x}} g_i(\boldsymbol{x}) g_j(\boldsymbol{x}) g_l(\boldsymbol{x}) g_k(\boldsymbol{x})| \leq \max_{\boldsymbol{i}} \mathbb{E}_{\boldsymbol{x}} |g_i(\boldsymbol{x})|^4 \lesssim \max_{\boldsymbol{i}} (\mathbb{E}_{\boldsymbol{x}} |g_i(\boldsymbol{x})|^2)^2 = 1. \tag{64}$$

As such, we only need to show that the growth of the number of such quadruples $(\boldsymbol{i}, \boldsymbol{j}, \boldsymbol{k}, \boldsymbol{l})$, as a function of $d$, is slower than $N(d,r)^2 \sim d^{2r}$. We proceed to prove this claim. For each fixed $\boldsymbol{i}$, let $t = |\boldsymbol{i} \cap \boldsymbol{j}|$ where $0 \leq t \leq r-1$ ($t \neq r$ since $\boldsymbol{i} \neq \boldsymbol{j}$). Let $J(\boldsymbol{i}; t)$ denote the set of such $\boldsymbol{j}$, whose cardinality is

$$|J(\boldsymbol{i}; t)| = \binom{r}{t} \binom{d-r}{r-t} < r^t d^{r-t} \lesssim d^{r-t}. \tag{65}$$

Next we estimate the number of tuples $(\boldsymbol{l}, \boldsymbol{k})$. Let $w = |(\boldsymbol{l} \cup \boldsymbol{k}) \setminus (\boldsymbol{i} \cup \boldsymbol{j})|$. Since $|\boldsymbol{l} \cup \boldsymbol{k} \cup \boldsymbol{i} \cup \boldsymbol{j}| \leq 2r$ and $|\boldsymbol{i} \cup \boldsymbol{j}| = 2r - t$, we have $w \leq t$. The cardinality of choosing such $\boldsymbol{k} \cup \boldsymbol{l}$ cannot exceed

$$\sum_{0 \leq w \leq t} \binom{d - (2r - t)}{w} \sum_{v=0}^{2r-w} \binom{2r}{v} \lesssim d^t. \tag{66}$$

With $\boldsymbol{k} \cup \boldsymbol{l}$ given, the pair of $(\boldsymbol{k}, \boldsymbol{l})$ cannot exceed $\binom{2r}{r}^2 \lesssim 1$. Thus, with $\boldsymbol{i}, \boldsymbol{j}$ and $t$ fixed, the number of pairs of $(\boldsymbol{k}, \boldsymbol{l})$ is $\lesssim d^t$. Using $|B_{\boldsymbol{l}\boldsymbol{k}}| \leq 1$ and $\max_i (\sum_j B_{ij}^2)^{1/2} \leq \|\boldsymbol{B}\|_{op} \leq 1$, we have

$$N(d, r)^2 I_{12} \lesssim \sum_{\boldsymbol{i}} \sum_{0 \leq t \leq r-1} \sum_{\boldsymbol{j} \in J(\boldsymbol{i};t)} B_{ij} d^t \tag{67}$$

$$\leq \sum_{\boldsymbol{i}} \sum_{0 \leq t \leq r-1} |J(\boldsymbol{i};t)|^{1/2} (\sum_{\boldsymbol{j} \in J(\boldsymbol{i};t)} B_{ij}^2)^{1/2} d^t \tag{68}$$

$$\lesssim \|\boldsymbol{A}\|_{op} \sum_{\boldsymbol{i}} \sum_{0 \leq t \leq r-1} d^{(r-t)/2} d^t \tag{69}$$

$$\lesssim N(d, r) d^{2r-1/2} \tag{70}$$

which gives $I_{12} \lesssim d^{-1/2}$. $\qquad\square$

*Proof of Lemma 2.* It suffices to prove that for any finite integer $j > 1$,

$$\int_{\mathbb{S}_{d-1}} \prod_{1 \leq t \leq j} x_t^2 d\boldsymbol{x} = d^{-j} + O(d^{-j-1}). \tag{71}$$

Indeed, assuming this estimate, we have $C_{d,j}^{-2} = d^{-j} + O(d^{-j-1})$ and $C_{d,j}^2 = d^j + O(d^{j-1})$. For any $\boldsymbol{i}$ and $\boldsymbol{j}$ with $\boldsymbol{i} \cap \boldsymbol{j} = \emptyset$,

$$\mathbb{E}_{\boldsymbol{x}} g_{\boldsymbol{i}}^2(\boldsymbol{x}) g_{\boldsymbol{j}}^2(\boldsymbol{x}) - 1 = C_{d,r}^4 \int_{\mathbb{S}_{d-1}} \prod_{t \in \boldsymbol{i} \cup \boldsymbol{j}} x_t^2 d\boldsymbol{x} = C_{d,r}^4 C_{d,2r}^{-2} - 1 = O(d^{-1}) \tag{72}$$

It remains to prove Eq. (71). By symmetries,

$$\int_{\mathbb{S}_{d-1}} x_t^2 d\boldsymbol{x} = \frac{1}{d} \int_{\mathbb{S}_{d-1}} \sum_{1 \leq t \leq d} x_t^2 d\boldsymbol{x} = \frac{1}{d} \int_{\mathbb{S}_{d-1}} 1 d\boldsymbol{x} = \frac{1}{d}. \tag{73}$$

By symmetries again,

$$\int_{\mathbb{S}_{d-1}} \prod_{1 \leq t \leq j-1} x_t^2 d\boldsymbol{x} \tag{74}$$

$$= \int_{\mathbb{S}_{d-1}} \prod_{1 \leq t \leq j-1} x_t^2 \left( \sum_{1 \leq i \leq j-1} x_i^2 + \sum_{j \leq i \leq d} x_i^2 \right) d\boldsymbol{x} \tag{75}$$

$$= (d - j + 1) \int_{\mathbb{S}_{d-1}} \prod_{1 \leq t \leq j} x_t^2 d\boldsymbol{x} + (j - 1) \int_{\mathbb{S}_{d-1}} x_1^2 \prod_{1 \leq t \leq j-1} x_t^2 d\boldsymbol{x} \tag{76}$$

We use hypercontractivities to bound the error term, namely, the second term. Recall that for any $q \geq 2$ and any polynomial defined on the sphere,

$$\left( \int_{\mathbb{S}_{d-1}} |f(\boldsymbol{x})|^q d\boldsymbol{x} \right)^{1/q} \leq (q-1)^{\deg(f)/2} \left( \int_{\mathbb{S}_{d-1}} |f(\boldsymbol{x})|^2 d\boldsymbol{x} \right)^{1/2}. \tag{77}$$

Setting $f(\boldsymbol{x}) = x_t$ (with $\deg(f) = 1$) gives

$$\int_{\mathbb{S}_{d-1}} |x_t|^q d\boldsymbol{x} \leq (q-1)^{q/2} \left( \int_{\mathbb{S}_{d-1}} |x_t|^2 d\boldsymbol{x} \right)^{q/2} = (q-1)^{q/2} d^{-\frac{q}{2}}. \tag{78}$$

By Holder's inequality and symmtries

$$\int_{\mathbb{S}_{d-1}} x_1^2 \prod_{1 \leq t \leq j-1} x_t^2 d\boldsymbol{x} \leq \left( \int_{\mathbb{S}_{d-1}} |x_1|^{2j} d\boldsymbol{x} \prod_{1 \leq t \leq j-1} \int_{\mathbb{S}_{d-1}} |x_t|^{2j} d\boldsymbol{x} \right)^{\frac{1}{j}} \tag{79}$$

$$= \int_{\mathbb{S}_{d-1}} |x_1|^{2j} d\boldsymbol{x} \leq (2j-1)^j d^{-j} \tag{80}$$

Thus

$$\int_{\mathbb{S}_{d-1}} \prod_{1 \leq t \leq j-1} x_t^2 d\boldsymbol{x} = (d - j + 1) \int_{\mathbb{S}_{d-1}} \prod_{1 \leq t \leq j} x_t^2 d\boldsymbol{x} + O(d^{-j}) \tag{81}$$

and

$$\int_{\mathbb{S}_{d-1}} \prod_{1 \leq t \leq j} x_t^2 d\boldsymbol{x} = \frac{1}{d - (j - 1)} \int_{\mathbb{S}_{d-1}} \prod_{1 \leq t \leq j-1} x_t^2 d\boldsymbol{x} + O(d^{-j-1}). \tag{82}$$

Finally, Eq. (71) is a consequence of this estimate and induction.

$\square$

## C Generalization

We aim to obtain the asymptotic formulas for the test error in this section. In the high-level, we decompose the empirical kernel $K(\boldsymbol{X}, \boldsymbol{X})$ into low-, critical- and high-frequency modes, where we have concentration in the low- and high-frequency parts of the kernel. The test error associated to these two parts are easier to handle. The critical-frequency part is more difficult in which random matrix behaviors emerge, namely, the Marchenko-Pastur distribution. As such, our first step is to remove the contribution in the test error coming from the non-critical frequency parts. After that, the remaining is essentially equivalent to computing the trace of certain functional forms related to the Marchenko-Pastur distribution.

We consider a general setting that includes the dot-product kernels, the one-hidden-layer and the multiple-layer convolutional kernels (NNGP and NT kernels.) In what follows, we use $\Delta_d, \Delta_d', \Delta_d''$, etc. to represent quantities that converge to 0 in probability (the absolute value of a scalar, the norm of a vector, the operator norm of a matrix, etc.), whose exact form may change from line to line.

### C.1 Setup

For $d \in \mathbb{N}^*$, let $\boldsymbol{\mathcal{X}}^{(d)} \subseteq \mathbb{R}^d$ be the input space associated with a probability measure $\sigma^{(d)}$ and a kernel function $K^{(d)}$. Assume the kernel function has the following eigen-structure

$$K^{(d)}(\boldsymbol{x}, \boldsymbol{x}') = \sum_{k \geq 1} \sum_{n \in [E_k]} (\sigma_{kn}^{(d)})^2 \sum_{l \in N_{kn}^{(d)}} \phi_{knl}^{(d)}(\boldsymbol{x}) \phi_{knl}^{(d)}(\boldsymbol{x}') \tag{83}$$

in the sense $K^{(d)}$, as the integral operator from $L^2(\boldsymbol{\mathcal{X}}^{(d)}, \sigma^{(d)})$ to itself,

$$K^{(d)} \phi_{knl}^{(d)}(\boldsymbol{x}) = \int K^{(d)}(\boldsymbol{x}, \boldsymbol{x}') \phi_{knl}^{(d)}(\boldsymbol{x}') \sigma^{(d)}(d\boldsymbol{x}') = (\sigma_{kn}^{(d)})^2 \phi_{knl}^{(d)}(\boldsymbol{x}). \tag{84}$$

Here $\{\phi_{knl}^{(d)}\}_{knl}$ is an orthonormal basis of $L^2(\boldsymbol{\mathcal{X}}^{(d)}, \sigma^{(d)})$. We also assume $K^{(d)}$ is a trace-class operator, i.e.,

$$\sum_{knl} \langle K^{(d)} \phi_{knl}^{(d)}, \phi_{knl}^{(d)} \rangle = \sum_{k \geq 1} \sum_{n \in [E_k]} N_{kn}^{(d)} (\sigma_{kn}^{(d)})^2 < \infty. \tag{85}$$

In the above notations, we use the triplet $(k, n, l)$ to index the eigenfunctions $\phi_{knl}^{(d)}$. The tuple $(k, n)$ determines the eigenspace, whose eigenvalue is of the form "$(\sigma_{kn}^{(d)})^2 = C_n d^{-s_k} + \text{Lower Order}$" and $l$ lists all eigenfunctions in the $kn$-eigenspace. We make the following assumptions.

**Kernel Assumptions.**

(1.) **Spectral Gap.** There are $\delta_0 > 0$ and a sequence of strictly increasing positive real numbers $\{s_k\}$ with $|s_k - s_{k-1}| \geq \delta_0$ for all $k \geq 2$ such that

$$(\sigma_{kn}^{(d)})^2 \sim d^{-s_k} \sim (N_{kn}^{(d)})^{-1} \tag{86}$$

Moreover, $\{E_k\} \subseteq \mathbb{N}^*$ is independent from $d$ which grows at most exponentially. We also assume that there is a sequence of real numbers $\{\hat{h}_{kn}^2\}_{kn}$ with $\hat{h}_{kn}^2 \neq 0$ unless $k$ is sufficiently large and

$$\sum_k \sum_{n \in [E_k]} \hat{h}_{kn}^2 < \infty \quad \text{and} \quad (\sigma_{kn}^{(d)})^2 N_{kn}^{(d)} = \hat{h}_{kn}^2 \quad \text{as} \quad d \to \infty \tag{87}$$

(2.) **Hypercontractivity Inequalities.** For any $p \geq 2$ there are constant $C_{p,k}$ such that for any function $f$ in the closure of $\mathrm{Span}\{\phi_{jnl}^{(d)}\}_{j \leq k}$

$$\|f\|_p \leq C_{p,k}\|f\|_2 \tag{88}$$

(3.) **Concentration of Quadratic Forms.** Let $\phi_k^{(d)}(\boldsymbol{x})$ denote the column vector consists of elements $\{\phi_{knl}^{(d)}(\boldsymbol{x})\}_{l \in [N_{kn}(d) n \in [E_k]}$. For every sequence of matrices $\{\boldsymbol{A}^{(d)}\}$ with uniformly bounded operator norm,

$$\left(\sum_{n \in [E_k]} N_{kn}^{(d)}\right)^{-2} \mathbb{E}_{\boldsymbol{x}}|\phi_k^{(d)}(\boldsymbol{x})^\top \boldsymbol{A}^{(d)} \phi_k^{(d)}(\boldsymbol{x}) - \mathrm{Tr}\, \boldsymbol{A}^{(d)}|^2 \to 0 \quad \text{as} \quad d \to \infty. \tag{89}$$

(4.) **Addition Theorem.** For $k \in \mathbb{N}^*$ and $n \in [E_k]$ and $\boldsymbol{x} \in \mathcal{X}^{(d)}$

$$\sum_{l \in [N_{kn}^{(d)}]} \phi_{knl}^{(d)}(\boldsymbol{x})^2 = N_{kn}^{(d)} \tag{90}$$

Let us briefly explain the assumptions. The **Spectral Gap** assumption basically says, we can classify the eigenvectors into countably many categories indexed by $k \in \mathbb{N}^*$. In the $k$-th category, it has exactly $E_k$ many eigenspaces, each of them has dimensions $\sim d^{s_k}$ and eigenvalues $d^{-s_k}$. It also implies the number of eigenfunctions with eigenvalues $\lesssim d^{-s_k}$ is $\sim d^{s_k}$. Assumptions (1.), (2.) and (4.) together are stronger than those in Theorem 6 in Mei et al. [32] (and slightly less technical), which allow us to apply kernel concentration from Mei et al. [32]. In particular, they imply concentration of the low- and high-frequency parts of the empirical kernel $K^{(d)}(\boldsymbol{X}, \boldsymbol{X})$. Finally, Assumption (3) is designed to meet the requirements in Lemma 1, which allows us to claim Marchenko-Pastur type behavior of the gram matrix induced by the feature map $\phi_k$. We provide a couple examples.

**Example 1** (Dot-product Kernels)**.** *When $\mathcal{X}^{(d)} = \mathbb{S}_{d-1}$ and $K^{(d)}$ is the dot-product kernel, we have $E_k = 1$, $s_k = k$, $N_{kn} = N(d, k) \sim d^k/k!$, and $\phi_{knl} = Y_{kl}$ (note that $n = 0$ since $E_k = 1$.) Note that by the Addition Theorem of spherical harmonics (Theorem 4.11 in Frye and Efthimiou [15]),*

$$\sum_{l \in [N(d,k)]} Y_{kl}(\boldsymbol{x})^2 = N(d, k) \tag{91}$$

**Example 2** (One-hidden-layer Convolutional Kernels)**.** *Sightly more general setting is the one-layer convolutional kernel (NNGP or NT kernels). In this case, $\mathcal{X}^{(d)} = \mathbb{S}_{d_0-1}^p$ where $p$ is the number of patches and the input dimension is $d = pd_0$. We can set either $p = O(1)$ (i.e. independent of $d_0 \to \infty$) or $p \sim d^{\alpha_p}$ for some $\alpha_p > 0$. This kernel is essentially the sum of $p$ dot-product kernels. As such, $E_k = 1$, $N_{kn}(d) = pN(d_0, k) \sim pd_0^k/k!$ and $(\sigma_{kn}^u pd)^2 \sim (pd_0^k)^{-1}$. If $p \sim d^{\alpha_p}$ and $d_0 \sim d^{\alpha_{d_0}}$ with $\alpha_{d_0} + \alpha_p = 1$, we have $s_k = \alpha_p + k\alpha_{d_0}$ and $d^{-s_k}$ is the decay rate of the $k$-th order spherical harmonics.*

**Example 3** (Multiple-layer Convolutional Kernels)**.** *General convolutional kernels are much more complicated [42]. In this case, $\mathcal{X}^{(d)} = \mathbb{S}_{d_0-1}^p$ where $p$ is the number of patches and the input dimension is $d = pd_0$. We additionally assume, $p = k_0^{L-1}$ for some $k_0 \in \mathbb{N}^*$ and the network has $L$ convolutional layers with filter size and strides being the same in each layer (equal to $d_0$ in the first layer and to $k_0$ for the remaining $(L-1)$ layers.) The eigenstructures of such kernels are studied in Xiao [42]. The eigenfunctions are tensor products of spherical harmonics defined on copies of $\mathbb{S}_{d_0-1}$,*

$$Y_{\boldsymbol{k},\boldsymbol{l}}(\boldsymbol{x}) = \prod_{i \in [p]} Y_{k_i l_i}(\boldsymbol{x}_i) \tag{92}$$

*The eigenvalues are more complicated to compute as they depend on both the frequencies of $Y_{k,l}$ and the topologies of the networks. When $d_0 \propto d^{\alpha_{d_0}}$ and $k_0 \propto d^{\alpha_{k_0}}$ with $\alpha_{d_0} + (L-1)\alpha_{k_0} = 1$ and $\alpha_{k_0}, \alpha_{d_0} > 0$, the eigenvalue of $Y_{kl}$ is $\propto d^{-(\mathscr{F}(k)+\mathscr{S}(k))}$, as $d \to \infty$. Here $\mathscr{F}(k) \equiv |k|\alpha_{d_0}$ is the frequency index of $Y_{kl}$ and $\mathscr{S}(k) = J_k \alpha_{k_0}$ is the spatial index, where $J_k$ is the number of edges in the sub-tree connecting all interacting patches (i.e. $k_i \neq 0$) to the output; see Xiao [42] for more details.*

*As there can possibly exist $k$ and $k'$ with $\mathscr{F}(k)+\mathscr{S}(k) = \mathscr{F}(k')+\mathscr{S}(k')$ (i.e., same order of decay) but $(\mathscr{F}(k),\mathscr{S}(k)) \neq (\mathscr{F}(k'),\mathscr{S}(k'))$ (i.e. different space-frequency combination), there can exist more than one eigenspaces whose eigenvalues decay to zero with the same rate $d^{-(\mathscr{F}(k)+\mathscr{S}(k))}$, but with different leading coefficients. This is the main reason why we need to allow $|E_k| > 1$ in Eq. (83).*

Next we discuss the assumptions on the label function. Let $X$ be the training set with $m \sim d^{s_r}$ many training samples for some $r \in \mathbb{N}^*$ fixed. Then let the ground true label function to be

$$f(x) = \sum_{k \in \mathbb{N}^*} \sum_{n \in [E_k]} \sum_{l \in [N_{kn}^{(d)}]} \hat{f}_{knl} \phi_{knl}^{(d)}(x). \tag{93}$$

Let $N_k^{(d)} = \sum_{n \in [E_k]} N_{kn}^{(d)}$. We assume, for $k \geq r$, $\hat{f}_{kn} = \{\hat{f}_{knl}\}_{l \in [N_{kn}^{(d)}]}$ is a random vector with

$$\mathbb{E}\hat{f}_{kn} = 0 \quad \text{and} \quad \mathbb{E}\hat{f}_{kn}\hat{f}_{kn}^\top = \frac{\hat{F}_{kn}^2}{N_{kn}^{(d)}} I_{N_{kn}^{(d)}}. \tag{94}$$

and $\{\hat{f}_{kn}\}_{n \in [E_k], k \geq r}$ are mutually independent. One concrete example is

$$\hat{f}_{kn} \sim \mathcal{N}\left(0, \frac{\hat{F}_{kn}^2}{N_{kn}^{(d)}} I_{N_{kn}^{(d)}}\right). \tag{95}$$

For $k < r$, we assume the coefficients are deterministic with $\sum_l \hat{f}_{knl}^2 = \hat{F}_{kn}^2$, $\sum_n \hat{F}_{kn}^2 = \hat{F}_k^2$ and

$$\sum_{k \in \mathbb{N}^*} \hat{F}_k^2 < \infty \tag{96}$$

Our goal is to compute the average test error over the random labels defined above in the scaling limit $m \sim d^{s_r}$.

## C.2 Structure of the Empirical Kernels

For convenience, denote

$$\phi_{\leq k}^{(d)}(x) = [\phi_{jnl}^{(d)}(x)]_{l \in [N_{jn}^{(d)}], 1 \leq j \leq k, n \in [E_j]}^\top, \tag{97}$$

$$N_k^{(d)} = \sum_{n \in [E_k]} N_{kn}^{(d)} \tag{98}$$

$$N_{\leq k}^{(d)} = \sum_{1 \leq j \leq k} N_{\leq j}^{(d)} \tag{99}$$

Let $x_i^\top$ be the $i$-th row of the training matrix $X$. Similarly,

$$Z_k(X) = [\phi_k^{(d)}(x_0), \ldots, \phi_k^{(d)}(x_{m-1})]^\top \qquad Z_{\leq k}(X) = [\phi_{\leq k}^{(d)}(x_0), \ldots, \phi_{\leq k}^{(d)}(x_{m-1})]^\top \quad (100)$$

$$\Lambda_k = \text{diag}\left([(\sigma_{kn}^{(d)})^2 I_{N_{kn}^{(d)}}]_{n \in [E_k]}\right) \qquad \Lambda_{\leq k} = \text{diag}\left([(\sigma_{jn}^{(d)})^2 I_{N_{jn}^{(d)}}]_{n \in [E_j], 1 \leq j \leq k}\right) \quad (101)$$

Note that $Z_k(X)$ ($Z_{\leq k}(X)$) is an $m \times N_k^{(d)}$ ($m \times N_{\leq k}^{(d)}$) matrix and $\Lambda_k$ ($\Lambda_{\leq k}$) is an $N_k^{(d)} \times N_k^{(d)}$ ($N_{\leq k}^{(d)} \times N_{\leq k}^{(d)}$) diagonal matrix. The followings are defined similarly,

$$Z_{<k}(X), \quad Z_{kn}(X), \quad \Lambda_{<k}, \quad \Lambda_{kn}, \quad N_{<k}^{(d)}, \quad N_{kn}^{(d)}. \tag{102}$$

Next, we decompose the train-train kernel into two parts: the $\leq r$ frequency part and the $> r$ frequency parts,

$$K^{(d)}(\boldsymbol{X}, \boldsymbol{X}) = \sum_{k \in \mathbb{N}^*} Z_k(\boldsymbol{X}) \boldsymbol{\Lambda}_k Z_k(\boldsymbol{X})^\top = Z_{\leq r}(\boldsymbol{X}) \boldsymbol{\Lambda}_{\leq} Z_{\leq r}(\boldsymbol{X})^\top + \sum_{k \geq r+1} Z_k(\boldsymbol{X}) \boldsymbol{\Lambda}_k Z_k(\boldsymbol{X})^\top \tag{103}$$

$$= Z_{\leq r}(\boldsymbol{X}) \boldsymbol{\Lambda}_{\leq} Z_{\leq r}(\boldsymbol{X})^\top + \sum_{k \geq r+1} \sum_{n \in [E_k]} (\sigma_{kn}^{(d)})^2 Z_{kn}(\boldsymbol{X}) Z_{kn}(\boldsymbol{X})^\top \tag{104}$$

$$\equiv K_{\leq r}^{(d)}(\boldsymbol{X}, \boldsymbol{X}) + K_{>r}^{(d)}(\boldsymbol{X}, \boldsymbol{X}) \tag{105}$$

**Assumptions (1.) (2.)** allow us to apply kernel concentration [17, 32], which implies that the low-frequency and high-frequency parts of the empirical kernels are concentrated. By saying concentration in the high-frequency part, we mean

**Claim 1.** *Let*

$$\Delta_{kn}^{(d)} \equiv \frac{1}{N_{kn}^{(d)}} Z_{kn}(\boldsymbol{X}) Z_{kn}(\boldsymbol{X})^\top - \boldsymbol{I}_m. \tag{106}$$

*Then*

$$\mathbb{E} \sum_{k > r} \sum_{n \in [E_k]} \|\Delta_{kn}^{(d)}\|_{\mathrm{op}} \to 0 \tag{107}$$

The proof of this claim essentially follows from the arguments and results in Theorem 6 of [32]; see Sec. C.7. Thus

$$K_{>r}^{(d)}(\boldsymbol{X}, \boldsymbol{X}) = \sum_{k \geq r+1} \sum_{n \in [E_k]} (\sigma_{kn}^{(d)})^2 N_{kn}^{(d)} (\boldsymbol{I}_m + \Delta_{kn}^{(d)}) \tag{108}$$

$$= \sum_{k \geq r+1} \hat{h}_k^2 \boldsymbol{I}_m + \sum_{k \geq r+1} \sum_{n \in [E_k]} \hat{h}_{kn}^2 \Delta_{kn}^{(d)} \tag{109}$$

Denote

$$\hat{h}_{>r}^2 = \sum_{k \geq r+1} \sum_{n \in [E_k]} N_{kn}^{(d)} (\sigma_{kn}^{(d)})^2 \sim 1 \tag{110}$$

$$\Delta_{>r}^{(d)} = \sum_{k \geq r+1} \sum_{n \in [E_k]} \hat{h}_{kn}^2 \Delta_{kn}^{(d)}, \tag{111}$$

we can write

$$K_{>r}^{(d)}(\boldsymbol{X}, \boldsymbol{X}) = \hat{h}_{>r}^2 \boldsymbol{I}_m + \Delta_{>r}^{(d)}, \quad \text{where} \quad \mathbb{E}\|\Delta_{>r}^{(d)}\|_{\mathrm{op}} \to 0 \tag{112}$$

By saying the low-frequency part of the kernel concentrates, we mean (Theorem 6 [32])

$$\frac{1}{m} Z_<(\boldsymbol{X})^\top Z_<(\boldsymbol{X}) = \boldsymbol{I}_{N_<^{(d)}} + \Delta_{<r}^{(d)}, \quad \text{where} \quad \mathbb{E}\|\Delta_{<r}^{(d)}\|_{\mathrm{op}} \to 0 \tag{113}$$

Finally, Lemma 1 and **Assumption (3.)** imply that if $N_r^{(d)}/m \to \alpha \in (0, \infty)$, then the empirical measure of the critical part of the kernel matrix $\frac{1}{m} Z_r(\boldsymbol{X})^\top Z_r(\boldsymbol{X})$, and the low-and-critical frequency parts $\frac{1}{m} Z_{\leq r}(\boldsymbol{X})^\top Z_{\leq r}(\boldsymbol{X})$ converge to the Marchenko-Pastur distribution $\mu_\alpha$ weakly by Lemma 1. In particular, $\|\frac{1}{m} Z_r(\boldsymbol{X})^\top Z_r(\boldsymbol{X})\|_{\mathrm{op}} + \|\frac{1}{m} Z_{\leq r}(\boldsymbol{X})^\top Z_{\leq r}(\boldsymbol{X})\|_{\mathrm{op}} = O(1)$ in probability as $d \to \infty$.

For convenience, we summarize the structure of the empirical kernel in the following.

**Corollary 3.** *Assume* **Assumptions (1.-4.)**. *Let* $r \in \mathbb{N}^*$ *and* $\alpha > 0$ *be fixed and* $m = m^{(d)}$ *be such that* $N_r^{(d)}/m \to \alpha \in (0, \infty)$ *as* $d \to \infty$. *Let* $\boldsymbol{X}$, *of shape* $m \times d$, *be the training set matrix whose rows are drawn, uniformly, iid from* $\mathcal{X}^{(d)}$. *Then we have the following structure for the empirical*

*kernel matrix*

**High-frequency Features:** $\qquad\qquad\qquad K^{(d)}_{>r}(\boldsymbol{X}, \boldsymbol{X}) = \hat{h}^2_{>r}\boldsymbol{I}_m + \Delta^{(d)}_{>r},$ $\qquad\qquad$ (114)

**Low-frequency Features:** $\qquad\qquad\qquad \dfrac{1}{m}Z_<(\boldsymbol{X})^\top Z_<(\boldsymbol{X}) = \boldsymbol{I}_{N^{(d)}_<} + \Delta^{(d)}_{<r}$ $\qquad\qquad$ (115)

**Critical-frequency Features:** $\qquad$ *the empirical measure of* $\dfrac{1}{m}Z_r(\boldsymbol{X})^\top Z_r(\boldsymbol{X}) \to \mu_\alpha$

$\qquad\qquad\qquad\qquad\qquad\qquad\qquad\qquad\qquad\qquad\qquad\qquad\qquad\qquad\qquad$ (116)

**Low-and-critical-frequency Features:** $\quad$ *the empirical measure of* $\dfrac{1}{m}Z_{\leq r}(\boldsymbol{X})^\top Z_{\leq r}(\boldsymbol{X}) \to \mu_\alpha$

$\qquad\qquad\qquad\qquad\qquad\qquad\qquad\qquad\qquad\qquad\qquad\qquad\qquad\qquad\qquad$ (117)

Let $0 \leq \lambda = O(1)$ be the regularization and $\gamma = \lambda + \hat{h}^2_{>r}$ be the effective regularization. To ease the notations, denote

$$\overline{\boldsymbol{Z}}_< = \frac{1}{\sqrt{m}}Z_{<r}(\boldsymbol{X}) \qquad \overline{\boldsymbol{Z}}_\leq = \frac{1}{\sqrt{m}}Z_{\leq r}(\boldsymbol{X}) \qquad \overline{\boldsymbol{Z}}_> = \frac{1}{\sqrt{m}}Z_{>r}(\boldsymbol{X}) \qquad \overline{\boldsymbol{Z}}_r = \frac{1}{\sqrt{m}}Z_r(\boldsymbol{X})$$

$$\overline{\boldsymbol{\Lambda}}_< = \gamma^{-1}m\boldsymbol{\Lambda}_< \qquad \overline{\boldsymbol{\Lambda}}_\leq = \gamma^{-1}m\boldsymbol{\Lambda}_\leq \qquad\quad \overline{\boldsymbol{\Lambda}}_> = \gamma^{-1}m\boldsymbol{\Lambda}_> \qquad\quad \overline{\boldsymbol{\Lambda}}_r = \gamma^{-1}m\boldsymbol{\Lambda}_r$$

Clearly, Corollary 3 and the assumption on the spectra imply that in probability as $d \to \infty$,

$$\|\overline{\boldsymbol{Z}}_<\|_{\mathrm{op}} + \|\overline{\boldsymbol{Z}}_\leq\|_{\mathrm{op}} + \|\overline{\boldsymbol{Z}}_r\|_{\mathrm{op}} + \|\overline{\boldsymbol{\Lambda}}_<\|_{\mathrm{op}} + \|\overline{\boldsymbol{\Lambda}}_\leq\|_{\mathrm{op}} + \|\overline{\boldsymbol{\Lambda}}_r\|_{\mathrm{op}} \lesssim 1 \qquad (118)$$

Then we can write $K^{(d)}$ and $K^{(d)}_\lambda$ as

$$K^{(d)}_\lambda(\boldsymbol{X}, \boldsymbol{X}) \equiv K^{(d)}(\boldsymbol{X}, \boldsymbol{X}) + \lambda\boldsymbol{I}_m = \gamma(\overline{\boldsymbol{Z}}_\leq\overline{\boldsymbol{\Lambda}}_\leq\overline{\boldsymbol{Z}}_\leq^\top + \boldsymbol{I}_m) + \Delta^{(d)}_{>r} \equiv \boldsymbol{K} + \Delta^{(d)}_{>r} \qquad (119)$$

where

$$\boldsymbol{K} = \gamma(\overline{\boldsymbol{Z}}_\leq\overline{\boldsymbol{\Lambda}}_\leq\overline{\boldsymbol{Z}}_\leq^\top + \boldsymbol{I}_m) \qquad\qquad (120)$$

Then by Sherman–Morrison–Woodbury formula

$$\boldsymbol{K}^{-1} = \gamma^{-1}\left(\boldsymbol{I}_m - \overline{\boldsymbol{Z}}_\leq\left(\overline{\boldsymbol{\Lambda}}_\leq^{-1} + \overline{\boldsymbol{Z}}_\leq^\top\overline{\boldsymbol{Z}}_\leq\right)^{-1}\overline{\boldsymbol{Z}}_\leq^\top\right) = \gamma^{-1}\left(\boldsymbol{I}_m - \overline{\boldsymbol{Z}}_\leq\overline{\boldsymbol{D}}^{-1}\overline{\boldsymbol{Z}}_\leq^\top\right) \qquad (121)$$

where

$$\overline{\boldsymbol{D}} = \overline{\boldsymbol{\Lambda}}_\leq^{-1} + \overline{\boldsymbol{Z}}_\leq^\top\overline{\boldsymbol{Z}}_\leq. \qquad\qquad (122)$$

The matrix $\overline{\boldsymbol{D}}$ plays a critical role in the remaining analysis. We have the following control regarding its eigenvalues, which says the eigenvalues of $\overline{\boldsymbol{D}}$ are away from 0 and $\infty$

**Lemma 3.** *There are constants $0 < \lambda_1 < \lambda_2$ independent of $d$ such that, in probability as $d \to \infty$,*

$$\lambda_1\boldsymbol{I} \prec \overline{\boldsymbol{D}} \prec \lambda_2\boldsymbol{I} \qquad\qquad (123)$$

We will prove the lemma later in Sec.C.6.

Note that

$$K^{(d)}_\lambda(\boldsymbol{X}, \boldsymbol{X})^{-1} = (K^{(d)}_\lambda(\boldsymbol{X}, \boldsymbol{X})^{-1}\boldsymbol{K})\boldsymbol{K}^{-1} = (\boldsymbol{I}_m + \boldsymbol{K}^{-1}\Delta^{(d)}_{>r})^{-1}\boldsymbol{K}^{-1} = (\boldsymbol{I}_m + \Delta'_d)\boldsymbol{K}^{-1} \qquad (124)$$

where $\|\Delta'_d\|_{\mathrm{op}} \to 0$ in probability since $\|\boldsymbol{K}^{-1}\|_{\mathrm{op}} \leq \gamma^{-1}$ and $\|\Delta^{(d)}_{>r}\|_{\mathrm{op}} \to 0$ in probability.

Similarly, we can write

$$M(\boldsymbol{X}, \boldsymbol{X}) \equiv \mathbb{E}K(\boldsymbol{X}, x)K(\boldsymbol{X}, x)^\top \qquad\qquad (125)$$

$$= \sum_{1 \leq k \leq r}\sum_{n \in [E_k]}(\sigma^{(d)}_{kn})^4 Z_k(\boldsymbol{X})Z_k(\boldsymbol{X})^\top + \sum_{k > r}\sum_{n \in [E_k]}(\sigma^{(d)}_{kn})^4 Z_{kn}(\boldsymbol{X})Z_{kn}(\boldsymbol{X})^\top \qquad (126)$$

$$= m^{-1}\gamma^2\overline{\boldsymbol{Z}}_\leq\overline{\boldsymbol{\Lambda}}_\leq^2\overline{\boldsymbol{Z}}_\leq^\top + m^{-1}\Delta''_d \qquad\qquad (127)$$

where

$$\Delta_d'' \equiv m \sum_{k>r} \sum_{n\in[E_k]} (\sigma_{kn}^{(d)})^4 Z_{kn}(\boldsymbol{X}) Z_{kn}(\boldsymbol{X})^\top = m \sum_{k>r} \sum_{n\in[E_k]} (\sigma_{kn}^{(d)})^4 N_{kn}^{(d)} (\boldsymbol{I}_m + \Delta_{kn}^{(d)}) \quad (128)$$

We have $\|\Delta_d''\|_{\mathrm{op}} \to 0$ in probability as $d \to \infty$, since

$$m \sum_{k>r} \sum_{n\in[E_k]} (\sigma_{kn}^{(d)})^4 N_{kn}^{(d)} \sim m \sum_{k>r} d^{-s_k} \sum_{n\in[E_k]} (\sigma_{kn}^{(d)})^2 N_{kn}^{(d)} \quad (129)$$

$$\lesssim d^{s_r} d^{-s_{r+1}} \sum_{k>r} \sum_{n\in[E_k]} (\sigma_{kn}^{(d)})^2 N_{kn}^{(d)} \quad (130)$$

$$\leq d^{s_r} d^{-s_{r+1}} \sum_{k>r} \sum_{n\in[E_k]} \hat{h}_{kn}^2 \lesssim d^{-\delta_0} \to 0 \quad (131)$$

Finally, the above estimates imply

$$\boldsymbol{H} \equiv K_\gamma^{(d)}(\boldsymbol{X}, \boldsymbol{X})^{-1} M(\boldsymbol{X}, \boldsymbol{X}) K_\gamma^{(d)}(\boldsymbol{X}, \boldsymbol{X})^{-1} \quad (132)$$

$$= (\boldsymbol{I}_m + \Delta_d') \boldsymbol{K}^{-1} M(\boldsymbol{X}, \boldsymbol{X}) \boldsymbol{K}^{-1} (\boldsymbol{I}_m + \Delta_d') \quad (133)$$

$$= m^{-1}(\boldsymbol{I}_m + \Delta_d') \boldsymbol{K}^{-1} (\gamma^2 \overline{\boldsymbol{Z}}_\leq \overline{\boldsymbol{\Lambda}}_\leq^2 \overline{\boldsymbol{Z}}_\leq^\top + \Delta_d'') \boldsymbol{K}^{-1} (\boldsymbol{I}_m + \Delta_d') \quad (134)$$

$$= m^{-1} \left( \overline{\boldsymbol{Z}}_\leq \left( \overline{\boldsymbol{\Lambda}}_\leq^{-1} + \overline{\boldsymbol{Z}}_\leq^\top \overline{\boldsymbol{Z}}_\leq \right)^{-2} \overline{\boldsymbol{Z}}_\leq^T + \Delta_d''' \right) \quad (135)$$

$$= m^{-1} \left( \overline{\boldsymbol{Z}}_\leq \overline{\boldsymbol{D}}^{-2} \overline{\boldsymbol{Z}}_\leq^T + \Delta_d''' \right) \quad (136)$$

with the error term $\|\Delta_d'''\|_{\mathrm{op}} \to 0$ in probability.

### C.3 Reduction I: Reducing the MSE to Traces

We will repeatedly use the following simple results.

**Lemma 4.** *Let $\boldsymbol{u}$ be a random vector with $\mathbb{E}\boldsymbol{u} = \boldsymbol{0}$ and $\mathbb{E}\boldsymbol{u}\boldsymbol{u}^\top = \sigma^2 \boldsymbol{I}_k$. Then for any $k \times k$ deterministic matrix $\boldsymbol{A}$,*

$$\mathbb{E}_{\boldsymbol{u}} \boldsymbol{u}^\top \boldsymbol{A} \boldsymbol{u} = \sigma^2 \operatorname{Tr}(\boldsymbol{A}) \quad (137)$$

$$\mathbb{E}_{\boldsymbol{u}} \|\boldsymbol{A}\boldsymbol{u}\|_2^2 = \sigma^2 \operatorname{Tr}(\boldsymbol{A}^\top \boldsymbol{A}). \quad (138)$$

Next, we compute the loss by decomposing it as follows. Recall that the observed labels is $f(\boldsymbol{X}) + \boldsymbol{\epsilon}$ where $\boldsymbol{\epsilon}$ is the iid noise term, which is centered and has variance $\sigma_\epsilon^2$. Thus the average test error is

$$\mathrm{Err}(\boldsymbol{X}; \lambda, \boldsymbol{F}, \boldsymbol{h})$$

$$= \mathbb{E}_{\boldsymbol{f}, \boldsymbol{\epsilon}, \boldsymbol{x}} \left| f(\boldsymbol{x}) - K^{(d)}(\boldsymbol{x}, \boldsymbol{X}) K_\gamma^{(d)}(\boldsymbol{X}, \boldsymbol{X})^{-1}(f(\boldsymbol{X}) + \boldsymbol{\epsilon}) \right|^2$$

$$= \mathbb{E}_{\boldsymbol{f}} \mathbb{E}_{\boldsymbol{x}} f^2(\boldsymbol{x}) - 2\mathbb{E}_{\boldsymbol{f}} \mathbb{E}_{\boldsymbol{x}} f(\boldsymbol{x}) K^{(d)}(\boldsymbol{x}, \boldsymbol{X}) K_\gamma^{(d)}(\boldsymbol{X}, \boldsymbol{X})^{-1} f(\boldsymbol{X}) + \mathbb{E}_{\boldsymbol{f}} f^\top(\boldsymbol{X}) \boldsymbol{H} f(\boldsymbol{X}) + \sigma_\epsilon^2 \operatorname{Tr}(\boldsymbol{H})$$

$$\equiv \mathbb{E}_{\boldsymbol{f}} I_1 + \mathbb{E}_{\boldsymbol{f}} I_2 + \mathbb{E}_{\boldsymbol{f}} I_3 + I_4.$$

Here

$$I_1 = \mathbb{E}_{\boldsymbol{x}} f^2(\boldsymbol{x}) \quad (139)$$

$$I_2 = -2\mathbb{E}_{\boldsymbol{f}} \mathbb{E}_{\boldsymbol{x}} f(\boldsymbol{x}) K^{(d)}(\boldsymbol{x}, \boldsymbol{X}) K_\gamma^{(d)}(\boldsymbol{X}, \boldsymbol{X})^{-1} f(\boldsymbol{X}) \quad (140)$$

$$I_3 = \mathbb{E}_{\boldsymbol{f}} f^\top(\boldsymbol{X}) \boldsymbol{H} f(\boldsymbol{X}) \quad (141)$$

$$I_4 = \sigma_\epsilon^2 \operatorname{Tr}(\boldsymbol{H}) \quad (142)$$

We estimate each $I_i$ individually.

**Estimate $I_1$.** We simply keep it unchanged at the moment.

**Estimate $I_2$.** Note that

$$\mathbb{E}_{\boldsymbol{x}} f(\boldsymbol{x}) K^{(d)}(\boldsymbol{x}, \boldsymbol{X}) = \sum_k \sum_{n,l} \hat{f}_{knl}(\sigma_{kn}^{(d)})^2 \phi_{knl}^{(d)}(\boldsymbol{X})^\top = \sqrt{m}\hat{\boldsymbol{f}}_{\leq}^\top \boldsymbol{\Lambda}_{\leq} \overline{\boldsymbol{Z}}_{\leq}^\top + \sum_{k>r} \sqrt{m}\hat{\boldsymbol{f}}_k^\top \boldsymbol{\Lambda}_k \overline{\boldsymbol{Z}}_k^\top$$

(143)

$$= \gamma \frac{1}{\sqrt{m}} \hat{\boldsymbol{f}}_{\leq}^\top \overline{\boldsymbol{\Lambda}}_{\leq} \overline{\boldsymbol{Z}}_{\leq}^\top + \sum_{k>r} \gamma \frac{1}{\sqrt{m}} \hat{\boldsymbol{f}}_k^\top \overline{\boldsymbol{\Lambda}}_k \overline{\boldsymbol{Z}}_k^\top$$

(144)

where $\hat{\boldsymbol{f}}_{\leq}$ is the column vector with elements $\{\hat{f}_{knl}\}_{k \leq r}$ and $\hat{\boldsymbol{f}}_<$, $\hat{\boldsymbol{f}}_>$, $\hat{\boldsymbol{f}}_k$, etc. are defined similarly. In addition,

$$f(\boldsymbol{X}) = \sqrt{m}\overline{\boldsymbol{Z}}_{\leq} \hat{\boldsymbol{f}}_{\leq} + \sqrt{m} \sum_{k>r} \overline{\boldsymbol{Z}}_k \hat{\boldsymbol{f}}_k$$

(145)

We then use the fact that $\hat{\boldsymbol{f}}_>$ is centered to eliminate all cross terms between $\hat{\boldsymbol{f}}_{\leq}$ and $\hat{\boldsymbol{f}}_>$. Denote $\mathbb{E}_>$, $\mathbb{E}_k$ and $\mathbb{E}_{kn}$ the expectation operator over $\hat{\boldsymbol{f}}_>$, over $\hat{\boldsymbol{f}}_k$ and over $\hat{\boldsymbol{f}}_{kn}$ resp. Under this notation, $\mathbb{E}_{\boldsymbol{f}} = \mathbb{E}_r \mathbb{E}_>$. Using Eq. (124) and Eq. (125),

$$\mathbb{E}_> I_2 = -2\hat{\boldsymbol{f}}_{\leq}^\top \overline{\boldsymbol{D}}^{-1} \overline{\boldsymbol{Z}}_{\leq}^\top \overline{\boldsymbol{Z}}_{\leq} \hat{\boldsymbol{f}}_{\leq} - 2\gamma \mathbb{E}_> \sum_{k>r} \hat{\boldsymbol{f}}_k^\top \left( \overline{\boldsymbol{\Lambda}}_k \overline{\boldsymbol{Z}}_k^\top K_\gamma^{(d)}(\boldsymbol{X}, \boldsymbol{X})^{-1} \overline{\boldsymbol{Z}}_k \right) \hat{\boldsymbol{f}}_k + \Delta_d \quad (146)$$

for some $\Delta_d \to 0$ in probability. The second term goes to zero since, for each $k > r$ and $n \in E_k$

$$\mathbb{E}_{kn} \hat{\boldsymbol{f}}_{kn}^\top \left( \overline{\boldsymbol{\Lambda}}_{kn} \overline{\boldsymbol{Z}}_{kn}^\top K_\gamma^{(d)}(\boldsymbol{X}, \boldsymbol{X})^{-1} \overline{\boldsymbol{Z}}_{kn} \right) \hat{\boldsymbol{f}}_{kn}$$

$$= \hat{F}_{kn}^2 / N_{kn}^{(d)} \operatorname{Tr} \overline{\boldsymbol{\Lambda}}_{kn} \left( \overline{\boldsymbol{Z}}_{kn}^\top K_\gamma^{(d)}(\boldsymbol{X}, \boldsymbol{X})^{-1} \overline{\boldsymbol{Z}}_{kn} \right)$$

$$= \hat{F}_{kn}^2 / N_{kn}^{(d)} \gamma^{-1} m (\sigma_{kn}^{(d)})^2 \operatorname{Tr} \left( \overline{\boldsymbol{Z}}_{kn}^\top K_\gamma^{(d)}(\boldsymbol{X}, \boldsymbol{X})^{-1} \overline{\boldsymbol{Z}}_{kn} \right)$$

$$\leq \gamma^{-1} \hat{F}_{kn}^2 / N_{kn}^{(d)} m (\sigma_{kn}^{(d)})^2 \|K_\gamma^{(d)}(\boldsymbol{X}, \boldsymbol{X})^{-1}\|_{\mathrm{op}} \operatorname{Tr} \left( \overline{\boldsymbol{Z}}_{kn} \overline{\boldsymbol{Z}}_{kn}^\top \right)$$

$$\lesssim \hat{F}_{kn}^2 / N_{kn}^{(d)} m (\sigma_{kn}^{(d)})^2 \operatorname{Tr}(\frac{1}{m} N_{kn}^{(d)} \frac{1}{N_{kn}^{(d)}} \boldsymbol{Z}_{kn} \boldsymbol{Z}_{kn}^\top)$$

$$= \hat{F}_{kn}^2 m (\sigma_{kn}^{(d)})^2 \frac{1}{m} \operatorname{Tr}(\boldsymbol{I}_m + \Delta_{kn}^{(d)})$$

$$\lesssim m (\sigma_{kn}^{(d)})^2 (1 + \|\Delta_{kn}^{(d)}\|_{\mathrm{op}})$$

$$= m / N_{kn}^{(d)} \hat{h}_{kn}^2 (1 + \|\Delta_{kn}^{(d)}\|_{\mathrm{op}})$$

$$\sim d^{s_r - s_k} \hat{h}_{kn}^2 (1 + \|\Delta_{kn}^{(d)}\|_{\mathrm{op}})$$

Clearly, the sum over $k > r$ and $n \in [E_k]$ of the above is bounded by $\lesssim d^{-\delta_0}$ in probability. Thus

$$\mathbb{E}_> I_2 = -2 \left( \hat{\boldsymbol{f}}_{\leq}^\top \overline{\boldsymbol{D}}^{-1} \overline{\boldsymbol{Z}}_{\leq}^\top \overline{\boldsymbol{Z}}_{\leq} \hat{\boldsymbol{f}}_{\leq} \right) + \Delta_d$$

(147)

**Estimate $I_3$.** Again, we use the fact that cross terms have mean zero

$$\mathbb{E}_> I_3 = f^\top(\boldsymbol{X}) \boldsymbol{H} f(\boldsymbol{X}) = m \left( \hat{\boldsymbol{f}}_{\leq}^\top \overline{\boldsymbol{Z}}_{\leq}^\top \boldsymbol{H} \overline{\boldsymbol{Z}}_{\leq} \hat{\boldsymbol{f}}_{\leq} + \sum_{k>r} \mathbb{E}_k \hat{\boldsymbol{f}}_k^\top \overline{\boldsymbol{Z}}_k^\top \boldsymbol{H} \overline{\boldsymbol{Z}}_k \hat{\boldsymbol{f}}_k \right)$$

(148)

$$= \hat{\boldsymbol{f}}_{\leq}^\top \overline{\boldsymbol{Z}}_{\leq}^\top \overline{\boldsymbol{Z}}_{\leq} \overline{\boldsymbol{D}}^{-2} \overline{\boldsymbol{Z}}_{\leq}^\top \overline{\boldsymbol{Z}}_{\leq} \hat{\boldsymbol{f}}_{\leq} + \sum_{k>r} \mathbb{E}_k \hat{\boldsymbol{f}}_k^\top \overline{\boldsymbol{Z}}_k^\top \overline{\boldsymbol{Z}}_{\leq} \overline{\boldsymbol{D}}^{-2} \overline{\boldsymbol{Z}}_{\leq}^\top \overline{\boldsymbol{Z}}_k \hat{\boldsymbol{f}}_k + \Delta_d$$

(149)

$$\equiv I_{3,1} + I_{3,2} + \Delta_d$$

(150)

Note that

$$I_{3,2} = \sum_{k>r} \sum_n \mathbb{E}_{kn} \operatorname{Tr} \overline{\boldsymbol{Z}}_{\leq} \overline{\boldsymbol{D}}^{-2} \overline{\boldsymbol{Z}}_{\leq}^{\top} \hat{\boldsymbol{f}}_{kn} \boldsymbol{f}_{kn}^{\top} \overline{\boldsymbol{Z}}_{kn} \overline{\boldsymbol{Z}}_{kn}^{\top} \tag{151}$$

$$= \sum_{k>r} \sum_n \hat{F}_{kn}^2 / N_k^{(d)} \operatorname{Tr} \overline{\boldsymbol{Z}}_{\leq} \overline{\boldsymbol{D}}^{-2} \overline{\boldsymbol{Z}}_{\leq}^{\top} \overline{\boldsymbol{Z}}_{kn} \overline{\boldsymbol{Z}}_{kn}^{\top} \tag{152}$$

$$= \sum_{k>r} \sum_n \hat{F}_{kn}^2 / m \operatorname{Tr} \overline{\boldsymbol{Z}}_{\leq} \overline{\boldsymbol{D}}^{-2} \overline{\boldsymbol{Z}}_{\leq}^{\top} (\boldsymbol{I}_m + \Delta_{kn}^{(d)}) \tag{153}$$

$$= m^{-1} \sum_{k>r} \sum_n \hat{F}_{kn}^2 \left( \operatorname{Tr} \overline{\boldsymbol{D}}^{-2} \overline{\boldsymbol{Z}}_{\leq}^{\top} \overline{\boldsymbol{Z}}_{\leq} \right) (1 + \Delta_d) \tag{154}$$

$$= m^{-1} \hat{F}_{>r}^2 \operatorname{Tr} \overline{\boldsymbol{D}}^{-2} \overline{\boldsymbol{Z}}_{\leq}^{\top} \overline{\boldsymbol{Z}}_{\leq} + \Delta_d' \tag{155}$$

where $\Delta_d' \to 0$ in probability since $\|\boldsymbol{D}^{-2} \overline{\boldsymbol{Z}}_{\leq}^{\top} \overline{\boldsymbol{Z}}_{\leq}\|_{\mathrm{op}} \lesssim 1$ in probability.

**Estimate $I_4$.** The $I_4$ is similar to $I_{3,2}$ above and we have

$$I_4 = m^{-1} \sigma_\epsilon^2 \operatorname{Tr} \overline{\boldsymbol{D}}^{-2} \overline{\boldsymbol{Z}}_{\leq}^{\top} \overline{\boldsymbol{Z}}_{\leq} + \Delta_d \tag{156}$$

**All Together.** Combining all terms we have

$$\mathbb{E}_> I_1 + I_2 + I_3 + I_4 \tag{157}$$

$$= \left\| \left( \boldsymbol{I} - \overline{\boldsymbol{D}}^{-1} \overline{\boldsymbol{Z}}_{\leq}^{\top} \overline{\boldsymbol{Z}}_{\leq} \right) \hat{\boldsymbol{f}}_{\leq} \right\|_2^2 + \left( 1 + m^{-1} \operatorname{Tr} \overline{\boldsymbol{D}}^{-2} \overline{\boldsymbol{Z}}_{\leq}^{\top} \overline{\boldsymbol{Z}}_{\leq} \right) \hat{F}_{>r}^2 + m^{-1} \operatorname{Tr} \overline{\boldsymbol{D}}^{-2} \overline{\boldsymbol{Z}}_{\leq}^{\top} \overline{\boldsymbol{Z}}_{\leq} \sigma_\epsilon^2 + \Delta_d \tag{158}$$

$$= \left\| \overline{\boldsymbol{D}}^{-1} \overline{\boldsymbol{\Lambda}}_{\leq}^{-1} \hat{\boldsymbol{f}}_{\leq} \right\|_2^2 + \left( 1 + m^{-1} \operatorname{Tr} \overline{\boldsymbol{D}}^{-2} \overline{\boldsymbol{Z}}_{\leq}^{\top} \overline{\boldsymbol{Z}}_{\leq} \right) \hat{F}_{>r}^2 + m^{-1} \operatorname{Tr} \overline{\boldsymbol{D}}^{-2} \overline{\boldsymbol{Z}}_{\leq}^{\top} \overline{\boldsymbol{Z}}_{\leq} \sigma_\epsilon^2 + \Delta_d \tag{159}$$

$$\equiv T_1 + T_2 + T_3 + \Delta_d \tag{160}$$

where

$$T_1 = \left\| \overline{\boldsymbol{D}}^{-1} \overline{\boldsymbol{\Lambda}}_{\leq}^{-1} \hat{\boldsymbol{f}}_{\leq} \right\|_2^2 \tag{161}$$

$$T_2 = \left( 1 + m^{-1} \operatorname{Tr} \overline{\boldsymbol{D}}^{-2} \overline{\boldsymbol{Z}}_{\leq}^{\top} \overline{\boldsymbol{Z}}_{\leq} \right) \hat{F}_{>r}^2 \tag{162}$$

$$T_3 = m^{-1} \operatorname{Tr} \overline{\boldsymbol{D}}^{-2} \overline{\boldsymbol{Z}}_{\leq}^{\top} \overline{\boldsymbol{Z}}_{\leq} \sigma_\epsilon^2 \tag{163}$$

As such, it remains to handle

$$\mathbb{E}_r (T_1 + T_2 + T_3) = \mathbb{E}_r T_1 + T_2 + T_3, . \tag{164}$$

## C.4 Reduction II: Reducing Traces to Integrals

Recall that $\overline{\boldsymbol{\Lambda}}_{\leq}$ is a diagonal matrix with elements $\gamma^{-1} m (\sigma_{kn}^{(d)})^2$, whose multiplicity is $N_{kn}^{(d)}$ for $k \leq r$ and $n \in [E_k]$. When $k = r$, $\gamma^{-1} m (\sigma_{rn}^{(d)})^2 \sim 1$, otherwise $\gamma^{-1} m (\sigma_{kn}^{(d)})^2 \sim d^{s_r - s_k}$. For convenience, we let

$$N_< \equiv N_<^{(d)}, \quad N_= \equiv N_r^{(d)}, \quad N_\leq \equiv N_\leq^{(d)} \tag{165}$$

Therefore,

$$\overline{\boldsymbol{\Lambda}}_{\leq}^{-1} = \begin{bmatrix} \Delta & 0 \\ 0 & \boldsymbol{R} \end{bmatrix} \tag{166}$$

where $\Delta$ is an $N_< \times N_<$ diagonal matrix whose entries are $\gamma m^{-1} (\sigma_{kn}^{(d)})^{-2} \sim d^{-(s_r - s_k)}$, and $\boldsymbol{R}$ is a $N_= \times N_=$ diagonal matrix whose entries are $\gamma m^{-1} (\sigma_{rn}^{(d)})^{-2} \sim 1$. As such, we claim that we

can replace $\hat{\boldsymbol{f}}_{\leq}$ by $[\boldsymbol{0}, \hat{\boldsymbol{f}}_{=}]$ in estimating $T_1$ and replace the $\Delta$ in $\overline{\boldsymbol{\Lambda}}_{\leq}^{-1}$ by any $\rho \boldsymbol{I}_{N_<}$ for any finite non-negative constant $\rho$ in estimating $T_2$. The first claim is obvious as, by Lemma 3,

$$\|\overline{\boldsymbol{D}}^{-1}\overline{\boldsymbol{\Lambda}}_{\leq}^{-1}[\boldsymbol{f}_<, \boldsymbol{0}]^\top\|_2 \leq \|\overline{\boldsymbol{D}}^{-1}\|_{\mathrm{op}}\|\overline{\boldsymbol{\Lambda}}_{\leq}^{-1}[\boldsymbol{f}_<, \boldsymbol{0}]^\top\|_2 \lesssim \lambda_1^{-1}(m\sigma_{r-1}^2\gamma^{-1}))\|\boldsymbol{f}_<\|_2 \lesssim d^{-(s_r-s_k)} \to 0 \tag{167}$$

To prove the second claim regarding estimating $T_2$, denote

$$\tilde{\boldsymbol{D}} = \begin{bmatrix} \rho\boldsymbol{I}_{N_<} & 0 \\ 0 & \boldsymbol{R} \end{bmatrix} + \overline{\boldsymbol{Z}}_{\leq}^\top \overline{\boldsymbol{Z}}_{\leq} \tag{168}$$

We claim that, in probability,

$$m^{-1}\operatorname{Tr}\left((\overline{\boldsymbol{D}}^{-2} - \tilde{\boldsymbol{D}}^{-2})\overline{\boldsymbol{Z}}_{\leq}^\top \overline{\boldsymbol{Z}}_{\leq}\right) \tag{169}$$

$$= m^{-1}\operatorname{Tr}\left((\overline{\boldsymbol{D}}^{-1}(\overline{\boldsymbol{D}}^{-1} - \tilde{\boldsymbol{D}}^{-1}) + (\overline{\boldsymbol{D}}^{-1} - \tilde{\boldsymbol{D}}^{-1})\tilde{\boldsymbol{D}}^{-1})\overline{\boldsymbol{Z}}_{\leq}^\top \overline{\boldsymbol{Z}}_{\leq}\right) \to 0 \tag{170}$$

We only bound the first term as the second term can be handled similarly.

$$m^{-1}\operatorname{Tr}\left(\overline{\boldsymbol{D}}^{-1}(\overline{\boldsymbol{D}}^{-1} - \tilde{\boldsymbol{D}}^{-1})\overline{\boldsymbol{Z}}_{\leq}^\top \overline{\boldsymbol{Z}}_{\leq}\right) \tag{171}$$

$$= m^{-1}\operatorname{Tr}\left((\overline{\boldsymbol{D}}^{-1} - \tilde{\boldsymbol{D}}^{-1})\overline{\boldsymbol{Z}}_{\leq}^\top \overline{\boldsymbol{Z}}_{\leq}\overline{\boldsymbol{D}}^{-1}\right) \tag{172}$$

$$= m^{-1}\operatorname{Tr}\left(\overline{\boldsymbol{D}}^{-1}(\tilde{\boldsymbol{D}} - \overline{\boldsymbol{D}})\tilde{\boldsymbol{D}}^{-1}\overline{\boldsymbol{Z}}_{\leq}^\top \overline{\boldsymbol{Z}}_{\leq}\overline{\boldsymbol{D}}^{-1}\right) \tag{173}$$

$$= m^{-1}\operatorname{Tr}\left((\tilde{\boldsymbol{D}} - \overline{\boldsymbol{D}})\tilde{\boldsymbol{D}}^{-1}\overline{\boldsymbol{Z}}_{\leq}^\top \overline{\boldsymbol{Z}}_{\leq}\overline{\boldsymbol{D}}^{-2}\right) \tag{174}$$

Then we use the facts that (1) the upper right $N_< \times N_<$ block matrix of $(\tilde{\boldsymbol{D}} - \overline{\boldsymbol{D}})$ is a diagonal matrix whose entries are in $[0, 1]$ and the three remaining block matrices are zeros, and (2) all entries in $\tilde{\boldsymbol{D}}^{-1}\overline{\boldsymbol{Z}}_{\leq}^\top \overline{\boldsymbol{Z}}_{\leq}\overline{\boldsymbol{D}}^{-2}$ are bounded above by a constant (each matrix in $\tilde{\boldsymbol{D}}^{-1}\overline{\boldsymbol{Z}}_{\leq}^\top \overline{\boldsymbol{Z}}_{\leq}\overline{\boldsymbol{D}}^{-2}$ has bounded operator norm[6]) to conclude that

$$\left|m^{-1}\operatorname{Tr}\left(\overline{\boldsymbol{D}}^{-1}(\overline{\boldsymbol{D}}^{-1} - \tilde{\boldsymbol{D}}^{-1})\overline{\boldsymbol{Z}}_{\leq}^\top \overline{\boldsymbol{Z}}_{\leq}\right)\right| \lesssim m^{-1}N_< \tag{175}$$

Thus

$$T_2 = \left(1 + m^{-1}\operatorname{Tr}\tilde{\boldsymbol{D}}^{-2}\overline{\boldsymbol{Z}}_{\leq}^\top \overline{\boldsymbol{Z}}_{\leq}\right)\hat{F}_{>r}^2 + \Delta_d \tag{176}$$

which will be handled later.

It remains to handle $T_1$. We make two steps of reductions in estimating $\mathbb{E}_r T_1$. The first one is to replace $\overline{\boldsymbol{\Lambda}}_{\leq}^{-1}$ by

$$\tilde{\boldsymbol{\Lambda}}_{\leq}^{-1} = \begin{bmatrix} 0 & 0 \\ 0 & \boldsymbol{R} \end{bmatrix} \tag{177}$$

and the second one is to replace $\overline{\boldsymbol{\Lambda}}_{\leq}^{-1} + \overline{\boldsymbol{Z}}_{\leq}^\top \overline{\boldsymbol{Z}}_{\leq}$ by

$$\boldsymbol{W} \equiv \begin{bmatrix} \boldsymbol{I}_{N_{\leq}} & \boldsymbol{B} \\ \boldsymbol{B}^\top & \boldsymbol{C} \end{bmatrix} \equiv \begin{bmatrix} \boldsymbol{I}_{N_<} & \overline{\boldsymbol{Z}}_<^\top \overline{\boldsymbol{Z}}_= \\ \overline{\boldsymbol{Z}}_=^\top \overline{\boldsymbol{Z}}_< & \overline{\boldsymbol{Z}}_=^\top \overline{\boldsymbol{Z}}_= + \boldsymbol{R} \end{bmatrix} \tag{178}$$

Here we have applied

$$\overline{\boldsymbol{Z}}_<^\top \overline{\boldsymbol{Z}}_< = \boldsymbol{I}_{N_<} + \Delta_d \tag{179}$$

The reason we could do so is exactly the same as we replaced $\overline{\boldsymbol{D}}$ by $\tilde{\boldsymbol{D}}$ above as we only perturb the entries in the upper $N_< \times N_<$ block by $O(1)$.

---

[6]Recall that $\overline{\boldsymbol{Z}}_{\leq}^\top \overline{\boldsymbol{Z}}_{\leq}$ follows the Marchenko-Pastur distribution.

Note that $W$ is symmetric and is also strictly positive definite, i.e. the minimal eigenvalue of $W$ is $\gtrsim 1$; see the proof in Sec.C.6. Thus by the Schur complement,

$$\mathbb{E}_r T_1 = \left\| \begin{bmatrix} I_{N_\le} & B \\ B^\top & C \end{bmatrix}^{-1} \begin{bmatrix} 0 & 0 \\ 0 & R \end{bmatrix} \begin{bmatrix} 0 \\ \hat{f}_= \end{bmatrix} \right\|_2^2 + \Delta_d \tag{180}$$

$$= \mathbb{E}_r \left\| \begin{bmatrix} -B(C - B^\top B)^{-1} R \hat{f}_= \\ (C - B^\top B)^{-1} R \hat{f}_= \end{bmatrix} \right\|_2^2 + \Delta_d \tag{181}$$

By the fact that $\hat{f}_=$ is mean zero and isotropic, we have the above equal to

$$\mathbb{E}_r T_1 = \operatorname{Tr} \left( R(C - B^\top B)^{-1} B^\top B (C - B^\top B)^{-1} R + R(C - B^\top B)^{-2} R \right) \hat{F}_r^2 / N_= + \Delta_d \tag{182}$$

$$= \operatorname{Tr}(R C^{-2} R) \hat{F}_r^2 / N_= + \Delta_d' + \Delta_d \tag{183}$$

where

$$\Delta_d' = \operatorname{Tr} \left( R(C - B^\top B)^{-1} B^\top B (C - B^\top B)^{-1} R \right) / N_= + \tag{184}$$

$$\operatorname{Tr} \left( R((C - B^\top B)^{-2} - C^{-2}) R \right) / N_= . \tag{185}$$

We claim that $\Delta_d' \to 0$ in probability. For the first term, we have

$$\operatorname{Tr} \left( R(C - B^\top B)^{-1} B^\top B (C - B^\top B)^{-1} R \right) / N_= \tag{186}$$

$$= \operatorname{Tr} \left( (C - B^\top B)^{-1} R^2 (C - B^\top B)^{-1} B^\top B \right) / N_= \tag{187}$$

$$= \| (C - B^\top B)^{-1} R^2 (C - B^\top B)^{-1} \|_{\mathrm{op}} \operatorname{Tr} \left( B^\top B \right) / N_= \tag{188}$$

$$\le \| (C - B^\top B)^{-1} \|_{\mathrm{op}} \| R^2 \|_{\mathrm{op}} \| (C - B^\top B)^{-1} \|_{\mathrm{op}} \operatorname{Tr} \left( B^\top B \right) / N_= \tag{189}$$

$$\lesssim \operatorname{Tr} \left( B^\top B \right) / N_= \sim N_< / N_= \to 0 \tag{190}$$

in probability as $d \to \infty$. We have used

$$\| R \|_{\mathrm{op}} \lesssim 1 \tag{191}$$

$$\| (C - B^\top B)^{-1} \|_{\mathrm{op}} \le \| W^{-1} \|_{\mathrm{op}} \lesssim 1 \tag{192}$$

$$\frac{1}{N_=} \operatorname{Tr} \left( B^\top B \right) \sim N_< / N_= . \tag{193}$$

The last one holds because $B^\top B$ is a rank $N_<$ matrix with operator norm $\lesssim 1$. Note that this also implies that $B^\top B$ has at most $N_<$ many non-zero singular values, which is upper bounded by $\lesssim 1$. Using Von Neumann's trace inequalities, for any matrix $A$, we have

$$| \operatorname{Tr} A B^\top B | \le \sum_j \sigma_j(A) \sigma_j(B^\top B) \lesssim N_< \| A \|_{\mathrm{op}} \tag{194}$$

where $\sigma_j(A)$ is the $j$-th (in descending order) singular value of a matrix $A$. Now we proceed to control the second term. Note that

$$(C - B^\top B)^{-2} - C^{-2} \tag{195}$$

$$= (C - B^\top B)^{-2} - (C - B^\top B)^{-1} C^{-1} + (C - B^\top B)^{-1} C^{-1} - C^{-2} \tag{196}$$

$$= (C - B^\top B)^{-2} B^\top B C^{-1} + (C - B^\top B)^{-1} B^\top B C^{-2} \tag{197}$$

As such, by Eq. (194) we have

$$| \operatorname{Tr} R(C - B^\top B)^{-2} B^\top B C^{-1} R | / N_= \tag{198}$$

$$= | \operatorname{Tr} C^{-1} R^2 (C - B^\top B)^{-2} B^\top B | / N_= \tag{199}$$

$$\lesssim N_< / N_= \| C^{-1} R^2 (C - B^\top B)^{-2} \|_{\mathrm{op}} \to 0 . \tag{200}$$

The other term can be bounded similarly. This finishes the proof of $\Delta_d' \to 0$ in probability. To sum up, we have the test error to be

$$\mathrm{Err}(X; \lambda, F, h) = \operatorname{Tr} \left( R^2 C^{-2} \right) / N_= \hat{F}_r^2 + \left( 1 + m^{-1} \operatorname{Tr} \tilde{D}^{-2} \overline{Z}_\le^\top \overline{Z}_\le \right) \hat{F}_{>r}^2 + \tag{201}$$

$$m^{-1} \sigma_\epsilon^2 \operatorname{Tr} \tilde{D}^{-2} \overline{Z}_\le^\top \overline{Z}_\le + \Delta_d \tag{202}$$

**Generalization Error via Marchenko-Pastur** The next step is to reduce the traces to the integral form when $d \to \infty$. That is evaluating the followings as $d \to \infty$,

$$\mathrm{Tr}\left(\boldsymbol{R}^2 \boldsymbol{C}^{-2}\right)/N_= \quad \text{and} \quad m^{-1}\sigma_\epsilon^2 \, \mathrm{Tr}\, \tilde{\boldsymbol{D}}^{-2}\overline{\boldsymbol{Z}}_\leq^\top \overline{\boldsymbol{Z}}_\leq \tag{203}$$

We begin with the simpler case $E_r = 1$ and then consider $E_r > 1$.

**The $E_r = 1$ case.** I.e., there is only one eigenspace with eigenvalues $\sim d^{-s_r}$. This is the case for one-hidden layer convolutional kernels and dot-product kernels. In this case, $n = 0$ and

$$\boldsymbol{R} = (\overline{\xi}_r^{(d)})^{-1}\boldsymbol{I}_{N_\leq}, \quad \text{with} \quad \overline{\xi}_r^{(d)} = \gamma^{-1}m(\sigma_{rn}^{(d)})^2 = \frac{m}{N_=}\hat{h}_r^2\gamma^{-1} \to \overline{\xi}_r = \alpha^{-1}\hat{h}_r^2\gamma^{-1} \tag{204}$$

Choosing $\rho = (\overline{\xi}_r^{(d)})^{-1}$ and applying Theorem 1, we have when[7] $N_=/m \to \alpha \in (0, \infty)$

$$\overline{\xi}_r^{-2}\,\mathrm{Tr}\left(\boldsymbol{C}^{-2}\right)/N_= \longrightarrow \int (1 + \overline{\xi}_r t)^{-2}\mu_\alpha(t)dt \tag{205}$$

$$\frac{N_\leq}{m}\frac{1}{N_\leq}\,\mathrm{Tr}\,\tilde{\boldsymbol{D}}^{-2}\overline{\boldsymbol{Z}}_\leq^\top\overline{\boldsymbol{Z}}_\leq \longrightarrow \alpha\overline{\xi}_r^2\int t(1+\overline{\xi}_r t)^{-2}\mu_\alpha(t)dt \tag{206}$$

Therefore,

$$\mathrm{Err}(\boldsymbol{X};\lambda,\boldsymbol{F},\boldsymbol{h}) = \left(\hat{F}_r^2\cdot\int\frac{\mu_\alpha(t)}{(1+\overline{\xi}_r t)^2}dt + \hat{F}_{>r}^2\right) + \left(\hat{F}_{>r}^2 + \sigma_\epsilon^2\right)\cdot\alpha\overline{\xi}_r^2\int\frac{t\mu_\alpha(t)}{(1+\overline{\xi}_r t)^2}dt + \Delta_d \tag{207}$$

Both integrals have closed form formulas and they are computed in Sec.C.5.

**The $E_r > 1$ Case.** Recall that $\boldsymbol{R}$ is a diagonal matrix with entries $\gamma(m(\sigma_{rn}^{(d)})^2)^{-1}$ whose multiplicity is $N_{rn}^{(d)}$. We assume the limiting density exist

$$\gamma(m(\sigma_{rn}^{(d)})^2)^{-1} \to \gamma\alpha\hat{h}_{rn}^{-2} \quad \text{and} \quad N_{rn}^{(d)}/\sum_{n\in[E_k]}N_{rn}^{(d)} \to \tau_{rn} \tag{208}$$

and let $\nu_{\boldsymbol{h}}(r)$ denote this distribution. For convenience, we still $\boldsymbol{R}$ to represent a (sequence of) diagonal matrix with limiting spectral $\nu_{\boldsymbol{h}}(r)$. By our assumptions on $\boldsymbol{h}$, the support of $\nu_{\boldsymbol{h}}(r)$ is bounded away from 0 and $\infty$. Thus, ignoring vanishing correction term between $\overline{\boldsymbol{Z}}_\leq^\top\overline{\boldsymbol{Z}}_\leq$ and $\overline{\boldsymbol{Z}}_=^\top\overline{\boldsymbol{Z}}_=$, we need to compute the limit of the following

$$\frac{1}{N_\leq}\,\mathrm{Tr}\,\boldsymbol{R}^2(\boldsymbol{R}+\overline{\boldsymbol{Z}}_\leq^\top\overline{\boldsymbol{Z}}_\leq)^{-2} = \boldsymbol{R}^{1/2}(1+\boldsymbol{R}^{-1/2}\overline{\boldsymbol{Z}}_\leq^\top\overline{\boldsymbol{Z}}_\leq\boldsymbol{R}^{-1/2})^{-1}\boldsymbol{R}^{-1}(1+\boldsymbol{R}^{-1/2}\overline{\boldsymbol{Z}}_\leq^\top\overline{\boldsymbol{Z}}_\leq\boldsymbol{R}^{-1/2})^{-1}\boldsymbol{R}^{1/2} \tag{209}$$

$$\frac{1}{N_\leq}\,\mathrm{Tr}(\boldsymbol{R}+\overline{\boldsymbol{Z}}_\leq^\top\overline{\boldsymbol{Z}}_\leq)^{-2}\overline{\boldsymbol{Z}}_\leq^\top\overline{\boldsymbol{Z}}_\leq = \frac{1}{N_\leq}\,\mathrm{Tr}\left((\boldsymbol{R}+\overline{\boldsymbol{Z}}_\leq^\top\overline{\boldsymbol{Z}}_\leq)^{-1} - (\boldsymbol{R}+\overline{\boldsymbol{Z}}_\leq^\top\overline{\boldsymbol{Z}}_\leq)^{-2}\boldsymbol{R}\right) \tag{210}$$

.

To evaluate the limit, we may need extra assumptions on the eigenfunctions $\phi_{knl}^{(d)}$ to ensure $\boldsymbol{R}$ and $\boldsymbol{Z}_\leq$ are asymptotically free. Nevertheless, under the freeness assumption, computing self-consistent equations that characterize the asymptotic values of the trace objects in Eq. (209) and Eq. (210) is then straightforward using tools from operator-valued free probability [33]. We do not elaborate on the details here, but refer the reader so many related works for examples of how to apply these tools [14, 1, 2, 38, 39].

## C.5 Computing the Integrals.

It remains to compute the above integrals. Note that

$$\overline{\xi}_r^2\int\frac{t\mu_\alpha(t)}{(1+\overline{\xi}_r t)^2}dt = \overline{\xi}_r\left(\int\frac{\mu_\alpha(t)}{(1+\overline{\xi}_r t)}dt - \int\frac{\mu_\alpha(t)}{(1+\overline{\xi}_r t)^2}dt\right) \tag{211}$$

---

[7]Note that $N_\leq = N_=(1 + o(1))$

As such we only need to compute, for $k = 1$ and $k = 2$,

$$\zeta_\alpha(\xi; \alpha, k) = \int \frac{\mu_\alpha(t)}{(1 + \xi t)^k} dt \tag{212}$$

Note that one only needs $\zeta_\alpha(\xi; \alpha, 1)$ as $\zeta_\alpha(\xi; \alpha, k)$ can be obtained from $\zeta_\alpha(\xi; \alpha, k - 1)$ by taking derivative w.r.t. $\bar{\xi}$. Denote $b_\pm = (1 \pm \sqrt{\alpha})^2$ and $\Delta = \alpha_+ - \alpha_-$. Then

$$\mu_\alpha(t) = \left(1 - \frac{1}{\alpha}\right)^+ \delta_0(t) + \frac{\sqrt{(\alpha_+ - t)(t - \alpha_-)}}{2\pi\alpha t} \mathbf{1}_{[\alpha_-, \alpha_+]}(t) dt \tag{213}$$

With $b = (1 + \bar{\xi}_r \alpha_-)/(\bar{\xi}_r(\alpha_+ - \alpha_-))$ and $c = \bar{\xi}_r \alpha_-/(\bar{\xi}_r(\alpha_+ - \alpha_-)) = \alpha_-/(\alpha_+ - \alpha_-)$,

$$\int (1 + \bar{\xi}_r t)^{-k} \mu_\alpha(t) dt \tag{214}$$

$$= \left(1 - \frac{1}{\alpha}\right)^+ + \int_{[\alpha_-, \alpha_+]} (1 + \bar{\xi}_r t)^{-k} \frac{\sqrt{(\alpha_+ - t)(t - \alpha_-)}}{2\pi\alpha t} dt \tag{215}$$

$$= \left(1 - \frac{1}{\alpha}\right)^+ + \frac{1}{2\pi\alpha\bar{\xi}_r} (\bar{\gamma}(\alpha_+ - \alpha_-))^{1-k} \int_0^1 (b + t)^{-k} (c + t)^{-1} \sqrt{t(1 - t)} dt \tag{216}$$

Thanks to wolframalpha.com, we have, after doing some algebra,

$$\int_0^1 \frac{\sqrt{t(1 - t)}}{(t + b)(t + c)} dt = \pi \left(-1 + \frac{1 + b + c}{\sqrt{bc(1 + b)(1 + c)}}\right) \tag{217}$$

$$\int_0^1 \frac{\sqrt{t(1 - t)}}{(t + b)^2(t + c)} dt = \frac{\pi}{2\sqrt{b^2 + b}((b + c + 2bc) + 2\sqrt{(b + 1)(c + 1)bc})} \tag{218}$$

## C.6 Proof of Lemma 3.

Note that this lemma is trivial if $\lim_{d \to \infty} m/N_{\leq r}^{(d)} = \alpha^{-1} > 1$ as $\overline{\mathbf{Z}}_\leq^\top \overline{\mathbf{Z}}_\leq$ follows the Marchenko-Pastur distribution, and the smallest eigenvalues is bounded from below by $\alpha_- = (1 - \sqrt{\alpha})^2$. When $\alpha^{-1} \leq 1$, we need to use the regularization term $\overline{\mathbf{\Lambda}}_\leq^{-1}$. We provide the details below.

Recall that $\overline{\mathbf{Z}}_\leq^\top = [\overline{\mathbf{Z}}_<^\top, \overline{\mathbf{Z}}_r^\top]$, where $\overline{\mathbf{Z}}_<^\top$ is a $m \times N_{<r}^{(d)}$ matrix consisting of low frequency modes and $\overline{\mathbf{Z}}_r^\top$ is a $m \times N_r^{(d)}$ is a matrix consisting of critical frequency modes. We have $N_{<r}^{(d)} \sim d^{s_r - 1}$, $N_r^{(d)} \sim N_{\leq r}^{(d)} \sim d^{s_r}$ and

$$\overline{\mathbf{Z}}_<^\top \overline{\mathbf{Z}}_< = \mathbf{I}_{N_{<r}^{(d)}} + \Delta_d \tag{219}$$

where $\mathbb{E}\|\Delta_d\|_{op} \to 0$ as $d \to \infty$ in probability. Let $\mathbf{u} = [\beta_< \mathbf{e}_<^\top, \beta_r \mathbf{e}_r^\top]^\top$ be a unit vector in $\mathbb{R}^{N_{\leq r}^{(d)}}$, where $\mathbf{e}_<$ and $\mathbf{e}_r$ are unit vectors in $\mathbb{R}^{N_{<r}^{(d)}}$ and $\mathbb{R}^{N_r^{(d)}}$ resp., and $\beta_<^2 + \beta_r^2 = 1$. We want to show that for some $\lambda_1 > 0$

$$\lambda_1 \leq \mathbf{u}^\top D\mathbf{u} = \mathbf{u}^\top \overline{\mathbf{\Lambda}}_\leq^{-1} \mathbf{u} + \mathbf{u}^\top \overline{\mathbf{Z}}_\leq^\top \overline{\mathbf{Z}}_\leq \mathbf{u} \tag{220}$$

Note that the entries in $\overline{\mathbf{\Lambda}}_\leq^{-1}$ corresponding to the critical-frequencies are $m(\sigma_{nr}^{(d)})^2 \sim 1$. Thus there is a constant $c > 0$ such that

$$\mathbf{u}^\top \overline{\mathbf{\Lambda}}_\leq^{-1} \mathbf{u} \geq c\beta_r^2 \tag{221}$$

In addition, if $C \equiv \|\overline{\mathbf{Z}}_r \mathbf{e}_r\|_2$ then $C \leq 2\alpha_+$ in probability. Thus by the triangle inequality,

$$\mathbf{u}^\top D\mathbf{u} \geq c\beta_r^2 + \|\beta_< \overline{\mathbf{Z}}_< \mathbf{e}_< + \beta_r \overline{\mathbf{Z}}_r \mathbf{e}_r\|_2^2 \tag{222}$$

$$\geq c\beta_r^2 + (\|\beta_< \overline{\mathbf{Z}}_< \mathbf{e}_<\|_2 - \|\beta_r \overline{\mathbf{Z}}_r \mathbf{e}_r\|_2)^2 \tag{223}$$

$$\geq c\beta_r^2 + ((1 - \Delta_d)|\beta_<| - C|\beta_r|)^2 \tag{224}$$

where $\Delta_d \to 0$ in probability. If $C|\beta_r| < \frac{1}{2}|\beta_<|$, the above is greater than $(1/2 - \Delta_d)\beta_<^2 + c\beta_r^2 \gtrsim 1$; otherwise $C|\beta_r| \geq \frac{1}{2}|\beta_<|$ and $\boldsymbol{u}^\top D\boldsymbol{u} \geq c\beta_r^2 \geq c(\frac{1}{2C}\beta_<)^2$ and as a result

$$\boldsymbol{u}^\top D\boldsymbol{u} \geq 2c\beta_r^2/2 \geq (c\beta_r^2 + c(\frac{1}{2C}\beta_<)^2)/2 \geq c\min(1, \frac{1}{2C})^2/2 \gtrsim 1 \tag{225}$$

The other direction is easier as both $\overline{\boldsymbol{\Lambda}}_{\leq}^{-1}$ and $\overline{\boldsymbol{Z}}_{\leq}^\top \overline{\boldsymbol{Z}}_{\leq}$ have operator norms bounded above.

## C.7 Proof of Claim 1

The proof is split into two part: the ultra-high frequency parts $k \geq j_0$ and the median-high-frequency part, $r < k < j_0$. The first part is done by a moment-based calculation and the second part is done by matrix concentration [40].

**Controlling the Ultra-high-frequency.**   Recall that

$$\Delta_{kn}^{(d)} \equiv \frac{1}{N_{kn}^{(d)}} Z_{kn}(\boldsymbol{X}) Z_{kn}(\boldsymbol{X})^\top - \boldsymbol{I}_m\,. \tag{226}$$

By **Assumption (4.)**, the diagonals are zero and we have

$$\Delta_{kn}^{(d)} = [Z_{kn}(\boldsymbol{x}_i)^\top Z_{kn}(\boldsymbol{x}_j)/N_{kn}^{(d)}]_{i,j\in[m], i\neq j} \tag{227}$$

Then

$$\mathbb{E}\|\Delta_{kn}^{(d)}\|_{\mathrm{op}}^2 \leq \mathbb{E}\|\Delta_{kn}^{(d)}\|_{\mathrm{F}}^2 = \mathbb{E}\sum_{i\neq j} |Z_{kn}(\boldsymbol{x}_i)^\top Z_{kn}(\boldsymbol{x}_j)/N_{kn}^{(d)}|^2 \tag{228}$$

$$= \frac{1}{(N_{kn}^{(d)})^2}\mathbb{E}\sum_{i\neq j}\sum_{l,l'} \phi_{knl}^{(d)}(\boldsymbol{x}_i)\phi_{knl}^{(d)}(\boldsymbol{x}_j)\phi_{knl'}^{(d)}(\boldsymbol{x}_i)\phi_{knl'}^{(d)}(\boldsymbol{x}_j) \tag{229}$$

$$= \frac{1}{(N_{kn}^{(d)})^2}\mathbb{E}\sum_{i\neq j}\sum_{l} \phi_{knl}^{(d)}(\boldsymbol{x}_i)^2\phi_{knl}^{(d)}(\boldsymbol{x}_j)^2 \tag{230}$$

$$= \frac{1}{N_{kn}^{(d)}}m(m-1) \leq \frac{1}{N_{kn}^{(d)}}m^2 \tag{231}$$

Recall that $E_k$ grows at most exponentially, i.e. $E_k \leq C^k$ for some constant $C$. Thus, choosing $d$ large enough such that $Cd^{-\delta_0/4} < 1$ and summing over $k > j_0 \equiv [4s_r/\delta_0 + 4] + 4$,

$$\mathbb{E}\sum_{k>j_0}\sum_{n\in E_k} \|\Delta_{kn}^{(d)}\|_{\mathrm{op}} \lesssim \sum_{k>j_0} C^k(m^2/N_{kn}^{(d)})^{1/2} \tag{232}$$

$$\lesssim \sum_{k>j_0} C^k d^{-s_k/2+s_r} \tag{233}$$

$$\leq \sum_{k>j_0} C^k d^{-k\delta_0/2+s_r} \tag{234}$$

$$\lesssim \sum_{k>j_0} d^{-k\delta_0/4+s_r} \lesssim d^{-j_0\delta_0/4+s_r} \lesssim d^{-\delta_0} \tag{235}$$

**Controlling the Median-high-frequency.**   It remains to show, for some $\epsilon > 0$

$$\mathbb{E}\sum_{r<k\leq j_0}\sum_{n\in E_k} \|\Delta_{kn}^{(d)}\|_{\mathrm{op}} \lesssim d^{-\epsilon}\,. \tag{236}$$

As there are only finitely many terms in this sum, we only need to show that for each $k$ and $n$,

$$\mathbb{E}\|\Delta_{kn}^{(d)}\|_{\mathrm{op}} \lesssim d^{-\epsilon}\,.$$

We use the following theorem from Vershynin [40] regarding matrix concentration to prove this claim.

**Theorem 6** (Theorem 5.62 Vershynin [40]). *Let $\boldsymbol{A}$ be a $N \times m$ ($N \geq m$) matrix whose columns $A_j$ are independent isotropic random vectors in $\mathbb{R}^N$ with $\|A_j\|_2 = N$ a.s. Let $K$ be defined as*

$$K = \frac{1}{N}\mathbb{E}\max_{j\leq m}\sum_{i\in[m],i\neq j}|A_j^\top A_i|^2 \tag{237}$$

*Then*

$$\mathbb{E}\|\boldsymbol{A}^\top \boldsymbol{A}/N - \boldsymbol{I}_m\|_{\mathrm{op}} \lesssim \sqrt{K\log(m)/N} \tag{238}$$

We apply this theorem to $\boldsymbol{A} = Z_{kn}(\boldsymbol{X})^\top$. The columns of $Z_{kn}(\boldsymbol{X})^\top$ are $A_j = \phi_{kn}^{(d)}(\boldsymbol{x}_j)$, $j \in [m]$ which are independent. Let $N = N_{kn}^{(d)}$. By Assumption (4.),

$$A_j^\top A_j = \sum_l \phi_{knl}^{(d)}(\boldsymbol{x}_j)^2 = N. \tag{239}$$

We claim that $K \lesssim_{k,q} m^{1+\frac{1}{q}}$ for any $q \geq 1$. Indeed, let

$$B_j = \sum_{i\in[m],i\neq j}|A_j^\top A_i|^2 \tag{240}$$

We then remove the maximal function by paying an $m^{1/q}$ factor

$$K = \frac{1}{N}\mathbb{E}\max_{j\leq m}B_j \leq \frac{1}{N}m^{1/q}|\mathbb{E}B_j^q|^{1/q} = \frac{1}{N}m^{1/q}\left|\mathbb{E}(\sum_{i\in[m],i\neq j}|A_j^\top A_i|^2)^{2q/2}\right|^{1/q} \tag{241}$$

Next we apply the Minkowski inequality to swap the $L^{2q}$-norm and the $l^2$-norm,

$$K \leq \frac{1}{N}m^{1/q}\sum_{i\in[m],i\neq j}(\mathbb{E}|A_j^\top A_i|^{2q})^{1/2q\times2} \leq \frac{1}{N}m^{1/q+1}(\mathbb{E}|A_j^\top A_i|^{2q})^{1/2q\times2} \leq C_{k,q}^2 m^{1/q+1} \tag{242}$$

if $(\mathbb{E}|A_j^\top A_i|^{2q})^{1/2q\times2} \leq C_{k,q}^2 N$, which is done by hypercontractivities below. Indeed, for $\boldsymbol{x}_j$ fixed, $Z_{rn}(\boldsymbol{x}_j)^\top Z_{rn}(\boldsymbol{x}_i)$ is a linear combination of $\phi_{knl}^{(d)}$, **Assumption (2.)** gives

$$\mathbb{E}_{\boldsymbol{x}_i}|A_j^\top A_i|^{2q} = \mathbb{E}_{\boldsymbol{x}_i}|Z_{rn}(\boldsymbol{x}_j)^\top Z_{rn}(\boldsymbol{x}_i)|^{2q} \tag{243}$$

$$\leq \left(C_{k,q}(\mathbb{E}_{\boldsymbol{x}_i}|Z_{rn}(\boldsymbol{x}_j)^\top Z_{rn}(\boldsymbol{x}_i)|^2)^{1/2}\right)^{2q} \tag{244}$$

$$= C_{k,q}^{2q}N^q \tag{245}$$

where we applied

$$\mathbb{E}_{\boldsymbol{x}_i}|Z_{rn}(\boldsymbol{x}_j)^\top Z_{rn}(\boldsymbol{x}_i)|^2 = \mathbb{E}_{\boldsymbol{x}_i}|\sum_l \phi_{knl}^{(d)}(\boldsymbol{x}_i)\phi_{knl}^{(d)}(\boldsymbol{x}_j)|^2 \tag{246}$$

$$= \mathbb{E}_{\boldsymbol{x}_i}\sum_{ll'}\phi_{knl}^{(d)}(\boldsymbol{x}_i)\phi_{knl}^{(d)}(\boldsymbol{x}_j)\phi_{knl'}^{(d)}(\boldsymbol{x}_i)\phi_{knl'}^{(d)}(\boldsymbol{x}_j) \tag{247}$$

$$= \sum_l \phi_{knl}^{(d)}(\boldsymbol{x}_j)^2 \tag{248}$$

$$= N \tag{249}$$

Therefore with $\boldsymbol{A} = Z_{kn}(\boldsymbol{X})^\top$, we have

$$\mathbb{E}\|\Delta_{kn}^{(d)}\|_{\mathrm{op}} = \mathbb{E}\|Z_{kn}(\boldsymbol{X})Z_{kn}(\boldsymbol{X})^\top/N_{kn}^{(d)} - \boldsymbol{I}_m\|_{\mathrm{op}} \lesssim_{k,q} \sqrt{m^{1+1/q}\log m/N_{kn}^{(d)}}. \tag{250}$$

For each fixed $k > r$, $s_k - s_r \geq \delta_0$, by choosing $q$ sufficiently large (depending on $k$ and $\delta_0$), we have

$$\mathbb{E}\|\Delta_{kn}^{(d)}\|_{\mathrm{op}} \lesssim_k d^{-\delta_0/2}. \tag{251}$$

# D    Additional Plots

To simulate the learning curves for higher-order scalings, e.g. $r = 4$, we must chose $d$ small. As such, we are in a strong finite-size correction regime. In this section, we vary $d$ to visualize the finite-size effect of the predictions. Note that for larger $d \, (= 60$ here$)$, we can only simulate up to the quadratic scaling. For smaller $d \, (d = 10)$, we observe noticeable finite-size correction. However, the predictions match the simulations quite well. When $d$ become larger $d = 60$, the predicted learning curves match the simulation almost perfectly.

# E    Further Analysis

## E.1    Reducing finite-size Effect.

There are two non-obvious improvements in our results that lead to near perfect agreements between simulations and predictions even for small $d$. The first one is to use $m = N(d, \leq r)$ to compute $r$-th peak vs. $m = N(d, r)$ (or $m = d^r / r!$). As it is shown in Fig. 9 (a), using $m = N(d, r)$ as the peak in the theoretical prediction, the prediction is a bit off to the left. The second one is to use the sum over all contributions from all critical scaling $m = N(d, \leq r)$ (i.e., Eq. (21)) rather than the contribution from a single critical scaling (i.e., Eq. (18).) As it is shown in Fig. 9 (b), the predictions are a bit smaller than the simulations when using the latter. These two improvements together lead to accurate agreement Fig. 9 (c).

## E.2    Choosing the number of peaks by choosing the right regularization.

Recall that the height of the $r$-th variance term scales like

$$\xi_r(\hat{\boldsymbol{h}}, \lambda, 1)^{1/2} = \left( \frac{\hat{h}_r^2}{\lambda + \hat{h}_{>r}^2} \right)^{1/2}. \tag{252}$$

If $\hat{h}_r^2 \gg \hat{h}_{>r}^2$ and $\lambda \leq \hat{h}_{>r}^2$, then $\xi_r(\hat{\boldsymbol{h}}, \lambda)^{1/2}$ is large, which could lead to a peak at $m = N(d, \leq r)$. To eliminate this peak, we could choose $\lambda \sim \hat{h}_r^2$ which implies $\xi_r(\hat{\boldsymbol{h}}, \lambda, 1) \lesssim 1$. We verify this observation in Fig. 10. When $\lambda = 0$, the unregularized learning curve have 4 peaks. By increasing $\lambda$ to $1e-7, 1e-5, 1e-3, 1e-1$, the number of peaks are reduced to 3, 2, 1, 0, respectively. A similar result has also been observed in a linear design setting [41]. The similarity between the linear design in [41] and the nonlinear design here shouldn't be surprising, as we prove a "Gaussian equivalence conjecture," which implies that the polynomial scalings are essential "replicas" of linear designs with different scales.

## E.3    Natural Data vs. Spherical Data

We compare the spectrum of the NTKs of CIFAR10 associated with three architectures (FCN: fully-connected networks, CNN-VEC: convolutional networks without pooling, and CNN-GAP: convolutional networks with a global average pooling) against the one-layer convolutional kernels with spherical-type of data. Recall that the larger spectral gap between eigenspaces triggers the multiple-descent phenomenon. This phenomenon disappears, and the learning curve becomes monotonic when the spectral gap is small. Fortunately, for CIFAR10, the spectrum of the NTKs are continuous, and the learning curves are monotonic (power-law decay.) As such, there is still a gap between our results/assumptions and natural data.

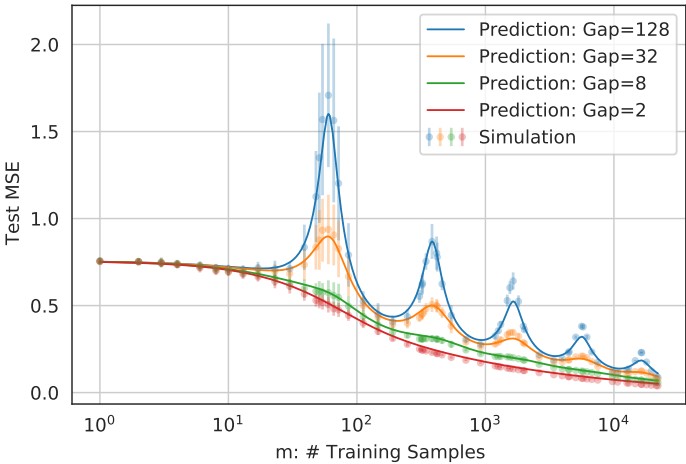

Figure 6: **Tiny** $d = 10$

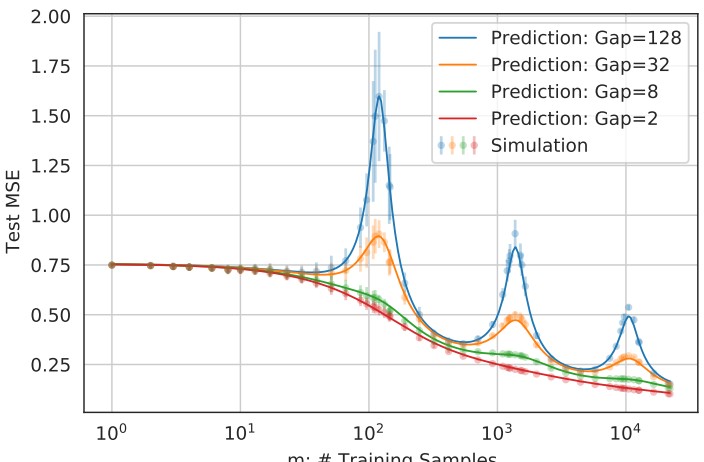

Figure 7: **Small** $d = 20$

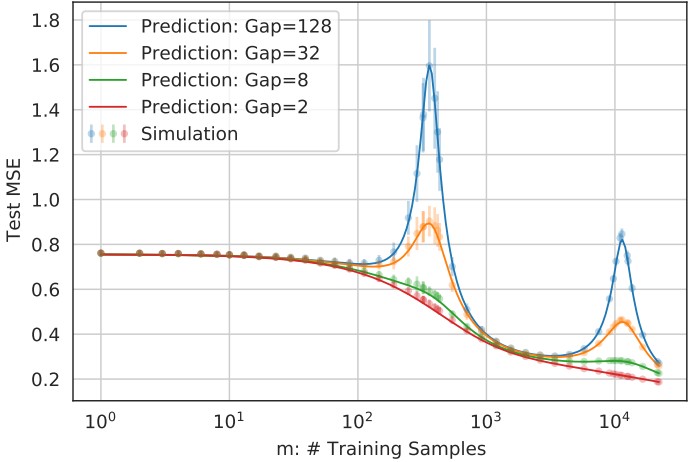

Figure 8: **Large** $d = 60$

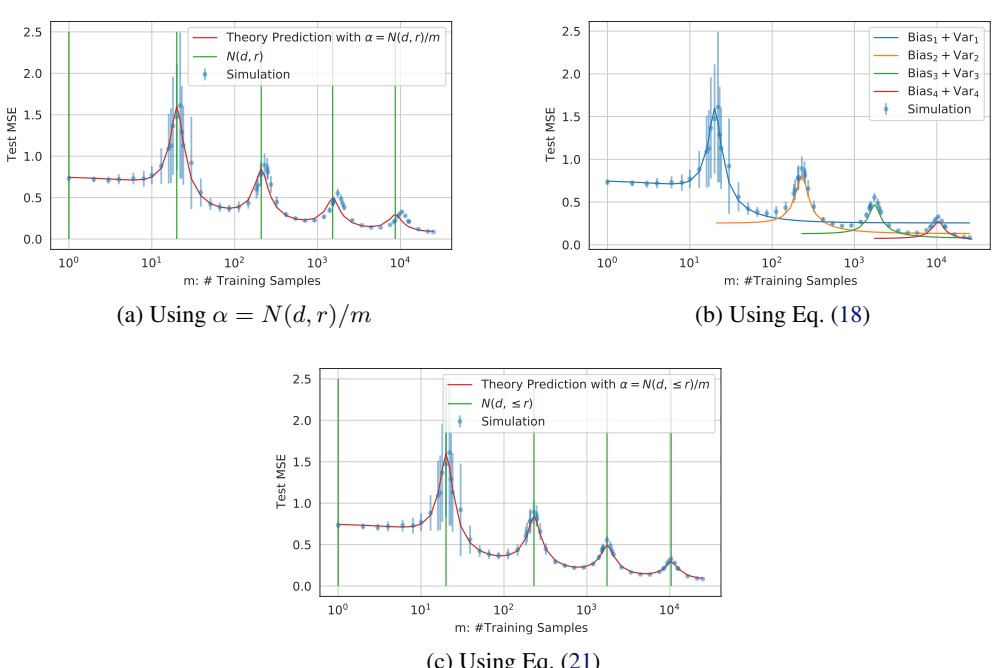

(a) Using $\alpha = N(d, r)/m$

(b) Using Eq. (18)

(c) Using Eq. (21)

Figure 9: **Two improvements reduce the finite-size effect.** (a) The theoretical prediction is a bit off to the left when estimating $\alpha$ using $N(d, r)/m$. (b) The prediction from Eq. (18) is a bit smaller than the simulation due to the finite-size effect. (c) Almost perfect agreement between the prediction and the simulation after two improvements (1) replacing Eq. (18) by Eq. (21) and (2) estimating $\alpha$ with $N(d, \leq r)/m$ rather than $N(d, r)/m$. Here $d = 20$ and $p = 1$, i.e., inner product kernel.

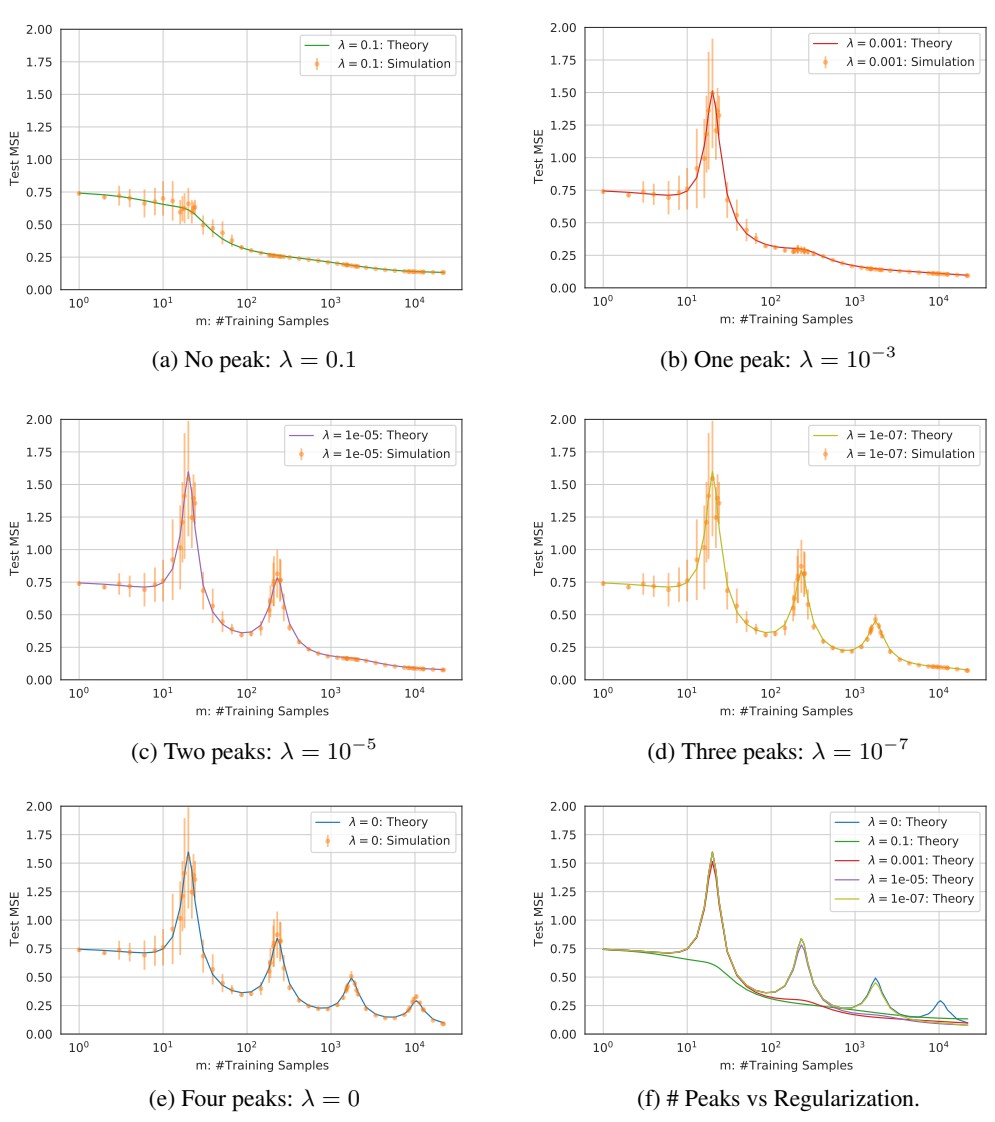

(a) No peak: $\lambda = 0.1$

(b) One peak: $\lambda = 10^{-3}$

(c) Two peaks: $\lambda = 10^{-5}$

(d) Three peaks: $\lambda = 10^{-7}$

(e) Four peaks: $\lambda = 0$

(f) # Peaks vs Regularization.

Figure 10: **Controlling the number of peaks by varying the strength of regularization.**

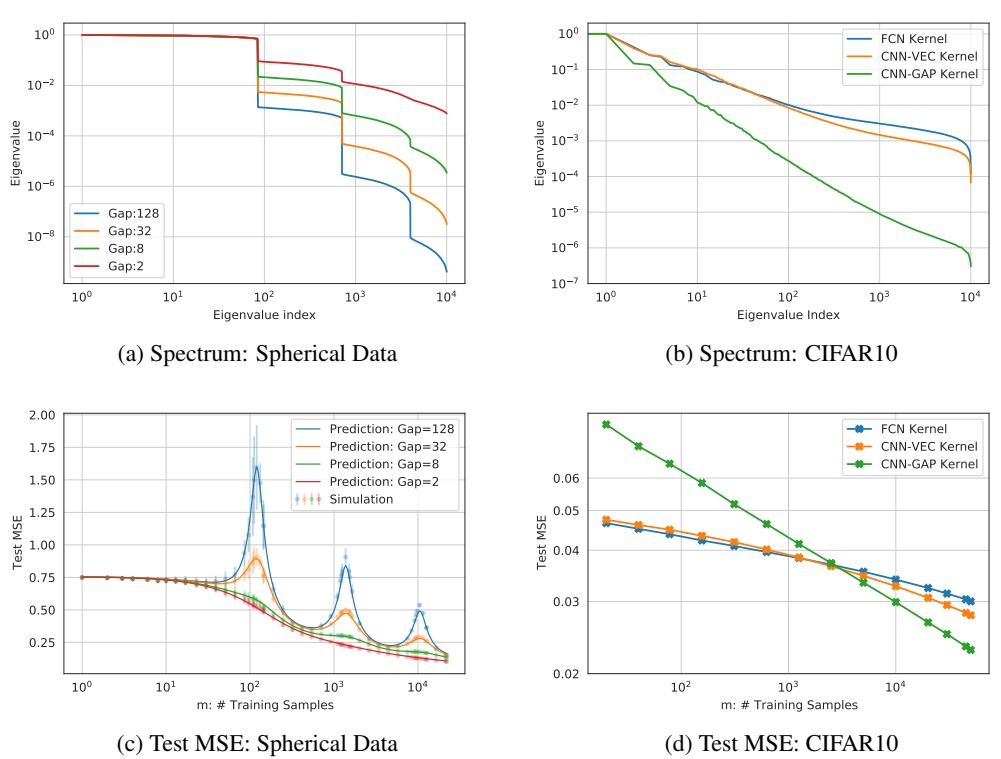

(a) Spectrum: Spherical Data

(b) Spectrum: CIFAR10

(c) Test MSE: Spherical Data

(d) Test MSE: CIFAR10

Figure 11: **Spectrum (top) and learning curves (bottom) of Spherical data (left) vs. CIFAR10 (right.)** The spectrum of CIFAR10 has a power-law decay and does not contain any sizable spectral gap, which is the main cause of the multiple-descent phenomena. The learning curves of CIFAR10 have power-law decay for all three kernels.