# OpenReview forum: "Precise Learning Curves and Higher-Order Scalings for Dot-product Kernel Regression  "
_NeurIPS.cc/2022/Conference — NeurIPS 2022 Accept_

### Official Review · Reviewer_Yvrw · 2022-07-08

**Rating:** 6
**Confidence:** 4
**Soundness:** 3 good
**Presentation:** 3 good
**Contribution:** 3 good

**Summary:**

Derive the precise generalization error of kernel ridge regression in the polynomial regime $n=\Theta(d^r)$ for dot-product kernels on unit sphere. The analysis generalizes [Mei et al. 2021] to the case where $r$ takes integer value. The key observation is that the degree-$r$ decomposition of the kernel matrix has a Marchenko-Pastur spectrum, which, together with a random label function assumption, gives a simple description of the generalization error similar to the case of linear regression. Theoretical predictions align with empirical findings.

**Questions:**

1. In the experiments, are the coefficients of the label function randomly sampled at each run? Do the theory still provide a good match if the coefficients are fixed?

2. Can the authors comment on the possibility of extending this analysis to other data distributions such as Gaussian?

**Strengths And Weaknesses:**

The generalization error of kernel methods in high dimensions is an important research problem. This submission considers the challenging polynomial scaling setting and completes the picture in [Mei et al. 2021] by covering the $r\in\mathbb{N}$ case. The asymptotic formulae provide a precise description of the multiple descent risk curve (without manipulating the spectrum of the input matrix). I feel that this is an interesting submission that is relevant to the NeurIPS community.

On the other hand, the current results are also a bit limited for the following reasons:
1. The analysis is based on the decomposition of kernel matrix into convenient orthogonal bases, which seems to work only for restricted input data (e.g. sphere or hypercube). It is not clear if the similar results can be shown for general settings.
2. To reduce the bias term to the Stieltjes transform of the decomposed kernel, the high-degree components of the label function is assumed to be random and isotropic. The authors should highlight this limitation in the abstract / introduction.

[Minor] the reference list for precise learning curves / Gaussian equivalence principle is incomplete. Please update the citations and include more relevant papers (e.g. see related work section of [Loureiro et al. 2021]).
Loureiro et al. 2021. Learning curves of generic features maps for realistic datasets with a teacher-student model.

---

> ### Author Response · Authors · 2022-08-02
> **Response**
>
> We thank the reviewer for the thoughtful questions and helpful suggestions. In what follows, we address the review's comments and the papers will be revised accordingly.
>
> 1. Assumptions on data distribution: We agree that current assumptions on input data is restrictive. In principle, similar results may still hold for other types of input data and kernel functions, as we can write out the spectral decomposition of the kernel function under given data distributions. One main challenge is that we do not have simple basis functions as in the sphere or hypercube cases. In order to obtain analytical formulas for the generalization errors, we need to know the expansion coefficients under the orthogonal basis. In the more general settings where we do not have simple basis functions, one possible way is to estimate these coefficients using the eigenvalues of the empirical kernel matrix. In terms of the rigorous proof, current technique is based on the specific forms of basis functions and cannot be applied directly, so we might need a different proof strategy.
>
> 2. Assumptions on high-degree coefficients: Thanks for this comment. We will highlight this restriction in the revised version.
>
> 3. Choice of coefficients in the experiments: In all the experiments, the coefficients are fixed although the theory part assumes random labels.
>
> 4. Extension to other input distributions such as Gaussian: Thanks for raising this point. We expect our technique could be extended to cover the case of Guassian in high dimension, although we may need to consider correction terms coming from approximating isotropic Gaussian by uniform sphere distribution in high dimension. On the other hand, going far beyond spherical-type of data distribution is mathematically challenging. This is because the spectral properties of general distributions in a high-dimensional setting is an underdeveloped area. We will leave this as an interesting topic for future research.
>
> 5. Incomplete reference list: Thanks for pointing out this. We will update the reference list in the revised version.

---

> > ### Comment · Reviewer_Yvrw · 2022-08-06
> > **Post-rebuttal update**
> >
> > I have read the authors' response, which addressed some of my concerns. Some followup comments:
> >
> > - The authors promised a set of changes regarding the explanation of the labeling assumption and references to related papers. I cannot find any of such changes in the revision. Please update the manuscript accordingly before the end of rebuttal period.
> >
> > - Regarding the new experiment on eliminating multiple peaks with regularization, Proposition 7 in [Wu and Xu 2020] showed that when the true coefficients are isotropic, appropriate $\lambda$ eliminates any non-monotonicity in the risk curve. My guess is that the underlying mechanism might be similar here, due to the random coefficients assumption and that the risk expression is parallel to the Gaussian design setting.
> > Perhaps the authors can comment on whether this is the case.
> > [Wu and Xu 2020] https://proceedings.neurips.cc/paper/2020/file/72e6d3238361fe70f22fb0ac624a7072-Paper.pdf

---

> > > ### Author Response · Authors · 2022-08-08
> > > **Update**
> > >
> > > We thanks the reviewer for the quick response and new comments.
> > >
> > > We have updated the paper to address the following.
> > >
> > > - Isotropic and random label assumptions. Please see the updated abstract and last paragraph of the intro.
> > >
> > > - References related to Gaussian equivalence conjecture / learning curves, please see line 30, line 44-46. Please also let us know if we still miss some other important references.
> > >
> > > Regarding [Wu and Xu 2020]. Thanks for bringing up this interesting paper. We agree on your point: "the underlying mechanism might be similar", as this similarity is owing to the "Gaussian equivalent conjecture". We added a brief discussion in Sec E.3 and copy-paste it here:
> > >
> > >  "A similar result (regularization eliminates multiple-descent) has also been observed in a linear design setting [Wu and Xu 2020]. The similarity between the linear design in [Wu and Xu 2020] and the nonlinear design here shouldn't be surprising, as we prove a "Gaussian equivalence conjecture," which implies that the polynomial scalings are essential "replicas" of linear designs with different scales. "
> > >
> > >
> > > Please let us know if you have other questions.

---

### Official Review · Reviewer_sFMD · 2022-07-11

**Rating:** 7
**Confidence:** 3
**Soundness:** 4 excellent
**Presentation:** 4 excellent
**Contribution:** 4 excellent

**Summary:**

This paper characterizes the prediction error for kernel ridge regression with dot-product kernels. Different from the previous work that focus on the linear scaling regime (sample size $m$ $\propto$ data dimension $d$), this work focuses on the higher-order scaling regimes ($m\propto d^r$). To establish this theoretical result, the author first study the limiting distribution (MP distribution) of the spectral density of the gram matrix under this scaling regime [Theorem 1]. With the help of this theorem, the author can express the testing error in terms of the MP distribution [Theorem 2]. Experimental results match their theoretical result. Extensions to convolutional kernels are also included.

**Questions:**

- The uniform distribution on the spherical type of data seems restrictive and existing literature also made similar assumptions. Could the author provide the literature review to compare assumptions on the data distributions for this problem?

**Limitations:**

See weakness part in previous section.

**Strengths And Weaknesses:**

**Originality**: Main theoretical result [Theorem 2] itself gives us an understanding of the learning curve of kernel regression for higher-order scaling regimes, extending previous analysis in literature. The intermediate theoretical result [Theorem 1] and supporting proof are also interesting. From my perspective, the key step is to view kernel function as harmonic series and then study the empirical spectral distribution of the non-trivial term from this series. Previous literature [Tao, 2012] only gives similar result for the case $r=1$, but the auhor shows that it holds for all degrees.

**Quality/Clarity**: Presentations of notations, theorem statements, and proofs are crystally clear. Theoretical results are highly non-trivial. Experimental results match their theory surprisingly well.

**Significance**: Their theory on learning curves applies for dot-product kernel and NNGP/NT kernel with one-layer convolution, which covers many scenarios in machine learning literature.

**Weakness**:
1. The author commented that it is promising to extend the theory for deep convolutional kernels, but due to the complicated structure of this type of kernel, it is left for future work. I agree on this point but I am looking forward to seeing some updates on deep convolutional kernels in the future.
2. As the author have commented in Section 7, the strong assumption on distributions of input data and the focus on kernel regression setting makes this work less popular. However, I believe similar technique can be used to handle more general settings, which can be finished in the future.

---

> ### Author Response · Authors · 2022-08-02
> **Response**
>
> We thank the reviewer for the positive comments and helpful suggestions. We address your questions below.
>
> The uniform distribution on the spherical type of data seems restrictive and existing literature also made similar assumptions. Could the author provide the literature review to compare assumptions on the data distributions for this problem?
>
> Response: Thanks for this question. We will add a literature review on high-dimensional asymptotics of kernel regression. Several papers have considered the linear scaling regime $m \propto d$ (see for example Bartlett, Montanari, Rakhlin (2021)). These papers use the linearization of the kernel matrix proved in El Karoui (2010), and can be applied to general distributions which have independent entries and some moment bounds. A more recent line of work started considering the polynomial scaling $m \propto d^r$: these apply under some abstract conditions on the eigendecomposition of the kernel, such as hypercontractivity of the top eigenvectors and concentration of the high frequency part of the kernel (see Mei, Misiakiewicz, Montanari, 2021). However, in order to be able to apply these abstract conditions to particular settings, one need to have access to the eigendecomposition of the kernel, which is in general intractable. Only a few examples are so far known: inner-product, group-invariant and convolutional kernels on the sphere, anisotropic product of spheres and hypercube (see Ghorbani et al. (2020,2021), Mei, Misiakiewicz, Montanari (2021), Xiao (2021)). It is an important open problem to extend these results to more general data distributions.
>
> Another line of work has instead considered statistical physics heuristics to derive these asymptotic formula for the prediction error (see e.g., Canatar, Bordelon and Pehlevan (2021)). More generally, behind all these asymptotic formulas is a general Gaussian equivalence conjecture, which seems quite robust across data sets (see “Learning curves of generic features maps for realistic datasets with a teacher-student model”, Loureiro et al. (2020)). One can consider our paper as a proof of this Gaussian equivalence conjecture in the particular case of inner-product/convolutional kernels on the sphere. There has been some progress on the Gaussian equivalence conjecture in the linear scaling (see Montanari and Saeed (2022)). Our results offer one of the first piece of evidence that such a Gaussian equivalence might extend to the polynomial scaling.

---

### Official Review · Reviewer_jzgb · 2022-07-11

**Rating:** 5
**Confidence:** 2
**Soundness:** 3 good
**Presentation:** 3 good
**Contribution:** 3 good

**Summary:**

The authors explicitly characterize the learning curve for kernel ridge regression under poly scaling regimes $m \sim d^r$, both theoretically and empirically. Despite the strict restrictions on the distribution and the setting, this interesting multiple-descent behavior in the sample-wise learning curve could help the understanding of neural networks.

**Questions:**

See above in weakness.

Update: the authors reasonably address my concerns and I have changed the scores.

**Limitations:**

Yes. As the authors pointed out in section 7, this works only focus on kernel ridge regression with uniformly distributed data, the generalization of the results is a very important topic. The impact of this work, beyond the interesting phenomena, should be more carefully explored.

I do not see what is the potential negative societal impact.

**Strengths And Weaknesses:**

Strength:
Theoretical results seem to be solid (did not check step by step though).
Experiments serve as good illustration of the key ideas.

Weakness:
1. (minor) readability could be improved by adding more details (could be placed at the beginning of the appendix in a separate (sub-)section):
(1) explicit definitions of asymptotic notations, e.g., $\sim$, $O(\cdot)$.
(2) background on decomposition via spherical harmonics

Moreover, are there (commonly seen) examples for the kernel function that eq. (1) is satisfied?

2. (major)The contribution/impact of the current finding should be more carefully articulated, instead of several lines in the intro (lines 54-56). Also a better summary of the existing works on both linear/non-linear scaling regimes would be better.

What is the connection (in terms of the empirical/theoretical behavior, technique in obtaining the theoretical results, etc.)?

3. (major)More experiments would be better: I would not say d=60 is very large (though I am not expert in this field). Since the goal is to demonstrate the (asymptotic) theoretical findings using numerical evidence, I would expect more extensive results (in terms of settings, such as dimensionality, distribution parameters, etc.).

For example, with larger dimension, would you observe better empirical scaling? To be precise, the authors state peak appears at $m \approx d^r/r!$, then I would expect experiments indicating as $d \rightarrow \infty$, the different between $m_{peak}$ and $d^r/r!$ shrinks to zero.

---

> ### Author Response · Authors · 2022-08-02
> **Response**
>
> We thank the reviewer for the insightful comments.
>
> - Regarding bullet points 1. We will incorporate your suggestion into the new version.
>
> - Regarding bullet points 2.  We will add more background and discussion of references on high-dimensional asymptotics for kernel ridge regression as suggested. In addition, we will also expand the discussion of the contribution/impact/limit of the current work in the introduction.
>
> - Regarding bullet points 3. We have added new experiments and related discussion in Sec E of SM.   $d=60$ may seem quite small at first glance. The challenge lies in the polynomial scaling (vs. linear scaling). For example, if we want to see the third peak (r=3), we need $m \sim 2 \times  d^3/3!$ (the 2x factor is to capture the descent after the peak) which is 72000; storing one $72000 \times 72000$ kernel would require >40 GB ram, while the largest kernel size we could linear-solve is around $25000 \times 25000$ using one A100.
>
> - Using $m_{peak} \sim d^r/r!$ to calculate the peak has a strong finite-width effect for small $d$, due to sub-leading order correction. We indeed derive the `exact` formula for the peak (without any correction), which is $m_{peak}=N(d, \leq r)$. We include an ablation study in Sec. E.2 to isolate this finite-size correction. Please also see the global response above.

---

### Official Review · Reviewer_HvxY · 2022-07-12

**Rating:** 7
**Confidence:** 4
**Soundness:** 3 good
**Presentation:** 2 fair
**Contribution:** 3 good

**Summary:**

The authors aim to theoretically analyze the relationship between number of samples ($m$) and test error (learning curve) in the case of Kernel regression for dot product kernels when the $m \sim d^r$ where $d$ is the dimensionality of data and $r$ is a natural number. Their analysis shows that when the input distribution and labeling function follow some regulatory conditions, under some asymptotics, a closed form formula precisely characterizes the learning curve for the mentioned set of kernels, and their experiments support this claim. Moreover, they asymptotically derive a closed-form distribution for the gram matrix of some dot product kernels, which seems valuable for future works.
Although the work indeed does have some limitations (mentioned in greater detail in the limitations section), both on the theoretical and experimental aspects, I find the contribution to be impactful in the direction of understanding/characterizing generalization error in modern machine learning.

**Questions:**

My questions have been mentioned in the weaknesses section.

**Ethics Review Area:**

["I don’t know"]

**Limitations:**

I have mentioned unaddressed limitations as part of the weaknesses.

**Strengths And Weaknesses:**

**Strengths:**

* The theoretical results of the work are sound and fills a gap in the study of learning curves for the case of non-linear relationship between the number of samples and dimensionality of the data.

* The predicted distribution for the spectral distribution of the mentioned dot product kernels seems very accurate and can potentially benefit future work.

* The conducted experiments, although with some limitations, strongly support the theoretical results.

**Weaknesses:**

* Some parts of the text are hard to follow. For instance, the notation section brings up the whole decomposition of the function of the dot product kernel without motivating it. This kind of introduction to some topic without motivating it beforehands happens a lot in the paper, and for me was generally confusing. Moreover, sometimes some equations or results are being referred to before being obtained or described. For instance, in line 171 there's a reference to Eq 29, which is undefined at this point for the reader. Likewise, $\chi_B$ and $\chi_V$ are introduced without much motivation.


* Various assumptions are not justified nor analyzed, and their importance could be better explained. As an example, the assumption of having according decomposable labeling function could be better explained in a sense that how close is it to the practical datasets that one might expect, at least for some toy examples.

* I believe the writing could enjoy more description about motivations, intuitions, the why of different assumptions and the relativity of them (how often do they hold in practical settings? why do we need them and what goes wrong if we don't have them?) instead of some extra formulas in the main text, such that the main text would give an overall intuition about the problem and the approach to solve it along with the derived theorems, and the proof sketchs could be delayed to the appendix instead. However, this is just my opinion.

* Although the experiments strongly support the theoretical results, the range of values used for $m, r, d$ seem a bit limited to me. In particular, there are some asymptotics in the theoretical results (high frequency and low frequency decompositions of the kernel value that converge or diverge asymptotically as $d \to \infty$) that I expected to result in very noisy experimental results, specially for the range of tested values, but surprisingly this doesn't happen in the provided experiments. I would be curious to know why this is the case. Moreover, experiments involving noise and ridge regularization could also benefit the paper.

* I believe more insights could be provided based on the solid theoretical results that are achieved. Having a closed formula for determining the precise learning curve in the mentioned context unlocks a lot of analyses. For instance, how does $r$ in $m \sim d^r$ affect the number of descents in the curve or the slope of the curve? How does the labeling function or the data affect the slope and/or the number of descents? The dependence on the spectral gap is mentioned, but when does one expect large or small spectral gap?

* Some more detailed concerns and questions:
  * What is the definition of a "fat" or "tall" matrix?
  * Why would one expect the coefficients of the legendre polynomials to be non-zero up to a large index in the decomposition of $h$, and why does it have to be independent of $d$?
  *  Could you please shed more light on the decomposition of the labeling function and the specific assumptions like the fact that $\hat f_k$ should be isotropic? How likely is it that practical datasets follow such decompositions?

---

> ### Author Response · Authors · 2022-08-02
> **Response**
>
> We thanks the reviewer for the detailed comments and kind suggestions. We will consolidate the suggestions in the first three points into the next version. In the meantime, we would like to focus on the technical parts of the review.
>
>  - Regarding "noisy experimental results".  The results are indeed quite noisy for all $d$ in the linear scaling regime; see the error bar in fig 1 and figure 4. The noises are reduced as the effective dimension (=d^r/r!) of the problem increases. We include a new subsection (E.1 of the appendix) to discuss the fluctuation vs input dimension. There are also two non-obvious improvements over naively using Theorem 2 that better compensate for the finite-size effect. We added Sec E.2 for an ablation study. Please also see the global response above.
>
> -  Regarding "Experiments involving noise and ridge regularization".  Thanks for the suggestion. We proved that the kernel regressor treats the unlearnable frequency signal as noise and the eigenvalues from such frequencies as “ridge”/regularization. As such, the regressor and the problem themself contain both “noise” and regularization, which is why we haven’t included related experiments in the initial submission. In the updated version, we run experiments to verify the role of regularization and how to choose it properly to eliminate one or more peaks in the multiple-descent curves. Please see Sec E.3 and the plots there for more details. We will include experiments involving noise in the next iteration.
>
> - We totally agree that our theoretical results, in particular, the precise learning curves,  unlocks a lot of analysis and can be used to answer the excellent questions you raised. For example, as mentioned above, we demonstrate how to choose the amount of regularization to eliminate any number of peaks in the multiple descent curves (Sec. E3). Regarding the spectral gap, it depends on both the activation functions and hyperparameters of the NNGP kernesl / NTKs, which can be evaluated numerically using the library Neural Tangents or analytically via decomposing the kernel functions into Fourier modes for many activation functions (e.g. Relu).
>
> - Fat vs Tall. For a $m\times n$ matrix, by a “fat” (”tall”) matrix, we simply mean $m \ll n$  ($m\gg n$)
>
> - "Why would one expect the coefficients of the legendre ...". This is a convenient assumption to simplify the presentation. The method allows any coefficients of the legendre polynomial to be zero. In this case, the corresponding Fourier modes associated with zero legendre coefficients are non-learnable and the kernel regressor will treat such modes as pure noise. Regarding the dependence of the decomposition of $h$ on $d$: yes, we could let them depend on $d$ as long as they converge to constants independent of $d$ as $d\to\infty$. We remove such dependence simply because we want to prioritize making the presentation cleaner.
>
> - "... decomposition of the labeling." The decomposition of the labeling function into Fourier modes (Legendre polynomials) is a very mild assumption as all $L^2$ functions have such decomposition. In practice, the labeling function is bounded (e.g. one-hot label) and bounded functions are in $L^2$ as long as the inputs are in a compact domain, which is always the case. The isotropic assumption (namely, random label) is a technical assumption required for our approach. The labeling function in practical datasets is always deterministic. Unfortunately, extending our results to deterministic labeling functions seems quite hard, which is still open even in the linear scaling regime [Mei&Montanari 2021]. We hope to share some progress in the future.

---

> > ### Comment · Reviewer_HvxY · 2022-08-08
> > **Thanks for the clarifications**
> >
> > I thank the authors for their response and am sorry about getting back to them a bit late.
> >
> > I would like to acknowledge that per my understanding:
> > * This work is still not readily applicable to describe learning curves in deep learning.
> > * I do not necessarily agree with the authors' response to reviewer jzgb that applying linear solve on bigger kernels is very hard for experiments, as one could use CPU computation for linear solve and fit way bigger kernel matrices, or use tools like https://gpytorch.ai/, thus, I believe further experimental evaluations could be added. This might sound like a general comment of "more evaluations please", but according to the reviewers' guide, we should value excellent experimental evaluation, and also, using larger $d$ or $r$ could help in further characterizing the noise in the experiments.
> >
> > These concerns prevent me from further increasing my score but I believe the contribution is already significant enough to have moderate to high impact in the recent line on analysis of high dimensional regression settings. I would be happy to change my mind if the authors disagree on the mentioned two points.

---

> > > ### Author Response · Authors · 2022-08-10
> > > **response**
> > >
> > > We thank the reviewer for the updates and for the new comments. Here are our answers.
> > >
> > > - Our results are applicable weakly to describe learning curves in deep learning due to the connection between neural kernels and the "lazy"/linearized training of neural networks. Obtaining learning curves for deep learning beyond this setting (e.g., in the feature learning regime) seems intractable to our best knowledge. Prior results cover only the linear scaling regime, and our results, more or less, represent the state-of-the-art knowledge we have regarding trackable learning curves.
> > >
> > > - Regarding scaling up. Our understanding is that most of the libraries (regarding linear solve) are `approximately` (not exactly) solving the (large) kernel regression. However, our experiments require solving the linear system `exactly` (with high precision) due to the need to capture low eigenvalue modes (e.g.,  the least eigenvalues in our experiments are of order $10^{-8}$ and the spectral gaps are of order $10^8$). Indeed, using float32 gives noisy and unreliable results even when the kernel size is ~ 10k. Mover, the new figure (Fig. 8) in the appendix empirically characterizes the dependence of the fluctuation/noise on $d$, which gives predictions for larger $d$. Finally, do you mind providing concrete values of $(d, r)$ that could help to clarify readers' concerns/questions? We are more than happy to run such experiments as long as they are feasible within our compute budget.
> > >
> > > Please let us know if you have further questions, and we look forward to hearing back from you. Thanks again for the new comments.

---

> > > > ### Comment · Reviewer_HvxY · 2022-08-10
> > > > **response**
> > > >
> > > > Dear authors,
> > > >
> > > > Thanks for the clarifications and the response.
> > > >
> > > > * I agree that the results can readily apply to the case of NTKs of networks with dense layers, but as the authors suggested, the kernels corresponding to networks involving more complex layers like convolution is not subject to rigorous analysis in this paper, but is somewhat delayed to future work (please correct me if I'm wrong). Regarding the learning curves for deep models, I agree that it seems very difficult to characterize, but I don't think I am knowledgable enough to judge if they are intractable or not.
> > > > * Regarding the approximation of fast linear solvers, at least to the best of my knowledge, GPytorch (Figure 2 - https://arxiv.org/pdf/1809.11165.pdf) and [Falkon](https://proceedings.neurips.cc/paper/2017/file/05546b0e38ab9175cd905eebcc6ebb76-Paper.pdf) claim to achieve preciese results in double precision. Moreover, a kernel of size 50k x 50k stored in double precision could still be solved by a machine that has 512GB of CPU ram with cholesky. Regarding figure 8, I'm sorry, I had missed that.
> > > >
> > > > Overall, after rethinking this thanks to the authors' clarifications, I think this work does have high impact in the line of theoretical analysis of high dimensional regression (and it's kernel regression, for some kernels at least) and it unlocks future directions of other analyses. Therefore, I will raise my score accordingly.

---

### Official Review · Reviewer_1Wx8 · 2022-07-18

**Rating:** 6
**Confidence:** 2
**Soundness:** 3 good
**Presentation:** 2 fair
**Contribution:** 3 good

**Summary:**

Summary.

The paper studies the kernel ridge regression and provides close form characterization on the loss curve in the finite sample regime of $m \propto d^{r}$, where $m$ is the sample size, d is the input dimension and $r \geq 1$ is arbitrary integer. The paper also proves the results hold broadly to the neural tangent kernel (NTK) of one-layer convolutional net.

Model.
The paper studies the classical kernel ridge regression under the assumption that input data draws from d-dimensional sphere and certain isotropic/independence assumption of higher order Fourier coefficients. While previous work of [Mei and Montanari 2021] considers mostly of linear regime, i.e., $m \propto d$, the paper extends the result to $m \propto d^{r}$ to arbitrary integer $r$.


Result.
The paper provides a close form characterization of the population loss. It is hard to describe in non-mathematical way, the close form characterization consists of Bias and Variance terms, where both terms are certain integral of Marchenko-Pastur distribution (limiting spectral distribution for random matrix) and exhibits the double descent phenomenon (bias always decay, but variance first goes up and then down)


Method

The method extends from the previous work of [Mei and Montanari 2021]. They use Legendre polynomial decomposition to the kernel and by standard concentration, argue one only needs to look at the $r$-th order decomposition. Then perform standard bias-variance decomposition and use random matrix theory to argue about the spectral. Some technical complications come out along the proof.


**Questions:**

1. Why the kernel result only apply to convolutional kernel but not other neural tangent kernel (say one-layer ReLU)?
2. I have difficulty understanding the setup. Line 76, why is $P_k$ a k-th order Legendre polynomial in 'd dimensions'? Isn't $P_k$ from $[-1,1]\to \mathbb R$? The equivalence between the Legendre polynomial and spherical harmonic representation looks non-travail to me, can you provide a reference for this?




**Limitations:**

.

**Strengths And Weaknesses:**

Strength. The paper provides close form characterization of loss curve for kernel regression and offers some insight on the double descent phenomenon. The technique follows largely from [Mei and Montanari 2021] but also turns out to be very interesting and non-trivial to me.

Weakness. There are no significant weaknesses but the writing could be significantly improved.  (See below)

1. Line 72. It is better to use $x^{\top}$ instead of $x^{T}$ for transpose. This applies to other matrix/vector transpose.
2. Line 46. “Extending these result “ - - > “Extending these results “
3. Line 40. “Large text corpora can contain trillions of tokens, wheares” I don't quite get it, the input dimension of language model might not be that large, but the entire models could still be quite large.
4. Eq. (5) what is $\delta_{0}(t)$.
5. Eq. (11) typo: need to add transpose for the variance and covariance.
6. Figure 3. Might be good to make bias and variance more clear.
7. Line 190. Would be good to make a separate paragraph for low frequencies and critical frequencies
8. The experiments are all simulations that should be emphasized somewhere.
9. The reference format seems wrong.
10. typo: Line 258 $y_k(\mathbf x)$ should be the summation over $j\in[d]$

---

> ### Author Response · Authors · 2022-08-02
> **Response**
>
> We thank the reviewers for carefully reading our paper. We have corrected the typos and the notations accordingly.
>
> - The dimension of the tokens and the length of the sequences are usually in the order of 1,000. We are using `sequence length x dim(token)` as the input dimensions, which is quite conservative. A less conservative estimate is to use `dim(token)`.
>
> - Applying to NTK /NNGP kernels?  Yes. The results apply to neural tangent kernel / NNGP kernels of fully-connected networks of any depth, as these kernels are inner product kernels; see, e.g., the last equation in page 3 of "Deep Equals Shallow for ReLU Networks in Kernel Regimes."
>
> - The equivalent between Legendre polynomial and spherical harmonic representation is indeed nontrivial, it is known as the Addition Theorem; please see Theorem 4.11 in the reference below. We will add discussion to unpack some background of this part in a newer version.
>
> Reference: SPHERICAL HARMONICS IN p DIMENSIONS; https://arxiv.org/pdf/1205.3548.pdf

---

### Author Response · Authors · 2022-08-02
**Global response**

We thank the reviewers for their careful reading and constructive and insightful feedback. Below we provide a list of the updates we made to the paper in response, and for convenience, we temporarily group them at the end of appendix Sec E.

- Finite-size effect. In Fig 8 of SM, we show that the mean of the simulations approaches the theoretical prediction and the standard deviation of the simulations approaches 0 as the input dimension $d$ increases. Here the simulations and the theoretical predictions are obtained using the second peak, namely, $m=N(d, \leq 2)$.

- Why does the agreement between theory and simulation work surprisingly well even when $d$ is not large? There are two non-obvious improvements over using theorem 2 as the theoretical predictions. These two improvements better account for the finite-size effect. (1) Using $m=N(d, \leq r)$ (obtained in the proof Eq (207)) instead of $m=N(d,r)$ to estimate the peak/critical scaling. The former exactly captures the peak while the latter only captures the leading term. (2) We use Corollary 1 rather than theorem 2 to simulate the theoretical prediction. Although they are equivalent when $d=\infty$, the former has a smaller finite-size correction owing to the sum over contributions from all critical scalings (vs. only one.) We conduct an ablation study; see Fig. 9 in Sec. E2.

- We agree with Reviewers HvxY that our results (closed-form formula for learning curves) unlock a lot of analysis. As one example, we added a discussion of the role of regularization. In Fig 10, we demonstrate that by choosing an appropriate regularization $\lambda$, one could eliminate any number of peaks in the learning curves. There are certainly many more analyses that can be done by exploring the learning curve formula and techniques of the paper, and we hope to share them either in a new version of the paper or in follow-up work.

- Comparison with natural data (CIFAR10.) We include a comparison study between the spectrum/learning curves of natural data vs. spherical data. As mentioned in the paper, the large spectral gap between eigenspaces triggers the multiple-descent phenomena. This phenomenon disappears, and the learning curve becomes monotonic when the spectral gap is small. Fortunately, for CIFAR10, the spectra of the (C)- NTK / C-NNGP are continuous and the learning curves are monotonic (power-law decay.) There is still a gap between our results/assumptions and natural data.

---

### Meta-Review · Area_Chair_f1hR · 2022-08-27

**Recommendation:** Accept
**Confidence:** Certain

**Metareview:**

The paper studies kernel ridge regression and characterizes its performance theoretically, which are interesting and highly relevant to machine learning venues. The paper is technically sound and the authors have done a good job in the rebuttal period. The paper is worth publishing in NeurIPS.


**Award:**

No

---

### Decision · Program_Chairs · 2022-09-14

Accept